# Silencing lipid catabolism determines longevity in response to fasting

Lexus Tatge [1], Juhee Kim [1], Rene Solano Fonseca [1], Kyle Feola[1], Jordan M. Wall[1], Gupse Otuzoglu[1], Ann C. Johnson[2,3], Kielen R. Zuurbier [1], Jaeyoung Oh[1], Shaghayegh T. Beheshti[1], Victor A. Lopez[1,4], Anthony J. Daley[5], Emma G. Werner[5], Patrick Metang[1], Sonja L. B. Arneaud [1], Abigail Watterson [1], Jeffrey G. McDonald [2,3], Vincent S. Tagliabracci [1,4,6], Michael E. French [3] & Peter M. Douglas [1,6] ✉

Oscillations between lipid anabolism and catabolism are essential for maintaining cellular health during metabolic fluctuations. Fasting, a conserved determinant of aging, improves disease outcomes and extends lifespan, yet the relative contributions of lipid catabolism versus its attenuation to fasting-induced longevity remain unresolved. The metabolic flexibility of *C. elegans* under variable nutrient availability provides a powerful system to address this question. We show that lifespan extension from fasting depends not on sustained activation of lipid catabolism, but on its silencing upon nutrient replenishment. The fasting-responsive nuclear hormone receptor NHR-49 activates β-oxidation; however, unlike classical ligand-regulated receptors, NHR-49 is regulated through ligand-independent mechanisms involving cofactor-mediated transcriptional attenuation and protein turnover. We identify casein kinase 1 alpha 1 (KIN-19) as a key regulator of metabolic plasticity and fasting-induced longevity that silences β-oxidation via primed phosphorylation of NHR-49. Thus, cooperative ligand-independent silencing of this conserved nuclear hormone receptor promotes fasting-associated longevity.

The coordinated sensing, storage, mobilization, and utilization of metabolic resources are essential for maintaining energy homeostasis across cells, tissues, and organs. When this regulation is impaired or metabolic transitions are inefficient, it can negatively impact physiology, leading to a variety of diseases and disorders[1]. Conversely, efficient metabolic transitions can benefit health and influence lifespan, as evidenced by reduced disease incidence and prolonged lifespan associated with fasting[2–6]. While macromolecules are distributed throughout the body via the circulatory system, cells within tissues have differential access to these systemic resources

due to variable feeding habits, vascularization, and absorption capacity[7]. Consequently, cells have developed autonomous surveillance strategies to monitor intracellular resources and adjust metabolic flux to meet their energetic demands. Understanding the complexity of these integrated signaling networks and how cells within tissues oscillate between metabolic states will provide valuable insights into overall physiology, disease progression, and the regulation of aging.

Caloric restriction is the most evolutionarily conserved method for lifespan extension, observed across species from budding yeast to

[1]Department of Molecular Biology, University of Texas Southwestern Medical Center, Dallas, TX, USA. [2]Department of Molecular Genetics, University of Texas Southwestern Medical Center, Dallas, TX, USA. [3]Center for Human Nutrition, University of Texas Southwestern Medical Center, Dallas, TX, USA. [4]Howard Hughes Medical Institute, Dallas, TX, USA. [5]Department of Chemistry and Biochemistry, University of Tampa, Tampa, FL, USA. [6]Hamon Center for Regenerative Science and Medicine, Dallas, TX, USA. ✉e-mail: peter.douglas@utsouthwestern.edu

primates[2,3,5,8–10]. However, rather than steady caloric reduction, dramatic oscillations in an animal's metabolic state, incurred by fasting, have proven more effective at maximizing the health and longevity benefits[3]. During prolonged fasting, cells rapidly deplete carbohydrates and rely heavily on the catabolism of lipids, the most energy-rich macromolecule, to sustain cellular functions[11–13]. Catabolism of lipids, known as lipolysis, involves the hydrolysis of acyl chains from glycerol backbones, which then undergo further catabolism through β-oxidation. This ultimately generates acetyl-CoA, a universal two-carbon metabolic intermediate utilized for adenosine triphosphate (ATP) production via the tricarboxylic acid (TCA) cycle or for the synthesis of essential macromolecules. Upon nutrient replenishment, cells suppress lipid catabolism and shift to an anabolic state to restore metabolic homeostasis[14,15]. However, the relative contributions of both the activation and subsequent attenuation of lipid catabolism in fasting-induced longevity remain poorly understood.

Herein, we report that silencing lipid catabolism is required for lifespan extension from fasting in *C. elegans*, while activation of this catabolic response during the initial starvation period appears relatively dispensable. We further define a ligand-independent mechanism of nuclear hormone receptor regulation during a fasted state that attenuates lipid catabolism upon nutrient replenishment. Although ligand binding influences the basal activity of the nuclear hormone receptor, NHR-49, it is not required for the transcriptional activation and subsequent silencing of β-oxidation genes during fasting and refeeding. Instead, NHR-49 is regulated by post-translational modifications. These modifications include a priming event in which phosphorylation at Serine 114 (S114) enables the Casein Kinase 1A1 ortholog, KIN-19, to phosphorylate adjacent residues, Serine 117 (S117) and Serine 120 (S120). The introduction of several negatively charged phosphates within the hinge region of NHR-49 can disrupt its chromosomal association through electrostatic repulsion with the negatively charged DNA phosphate backbone. In the absence of these phosphorylation events, NHR-49 can be atypically bound by XPO-1 and tethered to the nuclear pore for chromatin accessibility during times of stress. Thus, KIN-19 and this phosphorylation cascade on NHR-49 ensure metabolic plasticity by silencing β-oxidation gene expression upon the reintroduction of food. While NHR-49 activity is required for normal aging as well as several longevity-promoting interventions[16], hyperactivation of NHR-49 by limiting KIN-19 expression perpetuated a chronic state of lipid catabolism, abolishing the longevity benefits conferred by fasting. Our work provides a mechanistic basis for the attenuation of lipid catabolism in *C. elegans* and demonstrates its essential role in fasting-induced longevity.

## Results

### Dispensability of lipid catabolism in lifespan extension via fasting

Oscillating between catabolic and anabolic states is essential for cellular and organismal adaptation in an ever-changing environment. To model the breakdown of the animal's main energy reserve, we measured how triglycerides (TAGs) stored in intestinal lipid droplets[17] were altered after 24 h of fasting compared to a previously reported 48-h fast[18]. Thin-layer chromatography (TLC) revealed a 61% depletion in TAG levels at 24 h after fasting versus a 91% depletion after 48 h (Fig. 1a). Despite a more robust decrease in TAG levels at 48 h, the stark depletion of cholesterol raised concerns about the maintenance of membrane integrity and fluidity. Thus, the 24-h fast was selected for further experimentation as this regimen induced significant TAG catabolism and preserved cholesterol levels. Triglycerides are predominantly composed of long-chain fatty acids conjugated to their glycerol backbone[19–21]. Despite no apparent changes in total free fatty acids after 24 h of fasting by TLC, lipidomic profiling revealed selective reduction in specific long-chain fatty acids, including the α- and γ-linolenic acid, as well as oleic acid (Fig. 1b). In summary, fasting *C.*

*elegans* for 24 h triggers significant catabolism of TAGs and select long-chain fatty acids without altering the total pool of free fatty acids or cholesterol.

Next, we evaluated the animal's capacity to recover after fasting and restore lipid homeostasis upon dietary replenishment (see Supplementary Fig. 1a). Fasted worms were reintroduced to food for an additional 24 h, referred to as refeeding, and this was sufficient to restore free fatty acid profiles to pre-fasted states (Fig. 1b and Supplementary Fig. 1b). Moving forward, we used a fluorescence-based system to monitor, in real time, the dynamics of TAG-enriched lipid droplets in living animals during fasting and refeeding. We selected a well-characterized transgenic *C. elegans* strain expressing DHS-3::GFP, a lipid droplet-resident short-chain dehydrogenase fused to green fluorescent protein (Fig. 1c)[22–27]. Using the 24-h fasting and refeeding paradigm (Supplementary Fig. 1a), lipid droplet dynamics were monitored in relation to overall DHS-3::GFP fluorescence across a given worm population. Relative to Day 1 fed animals, the average volume and number of DHS-3::GFP positive lipid droplets after a 24-h fast were reduced by 84% and 65%, respectively, as determined by fluorescence microscopy (Fig. 1c and Supplementary Fig. 1c, d). This reduction in average volume and number corresponded to a 71% decrease in DHS-3::GFP fluorescence across the worm population as well as the 61% decrease in TAG levels observed by TLC (Fig. 1a, d). After refeeding for 24 h, both average volume and number of DHS-3::GFP positive lipid droplets recovered (Fig. 1c and Supplementary Fig. 1c, d), which correlated with the restoration of population-wide DHS-3::GFP fluorescence to pre-fasted levels upon refeeding (Fig. 1d). Thus, nutrient replenishment for 24 h after fasting enables restoration of neutral lipid levels.

The maintenance of cellular energetics is critical for survival during fasting periods[28]. As with mammalian adipose tissue, the primary energy reserves in *C. elegans* are stored within intestinal TAG-enriched lipid droplets[29]. During fasting, hydrolysis of TAGs liberates free fatty acids, which must be distributed to metabolically demanding peripheral tissues to be further catabolized via mitochondrial β-oxidation for adenosine triphosphate (ATP) production[30]. We sought to understand how animals manage energy homeostasis by redistributing resources provided by lipid catabolism during fasting and refeeding. To investigate this, we first examined transcriptional changes in mitochondrial-annotated genes. Gene pathway analysis after a 24-h fast revealed the transcriptional activation of genes involved in ATP biosynthetic process and a simultaneous repression of genes involved in energetically costly processes like mitochondrial transport and organization (Supplementary Fig. 1f). This coordinated response was fully reversed upon refeeding, demonstrating a rapid and reversible form of metabolic adaptation (Supplementary Fig. 1g). We then focused on the body-wall muscle, arguably the most energetically demanding tissue during fasting. Mitochondria in this tissue exhibit a higher membrane potential than mitochondria from other tissues and are poised to rapidly meet high ATP demands such as the dramatic increase in muscle contractions needed for fasting-induced foraging[31]. Using the established fluorescent reporter *myo-3*p::Queen-2m to monitor ATP levels in the body wall muscle[32], we detected a mild ~6% decline in ATP levels after 24 h of fasting, a level that recovered upon refeeding (Supplementary Fig. 1h). This ATP maintenance, coupled with sustained foraging activity (Supplementary Fig. 1i), demonstrates that these animals are capable of maintaining energy homeostasis and tissue function during the 24-h fast. Yet, despite their ability to sustain energetics and functionality in critical tissues, animals subjected to this fasting-refeeding cycle displayed lasting physiological changes, including a reduction in body size that persisted well into later life (Supplementary Fig. 1j, k). Our findings suggest that *C. elegans* exhibits significant metabolic plasticity, in which the modulation of lipid stores effectively accommodates energetic demand during periods of nutrient stress and subsequent recovery.

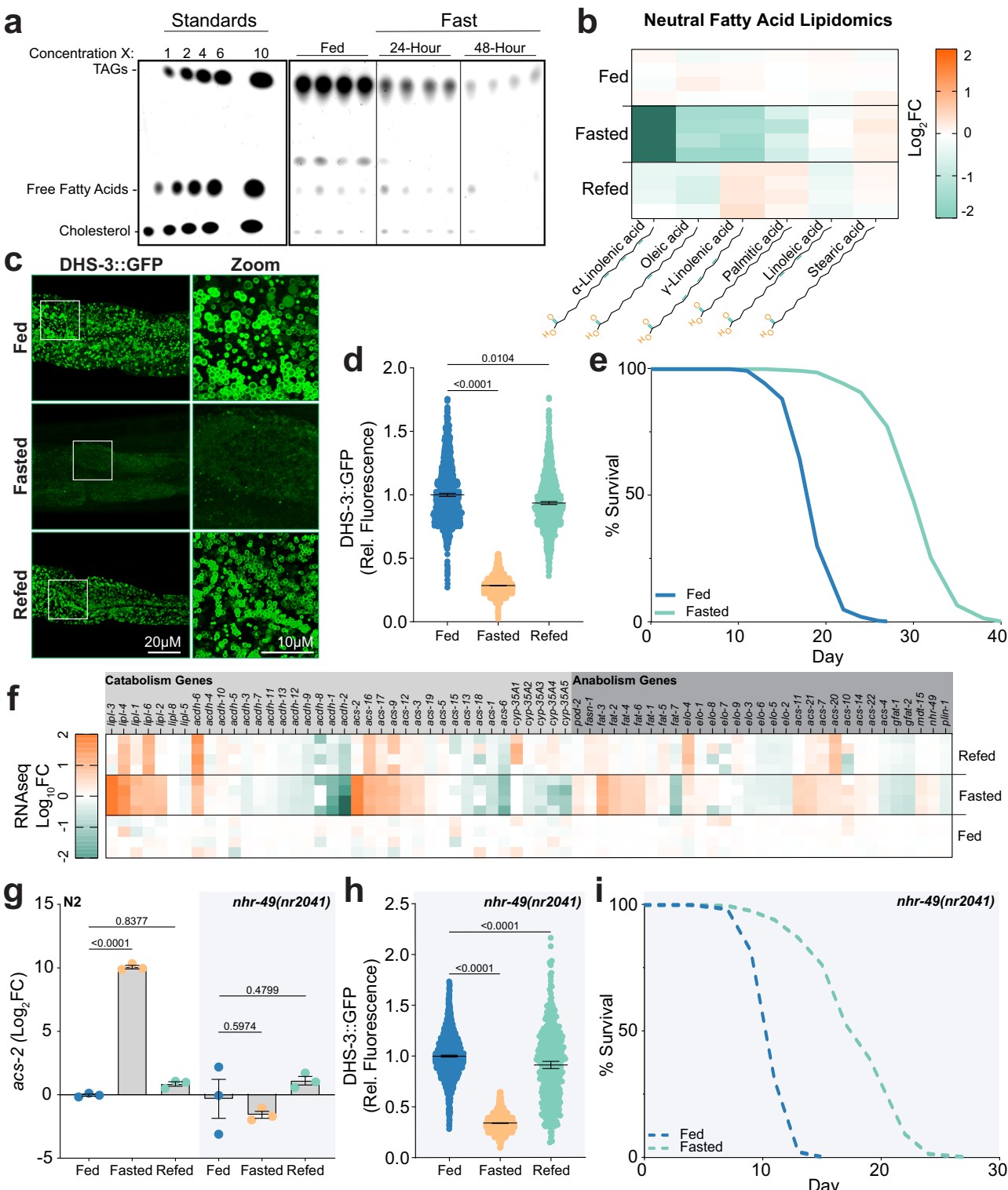

**a** Standards | Fast (Fed, 24-Hour, 48-Hour). Concentration X: 1 2 4 6 10. TAGs, Free Fatty Acids, Cholesterol.

**b** Neutral Fatty Acid Lipidomics. Fed, Fasted, Refed. Log$_2$FC. α-Linolenic acid, Oleic acid, γ-Linolenic acid, Palmitic acid, Linoleic acid, Stearic acid.

**c** DHS-3::GFP / Zoom. Fed, Fasted, Refed. 20μM, 10μM.

**d** DHS-3::GFP (Rel. Fluorescence). Fed, Fasted, Refed. <0.0001, 0.0104.

**e** % Survival vs Day. Fed, Fasted.

**f** Catabolism Genes / Anabolism Genes. RNAseq Log$_{10}$FC. Refed, Fasted, Fed.

**g** acs-2 (Log$_2$FC). N2 and nhr-49(nr2041). Fed, Fasted, Refed. 0.8377, <0.0001, 0.5974, 0.4799.

**h** DHS-3::GFP (Rel. Fluorescence). nhr-49(nr2041). Fed, Fasted, Refed. <0.0001, <0.0001.

**i** % Survival vs Day. nhr-49(nr2041). Fed, Fasted.

Dietary restriction is the most evolutionarily conserved method of lifespan extension[10]. Similar to other reported fasting and refeeding paradigms in *C. elegans*[18,33], we observe that 24 h of fasting in early adulthood was sufficient to extend median lifespan by 40.8% (Fig. 1e) and promote youthfulness, as evidenced by increased motility at older ages (Supplementary Fig. 1i). Analysis of the differentially regulated genes during fasting highlighted fatty-acid degradation as the most affected pathway (Supplementary Fig. 2a, b). Furthermore, fasting-induced fluctuations in lipid metabolism genes were restored upon refeeding. This restoration, similar to that of the mitochondrial-specific transcripts, suggests that the observed changes in lipid droplet availability are mediated, in part, by the transcriptional changes in these key metabolic enzymes (Fig. 1f). Using this fasting and refeeding paradigm, we next investigated the relative importance of lipid mobilization and restoration in fasting-induced longevity.

A substantial body of work establishes that the nuclear hormone receptor, NHR-49, is essential for activating and repressing the expression of genes involved in β-oxidation during starvation and plays a critical role in animal physiology and lifespan determination[16]. However, NHR-49's ability to recover after fasting,

**Fig. 1 | Dispensability of NHR-49 for fasting-induced lifespan extension.**
**a** Representative image of a TLC plate showing the lipid profiles of fed worms (control) and worms fasted for 24 and 48 h. Lanes 1-5 are standards in the form of TAGs, FFAs, and Cholesterol. *n* = 4 for each condition. **b** Heatmap depicts relative levels of long chain neutral fatty acids from C. elegans. Lipidomic analysis compares fed Day 1 adults to 24 h of fasting and subsequent refeeding for 24 h. *n* = 4.
**c** Fluorescence micrographs of C. elegans intestinal cells ectopically expressing the lipid droplet localized dehydrogenase, DHS-3::GFP under three conditions: Day 1 fed, Day 2 after 24-h fasting, and Day 3 after 24-h refeeding. Scale bars = 20 μm and 10 μm (zoom). *n* = 10 per group. **d** Scatter plot from large-particle flow cytometry of transgenic worms expressing DHS-3::GFP shows relative fluorescence per individual animal. Plot compares Day 1 adults under fed conditions to 24 h of fasting and subsequent refeeding for 24 h. Mean ± 95% CI. *n* = 1882, 3161, and 1538 from left to right, Kruskal-Wallis test and Dunn's multiple comparisons test used for statistics *p*.
**e** Lifespan analysis of control worms (N2) under fed or fasted conditions (24 h of dietary deprivation at Day 1 of adulthood). *p* < 0.0001 on 3 biological replicates determined by log-rank (Mantel-Cox) test. Additional statistics in Supplementary

Table 1. **f** Heatmap depicts relative transcriptional changes in lipid metabolism annotated genes. Analysis compares Day 1 adults under fed conditions to 24 h of fasting and subsequent refeeding for 24 h. Genes categorized by catabolic or anabolic based on their prominent annotated role in metabolism. **g** Relative abundance of acs-2 transcription determined by reverse transcription quantitative PCR (RT-qPCR). Analysis compares wild-type (N2) and nhr-49(nr2041) mutant worms at Day 1 of adulthood under fed conditions to 24 h of fasting and subsequent refeeding for 24 h. Mean ± SEM. *n* = 3, ordinary one-way ANOVA and Tukey's multiple comparisons test used for statistics. **h** Scatter plot from large-particle flow cytometry of nhr-49(nr2041) mutant worms expressing DHS-3::GFP shows relative fluorescence per individual animal. Plot compares Day 1 adults under fed conditions to 24 h of fasting and subsequent refeeding for 24 h. Mean ± 95% CI. *n* = 2436, 3072, and 430 from left to right, Kruskal–Wallis test and Dunn's multiple comparisons test used for statistics. **i** Lifespan analysis of nhr-49(nr2041) mutant worms under fed or fasted conditions (24 h of dietary deprivation at Day 1 of adulthood). *p* < 0.0001 on 3 biological replicates determined by log-rank (Mantel–Cox) test. Additional statistics in Supplementary Table 1.

and its subsequent impact on animal physiology and lifespan extension remained poorly understood. To this end, we first investigated whether NHR-49 activity could recover upon refeeding. Focusing on direct transcriptional targets of NHR-49 with its established roles in lipid metabolism[34–41], we confirmed that fasting-induced activation of the mitochondrial medium-chain acyl-CoA ligase (*acs-2*) and repression of the stearoyl-CoA desaturase (*fat-7*) were both abrogated in the loss-of-function *nhr-49(nr2041)* mutant animals (Fig. 1g and Extended Fig. 2c, d)[34,42]. In the same *nhr-49(nr2041)* mutants, we still observed a significant reduction in the levels of DHS-3::GFP positive lipid droplets during fasting, accompanied by a mild impairment in lipid droplet restoration upon refeeding (Fig. 1h). Therefore, TAG hydrolysis appears independent of NHR-49, which is further supported by the unaltered expression of critical lipolysis regulators, hormone sensitive lipase, *hosl-1*, and the adipose triglyceride lipase, *atgl-1*, in *nhr-49(nr2041)* mutant animals (Supplementary Fig. 2e). Since transcriptional dynamics dependent on NHR-49 appeared specific for genes involved in β-oxidation, we hypothesized that defective oxidation of the fatty acids would impair ATP maintenance in the *nhr-49(nr2041)* mutants. Indeed, ATP availability in these mutant animals was reduced by roughly threefold the levels observed in wild-type animals after fasting (Supplementary Fig. 2f). Despite deficits during fasting, ATP levels in mutant animals still recovered upon nutrient replenishment, indicating that other mechanisms may compensate to restore ATP levels upon refeeding. Thus, while transcriptional changes mediated by NHR-49 correspond with the maintenance of energy homeostasis during fasting, animals possessed the capacity to restore their ATP levels upon nutrient replenishment.

We next investigated whether metabolic deficits associated with *nhr-49(nr2041)* mutants impact fasting-induced longevity. While the mutants exhibited reduced lifespan under *ad libitum* feeding conditions[36], they demonstrated a notable 57.1% extension in lifespan when subjected to a 24-h fast at Day 1 of adulthood (Fig. 1i). These fasted mutants also exhibited physiological changes similar to those observed in wild-type fasted animals, including reduced body size and enhanced motility (Supplementary Fig. 2g–i). These data suggest that activation of β-oxidation is dispensable for lifespan extension via fasting. To further investigate the role of lipid catabolism in fasting-induced longevity, we examined two key regulators: Adipose Triglyceride Lipase-1 (ATGL-1), which initiates lipolysis, and Carnitine Palmitoyl Transferase-2 (CPT-2), which catalyzes carnitine removal and Coenzyme A conjugation to the fatty acid within the mitochondrial matrix[43,44]. Similar to the *nhr-49(nr2041)* mutant, fasting-induced lifespan extension was still observed in wild-type animals treated with *atgl-1* and *cpt-2* RNAi (Supplementary Fig. 3a, b). Moreover, administering *atgl-1* RNAi in the *nhr-49(nr2041)* background did not alter this

fasting-induced longevity effect (Supplementary Fig. 3c). Thus, disrupting multiple key steps in the lipid catabolism pathway had little impact on fasting-induced lifespan extension, indicating that the breakdown and utilization of lipids is not needed to confer the longevity benefits associated with fasting.

## Stress-induced ligand-independent regulation of NHR-49
Activating lipid catabolism appeared dispensable for fasting-induced lifespan extension. Therefore, we hypothesized that silencing this response upon refeeding plays a more significant role in longevity by restoring and preserving long-term lipid homeostasis. However, little is understood about how NHR-49 and its fasting-induced transcriptional response are attenuated upon dietary replenishment. While the activity of NHR-49 is regulated at multiple levels including cofactor interactions, heterodimerization with other nuclear receptors, and isoform-specific subcellular localization[23,42,45–48], nuclear hormone receptors are classically defined as ligand-regulated transcription factors[49–53]. Thus, we first examined the role of ligand binding as a means of regulating NHR-49 dynamics during fasting and refeeding. As an orphan receptor, no endogenous ligand has been identified for NHR-49. Previous studies identified an activating mutation within the ligand binding domain of NHR-49 at valine 411 where this hydrophobic valine was mutated to a charged glutamic acid[54,55]. Molecular modeling indicates that the substitution of a valine to glutamic acid at residue 411 reduces the binding affinity of linolenic acid, a reported exogenous ligand for the NHR-49 ortholog, HNF4α, as well as other potential fatty acid binding partners with comparable structure (Supplementary Fig. 4a, b)[56,57]. Due to the lipid-based nature of these putative ligands, we reasoned that nutrient replenishment would restore their intracellular abundance, thereby inactivating NHR-49.

While our efforts to define the endogenous ligand for NHR-49 were unsuccessful, mutating the ligand-binding pocket might allow us to determine the role of ligand binding in the transcriptional activity of NHR-49. Based on domain architecture comparison with HNF4α[58,59], we engineered an NHR-49 truncation (Δ295–422) to disrupt ligand binding by removing a majority of the ligand binding domain, while retaining its analogous N-terminal, DNA-binding, hinge, and C-terminal domains (Fig. 2a). Since the intestine is the primary site of neutral lipid storage in C. elegans[60], we generated both a full length and Δ295–422 truncated form of NHR-49::YFP isoform C under the control of the intestinal-specific *pept-1* promoter (Supplementary Fig. 4c, k). All experiments utilizing these ectopically overexpressed transgenes were performed in *nhr-49(nr2041)* mutant backgrounds to rule out contributions from the endogenous NHR-49 gene. Importantly, the full-length NHR-49::YFP fusion protein restored median lifespan of the *nhr-49(nr2041)* mutant to levels typically observed in wild-type animals, and fasting

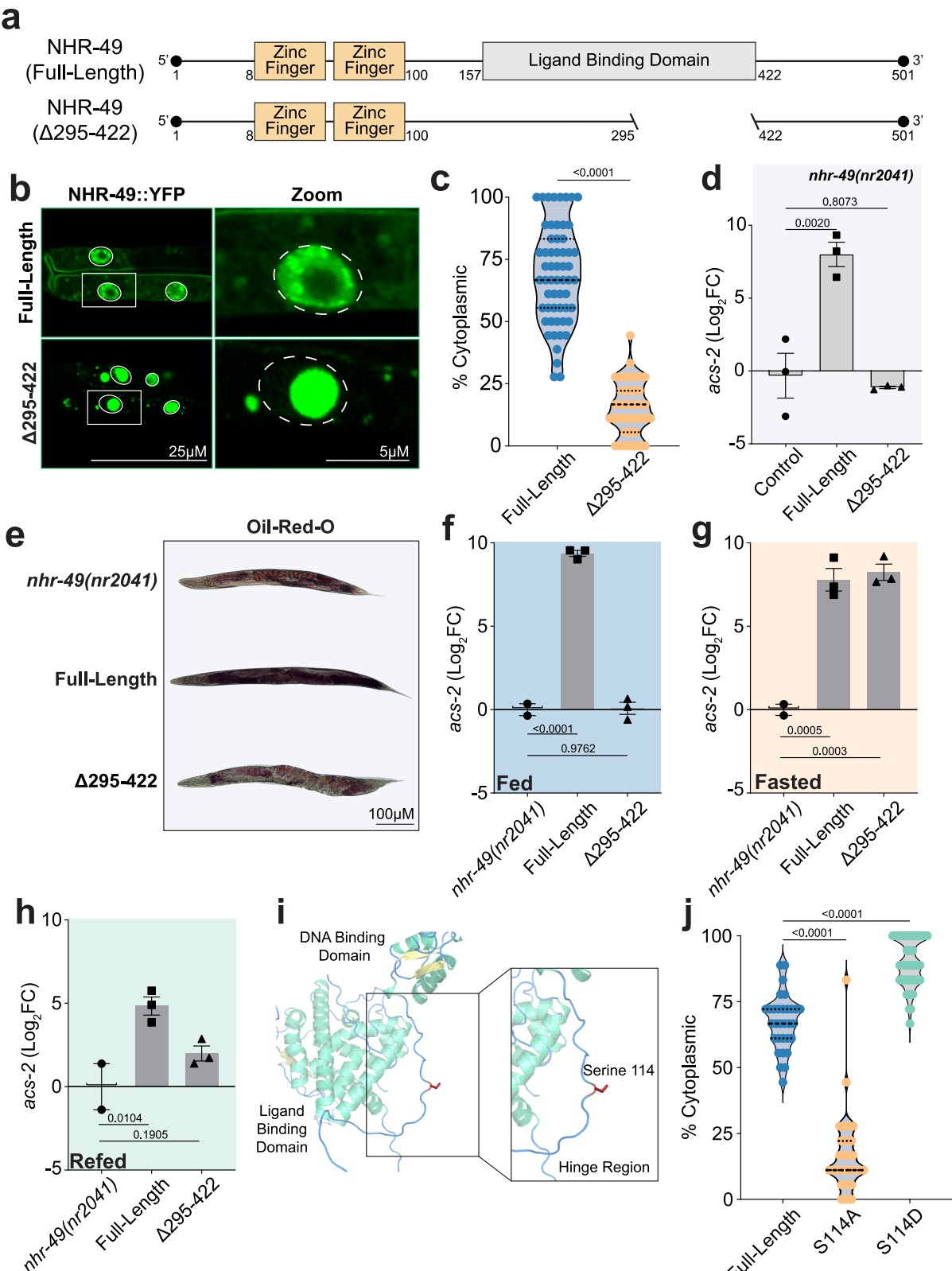

for 24 h further prolonged lifespan (Supplementary Fig. 4d). Therefore, the full-length isoform C transgene can functionally rescue the *nhr-49(nr2041)* mutant.

We next investigated the NHR-49::YFP Δ295–422 truncation to better understand how impaired ligand binding impacts its subcellular dynamics and transcriptional activity during fasting and refeeding. Consistent with other reports regarding C-terminal truncations of its

ortholog HNF4α[61], truncating the ligand-binding domain of NHR-49 altered its nucleocytoplasmic distribution, favoring a more pronounced nuclear localization under *ad libitum* feed conditions when compared to the full-length receptor (Fig. 2b, c). Despite its increased nuclear occupancy, the Δ295–422 receptor failed to maintain baseline expression of NHR-49 transcriptional targets, *acs-2* and *fat-7* (Fig. 2d and Supplementary Fig. 4e) and was incapable of phenotypically rescuing

**Fig. 2 | Ligand-independent regulation of NHR-49. a** Schematic of NHR-49 construct containing a deletion of amino acids 295–422 (Δ295–422). **b** Fluorescent micrographs of transgenic worms in L3-L4 larval stages ectopically expressing NHR-49::YFP (full length) or the ligand binding truncation (Δ295-422) in intestinal cells. Scale bar = 25 μm. **c** Quantification of cytosolic NHR-49::YFP by visual inspection. $n = 58$ and 51 from left to right over 3 independent trials. Unpaired $t$-test (two-tailed) used for statistics. **d** Relative abundance of acs-2 transcription determined by RT-qPCR. Analysis compares nhr-49(nr2041) mutant worms ectopically expressing NHR-49::YFP full-length or Δ295-422 in the intestine. Mean ± SEM. $n = 3$ independent trials, ordinary one-way ANOVA with Tukey's multiple comparisons test used for statistics. **e** Micrographs of stained neutral fatty acids by Oil-Red-O staining in nhr-49(nr2041) mutant worms ectopically expressing intestinal NHR-49::YFP full-length or Δ295-422 at Day 1 of adulthood. **f–h** Relative abundance of acs-2 transcription determined by RT-qPCR. Analysis compares nhr-49(nr2041) mutant worms ectopically expressing NHR-49::YFP full-length or Δ295-422 in the intestine during **f** fed, **g** fasting, and **h** refeeding conditions. Mean ± SEM. $n = 3$ independent trials, ordinary one-way ANOVA with Tukey's multiple comparisons test used for statistics. **i** Predicted secondary structure of NHR-49 modeled using AlphaFold2. Stick rendition of serine residue 114 (red). **j** Quantification of cytosolic NHR-49::YFP by visual inspection. $n = 31$, 35, and 64 from left to right over 2 independent trials. Ordinary one-way ANOVA with Šídák's multiple comparisons test used for statistics.

nhr-49(nr2041) mutant defects with respect to lipid deposition and fecundity (Fig. 2e and Supplementary Fig. 4f, g). Although atypical, other groups have reported ligand-independent means of nuclear hormone receptor regulation[62], and we further investigated whether this might be the case for NHR-49 during fasting conditions. Indeed, ectopic expression of the Δ295–422 receptor displayed similar transcriptional dynamics as the full-length receptor when subjected to fasting and refeeding as evidenced by the activation and subsequent recovery of acs-2 transcription (Fig. 2f–h). Thus, unlike ad libitum fed conditions, ligand binding was dispensable for both the activation and attenuation of NHR-49 upon fasting and refeeding. Furthermore, these NHR-49-mediated transcriptional fluctuations originated within the intestine.

Nuclear hormone receptors possess several ligand-independent modalities of regulation[63]. For instance, these receptors, including NHR-49, can alter their activity through heterodimerization with other nuclear receptors[46]. However, the inability of the Δ295–422 receptor to rescue baseline transcription (Fig. 2d and Supplementary Fig. 4e) and lipid-related phenotypes (Fig. 2e and Supplementary Fig. 4f, g) under ad libitum fed conditions indicates that it is not functioning through heterodimerization with other nuclear receptors in the intestinal cell. In further support, no nuclear receptor binding partners were detected by mass spectrometry of either the NHR-49::YFP full-length or Δ295–422 immunoprecipitations (see Supplementary Data 1). Alternatively, post-translational modifications have the potential to impact nuclear receptor function[62]. Starvation-induced phosphorylation of HNF4α by Protein Kinase A decreases DNA binding and reduces transcriptional activity[64]. While previous studies hint at possible kinase-regulated mechanisms for NHR-49, they lack residue-level specificity[16]. Utilizing Liquid Chromatography-Tandem Mass Spectrometry (LC-MS/MS), we performed post-translational modification analysis on NHR-49::GFP immunoprecipitations[42] under fed and fasted states and identified a single phosphorylation event with high confidence within the receptor's hinge region at serine 114 (Fig. 2i and Supplementary Fig. 4h). This S114 phosphorylation was further confirmed by proximity labeling with the biotin ligase tag, TurboID, which was inserted, in independent worm strains, at both the N- and C-terminus of the endogenous NHR-49 gene[48]. All proteins within an approximate radius of 10 nm[65] to either tagged NHR-49 fusion protein were biotinylated, affinity-purified, and subjected to LC-MS/MS[48]. Following post-translational modification analysis, we again detected phosphorylation at S114, with 15 PSMs in both the N- and C-terminal TurboID strains, ranking it as the 5th most abundant PTM overall. We also identified a phosphorylation site at S131 with a PSM of 2, observed only in the C-terminal strain. An additional low-confidence phosphorylation was detected between residues 110 to 128, though it could not be precisely localized, potentially due to its transient nature (Supplementary Fig. 4i). Thus, the serine residue at position 114 of NHR-49 and potentially others within the hinge-region of NHR-49 were detected across multiple experiments which used both ectopically overexpressed and endogenously tagged forms of NHR-49.

While HNF4α possesses several serine residues within its analogous hinge region, its paralog, HNF4γ, shares more sequence similarity to NHR-49 and exhibits a higher degree of conservation at the respective serine residue and its surrounding motif (Supplementary Fig. 4j). With respect to physiology, phosphorylation and cytosolic localization of HNF4α are linked with liver decompensation and cirrhosis, however evidence is lacking on the effect of phosphorylated HNF4γ[66]. Based on the supporting evidence in mammals, we sought to understand the significance of this putative phosphorylation event in worms. To this end, we first abrogated phosphate conjugation by mutating the serine residue at the 114 position to an alanine (S114A) within the ectopically expressed NHR-49::YFP transgene. The subcellular distribution of the S114A mutant favored a more pronounced nuclear localization with significantly less fluorescence in the cytosol compared to the full-length transgene (Fig. 2j and Supplementary Fig. 4k, l). In a complementary manner, an activating phosphomimetic mutation was engineered to permanently mimic the phosphate's bulky negative charge by replacing the serine with an aspartic acid (S114D). This phosphomimetic, S114D mutant, in NHR-49::YFP strongly favored a cytoplasmic localization, which was dramatically different from the wild-type and more so from the phospho-dead S114A mutant (Fig. 2j and Supplementary Fig. 4l). These results are consistent with the reported role of hinge-region serine phosphorylation in the cytoplasmic retention of nuclear hormone receptors[67].

## NHR-49 phosphorylation by the casein kinase KIN-19

After identifying a putative phosphorylation site at serine 114 and confirming its significance in controlling the subcellular dynamics of NHR-49 through mutational analysis, we next aimed to define the kinase responsible for this post-translational modification. To this end, we employed a multi-pronged strategy, which involved proteomics analysis, genetic screening, in silico binding, sequence motif validation, and in vitro confirmation. Building upon our extensive proteomics data sets, our initial analysis sought to filter kinases that were repeatedly detected in complex with and/or in proximity to NHR-49. We cross-referenced coimmunoprecipitation and proximity labeling data sets, which accounted for three different NHR-49 strains including the overexpressed all-tissue NHR-49::GFP under fed and fasted conditions[42] and the endogenous N- and C-terminal TurboID tagged NHR-49[48]. From these datasets, 96 kinases were detected in immunoprecipitations of overexpressed NHR-49::GFP in all tissues and 41 in the proximity labeling experiments with N- and C-terminal TurboID tagged NHR-49. Of these combined 137 kinases, 22 were mutually detected in both experiments (Supplementary Fig. 5a). Despite the limitations associated with identifying bona fide protein-protein interactions by coupling co-immunoprecipitation or proximity labeling with LC/MS-MS, we were able to confidently generate a shorter, more manageable list of 19 potential interacting kinases, predominantly intestinal, for further study.

Next, we coupled AI-based predictive modeling with RNAi screening to filter kinases with the potential to directly bind NHR-49 and alter its activity. First, we computationally modeled the interactions between NHR-49 and each of the 19 kinases, assuming a one-to-one stoichiometric ratio (see Supplementary Table 2). With an interface predicted templating model (ipTM) score greater than 0.6 as our confidence threshold, none of the modeled kinase/NHR-49

interactions scored over 0.5. While lacking confidence in an interaction does not preclude a genuine association, it highlights that no such interaction between these kinases and similarly structured nuclear receptors has been previously reported. In parallel, we employed a more labor-intensive, candidate-based RNAi screen in vivo to determine whether reduced expression of individual kinases impaired NHR-49 associated phenotypes. Complementary results from phospho-mimetic (S114D) and phospho-dead (S114A) forms of NHR-49::YFP indicate that S114 phosphorylation antagonizes nuclear residency of the receptor (Fig. 2j and Supplementary Fig. 4l). Therefore, we hypothesized that hindering S114 phosphorylation by RNAi depletion of a kinase would enhance the transcriptional activity of NHR-49. To test this, we used transgenic animals expressing the transcriptional *rab-11.2*p::YFP reporter, which serves as an effective fluorescent diagnostic of NHR-49 activation, though seemingly not a direct transcriptional target[23,42]. Exhibiting a broad range of fluorescent detection and ranking among the most activated genes during nutrient deprivation, we leveraged this reporter to systematically screen all 19 kinases. Two enzymes were identified whose reduced expression activated the *rab-11.2* fasting reporter KIN-2, a cAMP-dependent protein kinase β-subunit (PKA), and to a much greater extent, KIN-19, the ortholog of casein kinase 1 alpha 1 (CSNK1α1) (Fig. 3a and Supplementary Table 3). Through complementary approaches involving the analysis of multiple proteomics datasets and subsequent functional RNAi screening, we identified two potential kinases as candidate regulators. However, these candidates were absent from the initial in silico interaction networks, prompting us to investigate this discrepancy.

Protein kinases exhibit distinct and highly specific consensus binding motifs that dictate their enzymatic phosphorylation activity. To investigate the phosphorylation of NHR-49's S114 residue, we compared the consensus motifs of the highly conserved human orthologs of KIN-2 and KIN-19. Although neither kinase was predicted to phosphorylate S114 directly, a highly conserved sequence neighboring S114 was strongly predicted to be CSNK1α1 (KIN-19) phosphorylation sites (Supplementary Fig. 5b). Casein kinases are a broad family of serine/threonine kinases involved in diverse cellular processes, including signal transduction, circadian rhythm regulation, and metabolic pathways[68,69]. Substrates of casein kinase typically require a priming phosphorylation event to catalyze phosphate addition[70–72]. Sequence motif analysis indicated that KIN-19 would preferentially phosphorylate the conserved serine residues adjacent to S114 at positions S117 and S120 (Supplementary Fig. 5b)[73]. Thus, phosphorylation at S114 may act to prime NHR-49 for further phosphate addition by KIN-19. In support, AI-based structural algorithms[74] now predicted a significant interaction between KIN-19 and S114-phosphorylated NHR-49 with an ipTM of 0.69 (Fig. 3b, Supplementary Fig. 5c, d and Supplementary Table 2). To determine whether this was specific for KIN-19, we tested the other 18 kinases that were previously screened against the primed S114 and found only KIN-19 as a significant predicted interaction. In summary, a comprehensive analysis of 19 kinase candidates, which integrated LC-MS/MS data from several co-immunoprecipitations, in silico modeling, RNAi screening, and consensus sequence validation, identified KIN-19 as the most promising kinase candidate for further investigation.

Casein kinases are reported to act on a wide range of molecular targets, and we sought to determine whether KIN-19 could physically modify NHR-49. Using the *C. elegans* KIN-19 ortholog, CSNK1A1, we conducted [$^{32}$P] kinase-substrate in vitro assays. The human ortholog was chosen for its high sequence identity (87%, 274/315) and conserved secondary structure (RSMD = 0.220), ensuring functional equivalence to KIN-19 (Supplementary Fig. 5e). This assay utilized recombinant human CSNK1A1 and peptides corresponding to the hinge region of NHR-49 as the substrate to test for direct phosphorylation. Consistent with in silico modeling, CSNK1A1 exhibited basal phosphorylation activity on the unprimed NHR-49 peptide ($k_m$ = 106.6; $V_{max}$ = 10.98). However, pre-

phosphorylation of the NHR-49 peptide at the corresponding S114 (pS114) position markedly enhanced transfer of the γ-phosphate from [$^{32}$P] ATP onto the primed peptide ($k_m$ = 285.7; $V_{max}$ = 68.63) (Fig. 3c and Supplementary Fig. 5f). Pre-phosphorylation of the serine residue resulted in a 2.68-fold increase in the Michaelis constant ($K_m$) and a 6.25-fold increase in the maximum reaction rate ($V_{max}$) when compared to the unprimed site. This observation is consistent with previously reported mechanisms of the casein kinase 1 family, which are capable of phosphorylating non-primed substrates but do so far more efficiently when an upstream priming phosphorylation is present[75–77]. Specifically, casein kinase 1 family members are reported to act on substrates via sequential phosphorylation of serine/threonine residues following a canonical pSxx(S/T) motif. This priming-dependent enhanced phosphorylation aligns with our in vivo post-translational modification analysis: S114 phosphorylation is consistently detectable, whereas downstream residues S117 and S120 remain elusive, likely due to the transient nature of their phosphorylation following S114 priming.

To identify the kinase responsible for phosphorylating S114, we revisited our analytical pipeline. The second kinase that activated our fasting reporter, KIN-2, is the ortholog of PKA, which has previously been shown to phosphorylate HNF4α during fasting[64]. However, we did not detect KIN-2 as a conserved interactor in our co-immunoprecipitation assays, and its consensus phosphorylation motif does not align with residues surrounding S114. Conversely, while MPK-1's consensus motif closely matches the residues surrounding S114, it failed to activate our fasting reporter (Fig. 3a). Despite KIN-19 being the only kinase to satisfy our initial criteria, we performed in vitro kinase assays using [$^{32}$P] ATP with the human orthologs of both KIN-2 (PKA) and MPK-1 (ERK1). Both kinases yielded non-significant phosphorylation results for the peptide corresponding to the hinge region of NHR-49 (Supplementary Fig. 5g–j). These findings validated our initial selection, leading us to focus on the further characterization of the KIN-19/NHR-49 phosphorylation event.

Given the essentiality of KIN-19, we focused our investigation on its functional role. Repeated attempts to engineer an enzymatically dead kinase via a D135A mutation were unsuccessful, strongly suggesting that this kinase and its enzymatic activity are vital for animal viability. Therefore, we proceeded with RNAi to knockdown KIN-19 expression. We confirmed the knockdown efficiency with a 79% reduction in transcript levels and approximately a 95% reduction in steady-state protein levels of endogenous KIN-19, as determined by RNA-sequencing and whole-worm proteomics, respectively (Supplementary Fig. 5k, l). Animals treated with *kin-19* RNAi exhibited a gene expression profile characteristic of fasting-induced NHR-49 activation, including the differential regulation of established NHR-49 targets, *acs-2* and *fat-7* (Fig. 3d and Supplementary Fig. 5m). Gene Ontology (GO) analysis of differentially regulated genes revealed that the most significantly altered categories were related to lipid transport and metabolism (Supplementary Fig. 5n). Cross-referencing the transcriptional changes induced by *kin-19* RNAi with those induced by fasting showed a significant overlap in genes differentially regulated between the two conditions, including 20 partially dependent NHR-49 targets beyond *acs-2* and *fat-7* (Fig. 3e, f). Based on these transcriptional signatures, impairing KIN-19 expression appeared to mimic a chronically-fasted state in which NHR-49 is hyperactivated. While the precise nature of the priming event at S114 remains unclear, our findings demonstrate that the casein kinase ortholog, KIN-19, is capable of phosphorylating adjacent serine residues in the hinge region of NHR-49. This ultimately reveals that impairing KIN-19 function via RNAi mimics a perpetually fasted response.

## Silencing lipid catabolism is required for fasting induced longevity

Next, we investigated the impact of KIN-19 gene silencing on NHR-49 regulation and its subsequent effects on energy homeostasis, animal

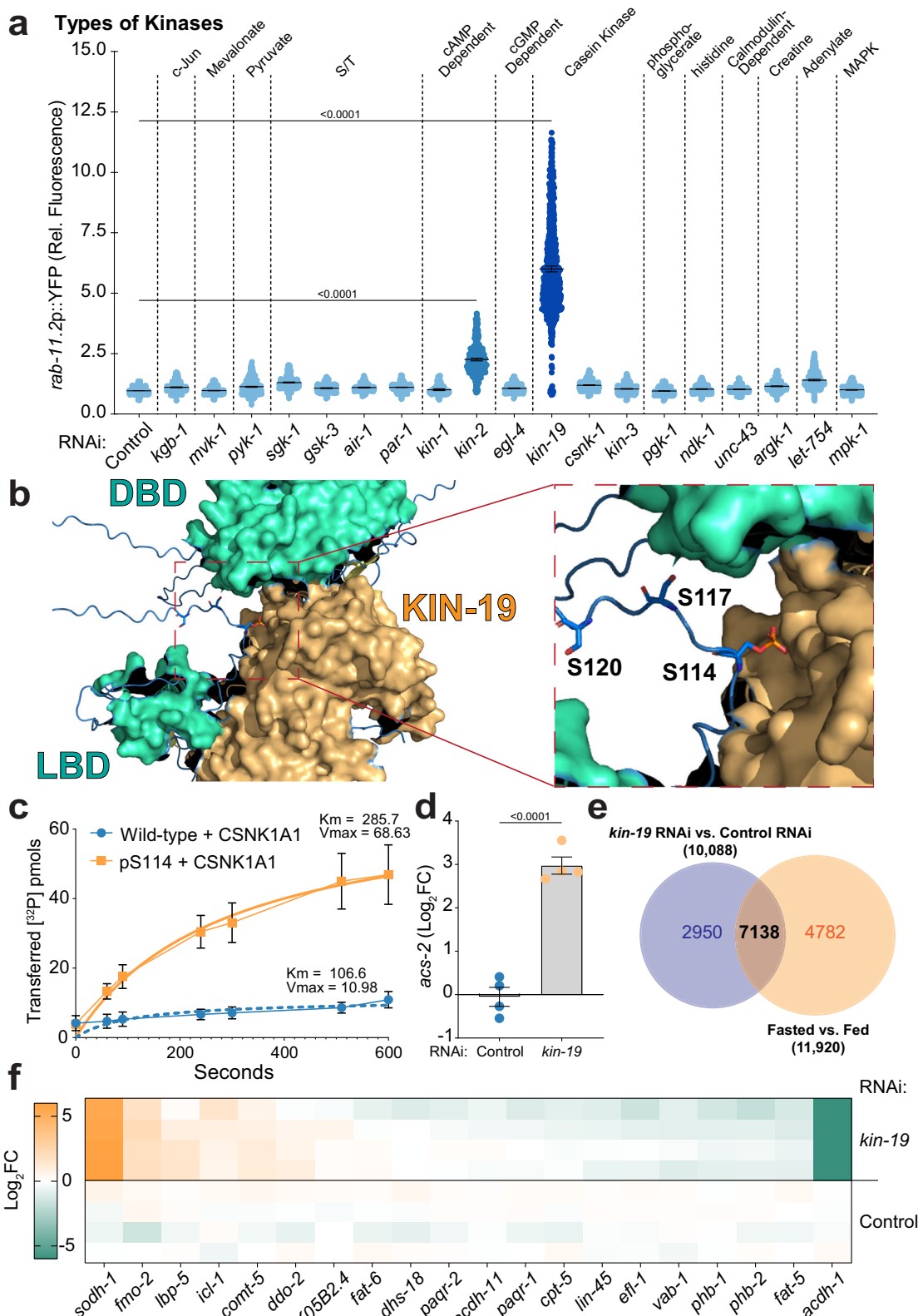

physiology, and age determination. Previous research links the worm casein kinase 1A1 to oocyte development, asymmetric cell division, and embryogenesis[78,79]; while its function in adulthood and more particularly in metabolism and aging, remains largely unexplored. Though some studies report that KIN-19 overexpression can lead to extensive protein aggregation in older worms[80–82], our analysis of Day 1 adult worms revealed no evidence of endogenous KIN-19

aggregates (Supplementary Fig. 6a). Our study found that *kin-19* RNAi in adult worms resulted in several physiological changes consistent with chronic fasting. We observed a significant reduction in lipid deposition and body size, traits typical of fasted animals (Fig. 4a–c and Supplementary Fig. 6b). Additional transcriptomic analysis of lipid metabolism or mitochondrial-associated genes revealed similar transcriptional profiles between *kin-19* RNAi treated

**Fig. 3 | NHR-49 phosphorylation by casein kinase, KIN-19. a** Scatter plots from large-particle flow cytometry of transgenic worms expressing rab-11.2p::YFP show relative fluorescence per individual animal. Plots compare Day 1 adults under empty vector control conditions to the respective RNAi conditions. Mean ± 95% CI. $n = 1067, 664, 1017, 755, 563, 408, 333, 1033, 131, 536, 799, 1023, 555, 631, 1036, 853, 566, 633, 521$ and $886$ from left to right, ordinary one-way ANOVA with Dunnett's multiple comparisons test used for statistics (see Supplementary Table 3). **b** Predicted secondary structure and intermolecular interface of KIN-19 and NHR-49, modeled using AlphaFold3, an advanced AI-driven deep learning algorithm for protein structure and interaction prediction. Surface density for KIN-19 (orange) and NHR-49 (teal) with stick rendering of regulatory serine residues. **c** Quantification of γ-phosphate transferred from [$^{32}$P] ATP to either wild-type (blue) or pS114 (teal) primed NHR-49 polypeptides by recombinant CSNK1A1 over 10 min, taken over three independent repeats. A non-linear fit to each line. Error bars represent SEM. **d** Relative abundance of acs-2 transcripts determined by RNA-sequencing. Analysis compares Day 1 adults on an empty vector control and kin-19 RNAi. Mean ± SEM. $n = 4$, statistics represent Differential Expression Analysis in Two Groups with FDR correction (Qiagen CLC Workbench v9.5). **e** Venn diagram highlighting the overlap between transcriptional changes observed in animals treated with kin-19 RNAi and those fasting for 24 h. **f** Heatmap depicts relative transcriptional changes in genes previously reported to be partially regulated by NHR-49. Analysis compares Day 1 adults on an empty vector control and kin-19 RNAi conditions.

and fasted animals (Supplementary Fig. 6c, d). These findings demonstrate that silencing KIN-19 triggers the physiological hallmarks of fasting even when animals are under *ad libitum* conditions, indicating that KIN-19 plays a critical role in mediating the fasting-induced response impacting both metabolic and physiological processes.

Upon analysis of the *kin-19* RNAi-dependent metabolic response, treated animals lacked the metabolic plasticity that was observed in wild-type or *nhr-49(nr2041)* animals. Specifically, while fasting typically reduces body size, it had no effect on *kin-19* RNAi-treated animals on Day 3 of adulthood, with even a slight increase in body size by Day 7 (Supplementary Fig. 6e, f). Despite starting with significantly lower lipid reserves, *kin-19* RNAi treated animals displayed no additional decrease in lipid availability after 24 h of fasting (Fig. 4a, c). This was in sharp contrast to the pronounced lipid depletion observed in both wild-type and *nhr-49(nr2041)* animals (Fig. 1d, h). Moreover, these worms lack the ability to fully restore lipid droplet levels after refeeding, recovering only 58.27% of their pre-fasted levels (Fig. 4d). Lipidomic profiling exasperated this lack of metabolic plasticity, revealing that several long-chain neutral fatty acids, namely palmitic, oleic, and γ-linolenic acids exhibited defective transitions during fasting in *kin-19* RNAi treated animals (Fig. 4e–g). Collectively, this evidence indicates that KIN-19 is essential for metabolic flexibility, with its absence mimics a permanent fasting state that severely impairs the animal's ability to adapt to nutrient changes.

To understand how these changes in lipid dynamics affect cellular energy homeostasis, we directly measured mitochondrial respiration by quantifying the oxygen consumption rates (OCR) in live animals under fasting and refeeding. Worms treated with *kin-19* RNAi consumed 50.4% less oxygen than control animals (average 61.3 pmol O$_2$/min vs. 121.7 pmol O$_2$/min, respectively), confirming a significant impairment in their bioenergetic capacity (Fig. 4h). To further investigate, we assessed mitochondrial morphology using an endogenously tagged COX-4::GFP[83] strain and a transgenic muscle-specific *myo-3*p::GFP reporter[84]. To validate our findings, we used *atp-3* RNAi, a positive control, which is known to cause mitochondrial fragmentation due to ATP synthase dysfunction[85,86]. In both intestinal and muscle tissues, we observed that *kin-19* RNAi-treated animals display a similar fragmented mitochondrial network, mimicking the phenotype observed in the *atp-3* control animals (Supplementary Fig. 6g, h). Lastly, we measured ATP levels and found that unlike wild-type and *nhr-49(nr2041)* animals, *kin-19* RNAi treated worms failed to restore ATP levels upon refeeding (Supplementary Figs. 6i, 1h, and 2f). This defective recovery of both cellular energetics and intracellular lipid availability highlights a critical role for KIN-19 in the restoration of energy homeostasis after fasting.

Given that KIN-19 reduction modulates NHR-49-dependent gene expression, we sought to determine whether this response was acting through NHR-49 to impact lipid metabolism. To this end, we utilized *nhr-49(nr2041)* mutants to investigate whether the loss of NHR-49 function would abrogate physiological and transcriptional changes observed with *kin-19* RNAi. Loss of lipid deposition caused by *kin-19*

RNAi was not observed in the *nhr-49(nr2041)* mutant animals as monitored by DHS-3::GFP fluorescence (Fig. 4i). Furthermore, the altered expression of fasting-induced genes by *kin-19* RNAi was also absent in the *nhr-49(nr2041)* mutants (Fig. 4j and Supplementary Fig. 6j). This suggests that the chronic fasting state induced by *kin-19* RNAi under *ad libitum* fed conditions is, at least in part, dependent on NHR-49 activity. To further examine this, we introduced the transgenic ATP sensor into *nhr-49(nr2041)* mutant animals. Unlike wild-type animals treated with *kin-19* RNAi (Supplementary Fig. 6i), these mutants efficiently restored their ATP levels upon refeeding (Supplementary Fig. 6k). These findings underscore a critical role for KIN-19 in facilitating the animal's recovery from fasting-induced β-oxidation. When KIN-19 is disrupted, it impairs the restoration of energy, ultimately compromising the metabolic plasticity needed for a successful transition during nutrient replenishment.

While impairing the activation of lipid catabolism was dispensable for fasting-induced longevity, the importance of silencing this catabolic response remained unclear. To test the necessity for NHR-49 attenuation in this process, we used *kin-19* RNAi to prevent its phosphorylation and subsequent inactivation. This manipulation dramatically reduced the benefits of fasting, leading to an average lifespan increase of only 6% and negating the positive effects on longevity (Fig. 4k). This result demonstrates that the ability to properly silence lipid catabolism via KIN-19 is essential for animals to achieve the full, age-related benefits of fasting.

## Silencing of NHR-49 by KIN-19 mediated protein turnover

Although KIN-19-dependent phosphorylation is required to silence NHR-49 activity, the downstream effects of this modification remain unclear. We found that phosphorylation of the S114 priming residue altered the nucleocytoplasmic distribution of NHR-49 and was necessary for subsequent phosphorylation of adjacent residues by recombinant KIN-19 (Figs. 2j and 3c). To examine how the loss of KIN-19 impacts NHR-49 protein dynamics, we immunoprecipitated NHR-49::GFP from transgenic animals and observed a threefold increase in the phosphorylated S114 peptide upon *kin-19* RNAi (Fig. 5a), indicating that reduced phosphorylation at S117 and S120 leads to accumulation of the S114 mark. Given that incomplete phosphorylation of S117 and S120 could stabilize NHR-49, we next quantified total protein levels. Casein kinases from the CK1 family are known to promote proteasomal degradation of substrates involved in circadian rhythm, cancer immunotherapy, and hepatic lipogenesis[87–91]. Consistent with this, *kin-19* RNAi significantly increased steady-state levels of NHR-49::YFP (Fig. 5b and Supplementary Fig. 7a). Despite the fact that peptides phosphorylated at S117 and S120 were not detected, these transient modifications may facilitate NHR-49 inactivation through nuclear export or turnover. Since mutation of the S114 priming site altered the subcellular distribution of NHR-49 (Fig. 2j), we next asked whether additional cofactors contribute to its shuttling or turnover.

To identify factors regulating NHR-49 turnover, we utilized our previously performed TurboID proximity labeling data set that tagged endogenous NHR-49 at either the N- or C-terminus[48]. Because this

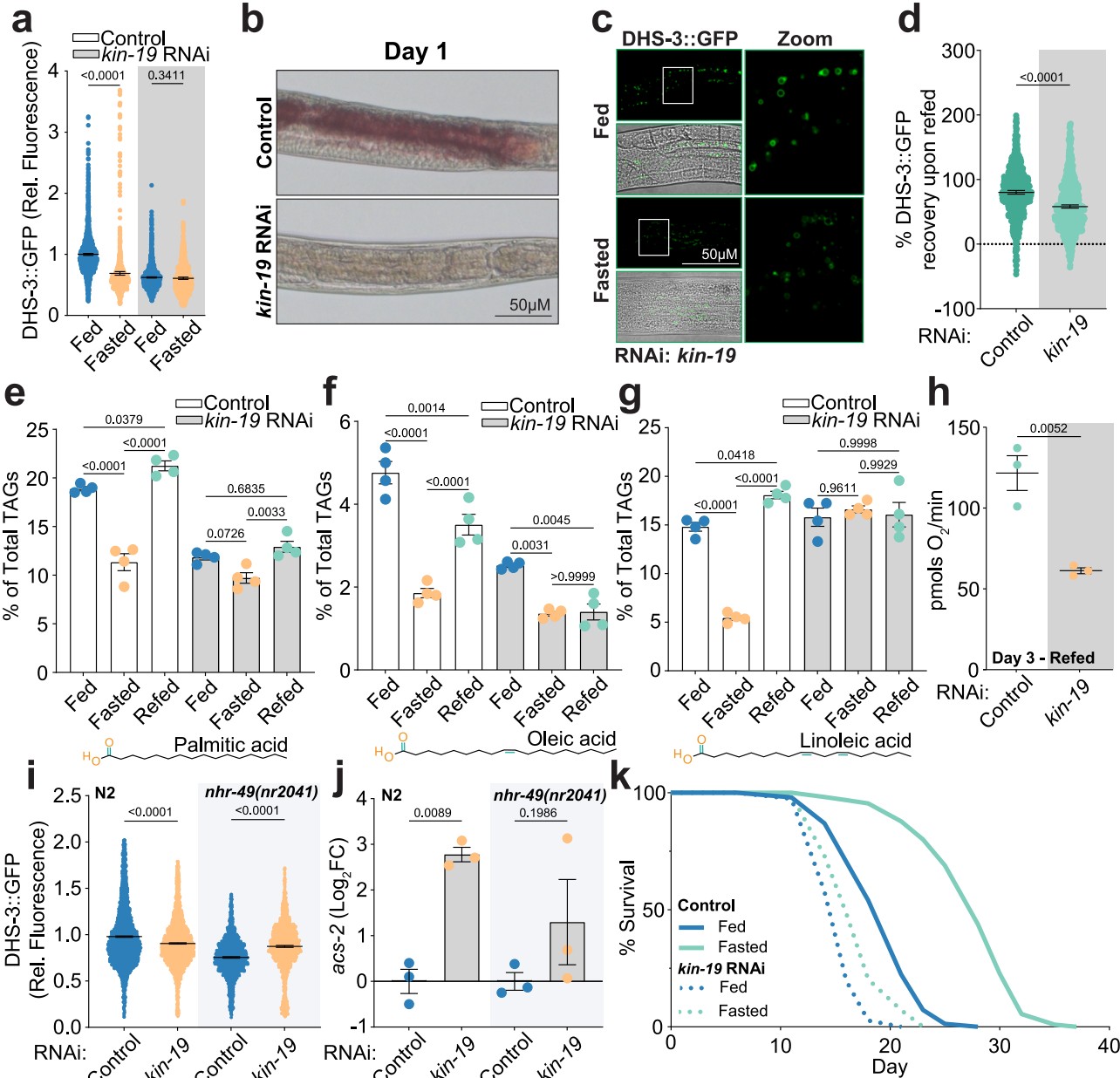

**Fig. 4 | Silencing lipid catabolism is required for fasting induced longevity.**
**a** Scatter plots from large-particle flow cytometry of transgenic worms expressing DHS-3::GFP show relative fluorescence per individual animal. Plots compare Day 1 adult animals on empty vector control or kin-19 RNAi under fed conditions or after 24 h of fasting. Mean ± 95% CI. $n$ = 1855, 953, 1774, and 797 from left to right, ordinary one-way ANOVA with Šídák's multiple comparisons test used for statistics. **b** Micrographs of stained neutral fatty acids by Oil-Red-O staining in wild-type, Day 1 adult worms on empty vector control or kin-19 RNAi. $n$ = 22 for empty vector control, $n$ = 19 for kin-19 RNAi. **c** Fluorescence micrographs of C. elegans intestinal cells ectopically expressing the lipid droplet localized dehydrogenase, DHS-3::GFP. Micrographs compare kin-19 RNAi treated worms under fed or fasting conditions (24 h). Scale bar = 50 μm. **d** Recovery of DHS-3::GFP fluorescence upon refeeding. Graph compares transgenic worms treated with empty vector control or kin-19 RNAi. Mean ± 95% CI. $n$ = 736 and 927 from left to right, unpaired $t$-test (two-tailed) was used for statistics. **e**–**g** Lipidomic analysis compares the relative abundance of different long chain neutral fatty acids as a percentage of the total TAG content from worms on empty vector control and kin-19 RNAi

under fasting and refeeding conditions. **e** palmitic acid, **f** oleic acid, and **g** γ-linolenic acid. Mean ± SEM. $n$ = 4. Šídák's multiple comparisons test used for statistics. **h** Oxygen consumption rate of Day 3 N2 worms on control or kin-19 RNAi after a 24-h fast, followed by a 24-h refeed. Mean ± SEM. $n$ = 10 per trial over three independent repeats, unpaired $t$-test (two-tailed) used for statistics.
**i** Scatter plots from large-particle flow cytometry of transgenic worms expressing DHS-3::GFP show relative fluorescence per individual animal. Plots compare Day 1 adult wild-type or nhr-49(nr2041) mutants on an empty vector control or kin-19 RNAi. Mean ± 95% CI. $n$ = 6748, 6248, 2452, and 2446 from left to right, ordinary one-way ANOVA with Šídák's multiple comparisons test used for statistics.
**j** Relative abundance of acs-2 transcription determined by RT-qPCR. Analysis compares Day 1 adult wild-type or nhr-49(nr2041) mutant worms on empty vector control or kin-19 RNAi. $n$ = 3, ordinary one-way ANOVA with Šídák's multiple comparisons test used for statistics. **k** Lifespan analysis of wild-type (N2) worms on empty vector or kin-19 RNAi under fed or fasted conditions (24 h of dietary deprivation at Day 1 of adulthood). Based on 3 biological replicates, log-rank (Mantel–Cox) test used for statistics (see Supplementary Table 1).

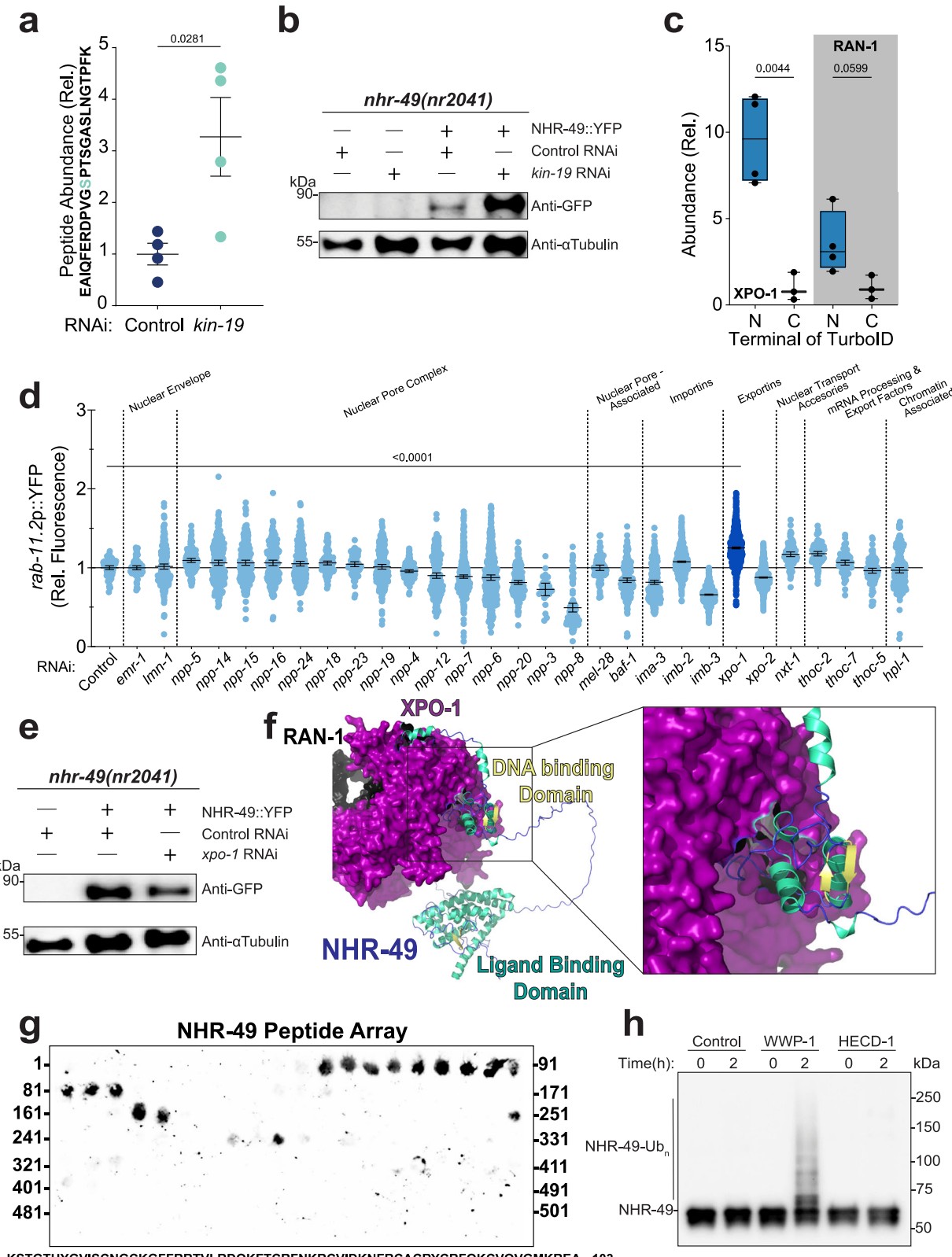

phosphorylation cluster resides within the first 120 amino acids, we anticipated that N-terminal tagging would preferentially biotinylate key regulators coordinating with the hinge region. After filtering for common proteins between the N- and C-terminal data sets, 149 candidates were enriched at least twofold ($p < 0.05$) in the N-terminal dataset compared to the C-terminal one. Among these, XPO-1, highly conserved exportin, ranked 14th with a 9.59-fold enrichment

($p = 0.0044$), and its binding partner RAN-1 showed a 3.56-fold enrichment ($p = 0.0599$) (Fig. 5c and Supplementary Fig. 7b). We next performed RNAi knockdown of nucleocytoplasmic trafficking candidates identified both here and in the fasted NHR-49::GFP immunoprecipitation datasets. Depletion of XPO-1 significantly activated the *rab-11.2*p::YFP fasting reporter (Fig. 5d) and increased fluorescence of the *acs-2*p::GFP reporter by 68% (Supplementary Fig. 7c). Unlike *kin-19*

**Fig. 5 | Silencing NHR-49 via primed phosphorylation. a** Relative abundance of phosphorylated S114 NHR-49 peptides detected by mass spectrometry in empty vector control and kin-19 RNAi conditions. Mean ± SEM. *n* = 4 biological replicates. A paired *t*-test was used for statistical analysis. **b** Western blot of NHR-49::YFP in nhr-49(nr2041) background under empty vector control and kin-19 RNAi conditions at Day 1 of adulthood. **c** TurboID proximity labeling of N- or C-terminally tagged NHR-49 to XPO-1 and RAN-1. Normalized to the C-terminally tagged strain. *n* = 3 (C-terminal), 4 (N-terminal) biological replicates. Box boundaries = 25% (lower), 75% (upper) percentile. Whiskers = Min/Max. Centre line = Median. Ordinary one-way ANOVA with Šídák's multiple comparisons test used for statistics. **d** Scatter plots from large-particle flow cytometry of transgenic worms expressing rab-11.2p::YFP show relative fluorescence per individual animal. Plots compare Day 1 adults under empty vector control conditions to the respective nucleocytoplasmic RNAi conditions. Mean ± 95% CI. *n* = 80, 70, 245, 118, 254, 215, 188, 235, 125, 75, 225, 141, 233, 450, 345, 224, 24, 92, 81, 222, 278, 1259, 909, 1243, 1311, 94, 94, 74, 104 and 203 from left to right, ordinary one-way ANOVA with Dunnett's multiple

comparisons test used for statistics (see Supplementary Table 4). **e** Western blot of NHR-49::YFP in nhr-49(nr2041) background under empty vector control and xpo-1 RNAi conditions at Day 1 of adulthood. **f** Predicted secondary structure and intermolecular interface of XPO-1, RAN-1 and NHR-49, modeled using AlphaFold3, an advanced AI-driven deep learning algorithm for protein structure and interaction prediction. Surface density for XPO-1 (purple) and RAN-1 (black) and a cartoon rendering of NHR-49 (ligand binding domain in blue and DNA binding domain in yellow). **g** Immunoblot analysis of XPO1 binding to an NHR-49 polypeptide array. A total of 123 overlapping 15mer peptides (spanning the full length of NHR-49 isoform c and overlapping by 11 amino acids) were immobilized on cellulose and incubated with recombinant XPO1 and the RAN-1 ortholog, GSP1, prior to XPO1 immunoblotting. See Supplementary Fig. 7f for schematic. **h** Western blot of T7 epitope tagged N-terminal NHR-49 (1-155). Utilizing recombinant enzymes and substrates, either HECD-1 or WWP-1 were incubated with NHR-49 in the presence of ubiquitin and the E1 and E2 ubiquitin ligases.

knockdown, however, loss of *xpo-1* did not cause NHR-49 accumulation (Fig. 5e and Supplementary Fig. 7d), suggesting an alternative role for this interaction, potentially through nuclear-based degradation or functions independent of canonical nuclear export.

Sequence analysis of NHR-49 did not identify a canonical nuclear export signal (NES) to explain its interaction with XPO-1; however, XPO-1 has recently been reported to mediate transcription factor–dependent chromatin docking at the nuclear pore complex independently of a canonical NES[92]. Consistent with this, AI-based structural prediction[74] indicated a direct interaction between NHR-49 and XPO-1 that required its small GTPase cofactor RAN-1 (ipTM = 0.66 versus 0.54 without RAN-1) (Fig. 5f and Supplementary Fig. 7e). The predicted binding interface was located at the N-terminus of NHR-49, consistent with our proximity labeling results. To further validate this interaction, we employed an immobilized peptide array spanning the length of NHR-49 and probed with recombinant human XPO1 together with the yeast RAN-1 ortholog, GSP1. XPO1 specifically bound to NHR-49 peptides corresponding to the AI-predicted N-terminal interfaces (Fig. 5g and Supplementary Fig. 7f). Together, these results support an atypical interaction between XPO-1 and NHR-49 that may facilitate translocation of NHR-49–bound DNA to the nuclear pore.

Building on our observation that NHR-49 localizes in proximity to nuclear pore components[48], we revisited our nucleocytoplasmic trafficking RNAi screen. Knockdown of several nuclear pore proteins, most notably the inner ring component NPP-8, blocked activation of the *rab-11.2* fasting reporter (Fig. 5d). We previously reported that NPP-8 was 2.42-fold enriched at the N-terminus of NHR-49, suggesting a potential cooperative interaction with XPO-1[48]. These findings support a model in which XPO-1 binding to the N-terminus of NHR-49 serves as a critical upstream event that promotes NHR-49's association with the nuclear pore complex, potentially coordinating its nuclear engagement under conditions of stress. The observation that XPO-1 knockdown does not cause NHR-49 accumulation, unlike *kin-19* knockdown, further suggests that XPO-1 regulates NHR-49 through a distinct, non-canonical mechanism. Whereas KIN-19 directly influences protein stability, XPO-1 appears to act earlier in the trafficking pathway.

While KIN-19 appears to influence the steady-state levels of NHR-49, we sought to investigate the mechanisms underlying NHR-49 degradation. A common pathway for nuclear hormone receptor degradation is the ubiquitin-proteasome system, wherein specific E3 ubiquitin ligases mediate receptor ubiquitination and degradation[93,94]. Consistent with this, phosphorylation of other nuclear receptors, such as the androgen and progesterone receptors, has been shown to enhance their recognition by E3 ligases[95]. Based on proximity labeling and co-immunoprecipitation, seven E3 ubiquitin ligases were identified as potential interactors of NHR-49, with the top candidates being the HECT domain-containing E3 ligase, HECD-1, with 30 PSMs detected via proximity labeling, and the WW domain-containing E3 ligase, WWP-

1, with 110 PSMs identified through co-immunoprecipitation (Supplementary Fig. 7g). When testing their ability to conjugate ubiquitin to NHR-49 in vitro, WWP-1 successfully transferred ubiquitin, resulting in detectable polyubiquitinated species, whereas HECD-1 failed to do so (Fig. 5h). Using both recombinant proteins, we confirmed that WWP-1 directly binds to NHR-49 with a 1-to-1 stoichiometry in vitro (Supplementary Fig. 7h). While WWP-1 has the capacity to ubiquitinate NHR-49 and has previously been implicated as an aging regulator during dietary restriction[96,97], *wwp-1* RNAi did not reproducibly stabilize steady-state NHR-49 levels within the worm as determined by western blot (Supplementary Fig. 7i, j). While this discrepancy could be due to systemic redundancy, the action of deubiquitinating enzymes, or a requirement for adapter proteins or specific post-translational modifications, WWP-1 remains a promising candidate capable of ubiquitinating and modulating the activity of NHR-49 during metabolic stress. and modulating its activity.

## Discussion

Our findings, in conjunction with previous research, support a model in which KIN-19 phosphorylation of NHR-49 facilitates its turnover. However, the precise mechanism by which this turnover occurs remains an interesting avenue for future research. The mammalian ortholog, casein kinase 1 alpha (CSNK1A1), is known to prime substrates for E3 ligase recognition by phosphorylating short amino acid sequences called phospho-degrons, which leads to ubiquitination and rapid proteasomal degradation[70,76,77,87,89]. This mechanism has been observed with other nuclear receptors, where phosphorylation of the androgen and progesterone receptors enhances their recognition by E3 ligases[67]. Given this, we partially investigated whether KIN-19 employs a similar mechanism to regulate the stability of NHR-49.

While several E3 ligases were found in complex with NHR-49::GFP in vivo under fasted conditions[42], we chose the top two interactors to screen: the WW domain-containing E3 ligase, WWP-1, and HECT domain E3 ligase, HECD-1. In our in vitro assay, WWP-1, but not HECD-1, efficiently transferred ubiquitin molecules onto recombinant NHR-49, resulting in detectable polyubiquitinated species. This direct, stoichiometric interaction between WWP-1 and NHR-49 indicates a direct physical link between the two proteins. This discovery is particularly notable as previous studies have already implicated WWP-1 as an important mediator of lifespan extension by dietary restriction[96,97]. Our results extend this role, positioning WWP-1 as a potential central regulator of lipid metabolism and aging through its interaction with NHR-49.

This study reveals that the suppression of lipid catabolism is a key determinant of longevity in fasting paradigms, while the ability to activate lipid catabolism appears surprisingly dispensable for these pro-longevity effects. As one of the most evolutionarily conserved lifespan-extending strategies, dietary restriction has emerged as a promising intervention for promoting healthy aging. Rather than

continuous caloric limitation, fasting has proven to be a more effective means of harnessing the benefits of dietary restriction. The metabolic fluctuations induced by fasting and subsequent refeeding further enhance longevity and reduce disease risk[3,5,6,9,11].

While much research has focused on the activation of stress response pathways during metabolic challenge, the mechanisms by which these pathways are attenuated once the stress subsides remain largely unexplored. To address this gap, we disrupted lipid catabolism during fasting by impairing key factors involved in triglyceride hydrolysis, coenzyme A conjugation to mitochondrial acyl chains, and the transcriptional activation of β-oxidation. Despite these disruptions, fasting consistently extends lifespan across all conditions. Instead, our findings suggest that the suppression of lipid catabolism upon nutrient replenishment plays a more pivotal role in determining longevity during fasting. Further investigating these poorly defined attenuation mechanisms and their physiological contributions across different fasting paradigms is therefore essential for a more comprehensive understanding of age regulation.

Our findings highlight a reciprocal relationship between the activation and suppression of lipid catabolism in lifespan determination. Under *ad libitum* fed conditions, the ability to activate β-oxidation plays a predominant role in longevity, as evidenced by a 41% reduction in lifespan in *nhr-49(nr2041)* mutants. However, during fasting, NHR-49 and other key regulators of lipolysis and β-oxidation appear dispensable for lifespan extension. Conversely, KIN-19 is required for the suppression of lipid catabolism. While its reduced expression leads to a modest 14.2% decrease in lifespan under *ad libitum* fed conditions, it is essential for the longevity benefits of fasting. These findings strongly suggest that alternate metabolic pathways support survival and viability when lipid catabolism is impaired. For example, worms store glycogen in their intestines, which could serve as an alternative energy source in the absence of lipid breakdown[98]. Additionally, enhanced amino acid catabolism, as observed in mouse models of dietary deprivation[99], may contribute to sustaining metabolic demands during fasting. Further research is needed to elucidate these compensatory mechanisms.

The suppression of catabolism during fasting is mediated by the nuclear hormone receptor NHR-49, a key regulator of β-oxidation in *C. elegans*. Traditionally, nuclear hormone receptors are ligand-gated transcription factors that orchestrate complex transcriptional programs governing hormonal regulation, sexual differentiation, and metabolism[100–102]. However, our findings reveal a ligand-independent mechanism in which the *C. elegans* casein kinase ortholog, KIN-19, inactivates NHR-49 through primed phosphorylation. This regulatory cascade is initiated by phosphate priming at S114 within the hinge region of NHR-49, which facilitates subsequent phosphorylation at S117 and S120 by KIN-19. However, the mechanism underlying KIN-19 activation during fasting remains an outstanding question. Since its subcellular localization and multimerization appear unaffected during fasting, KIN-19 activity is most likely regulated through cofactor interaction or posttranslational modification.

Our mutational analysis indicates that the S114A mutant accumulates in the nucleus, suggesting that S114 phosphorylation is required for nuclear exit and subsequent phosphorylation by KIN-19. However, the identity of the kinase responsible for the priming phosphorylation at S114 and the precise mechanism by which this modification triggers protein turnover remains unclear. While electrostatic repulsion from phosphate group addition is known to disrupt protein-nucleotide interactions[103], our data indicates that S114 phosphorylation does not alter binding by the nuclear export receptor, XPO-1. Therefore, we propose that this modification more likely promotes the release of NHR-49 from chromosomal DNA via electrostatic repulsion. This proposed mechanism would ensure that newly synthesized or vesicle-released NHR-49 remains protected from premature degradation in the cytosol and must first enter the nucleus to acquire competence for turnover in the cytosol.

Lastly, phosphorylation-dependent regulation is well-established among nuclear hormone receptors. For instance, phosphorylation of PPARγ or the glucocorticoid receptor (GR) by Casein Kinase 2 or c-Jun N-terminal kinase (JNK) promotes nuclear export of the respective receptor[62,104,105]. However, the regulatory paradigm of NHR-49 diverges from the canonical ligand-driven activation model that defines the nuclear receptor superfamily. Instead, this form of ligand-independent regulation more closely resembles mechanisms governing transcription factors outside this family, highlighting the functional versatility of nuclear receptors as regulatory hubs.

Our findings suggest that similar phosphorylation-driven mechanisms may regulate other orphan nuclear receptors, many of which lack well-defined endogenous ligands. This raises the possibility that the traditional assumption of ligand binding as a universal activation mechanism for nuclear receptors may require reassessment. A broader investigation into alternative regulatory modalities, such as ligand-dependent cytoplasmic sequestration or post-translational modifications, could provide new insights into nuclear receptor biology and help address the challenges of deorphanization.

## Methods

### Strain maintenance and growth conditions
*Caenorhabditis elegans* strains were maintained at 15 °C on standard nematode growth medium (NGM) plates seeded with *Escherichia coli* OP50. For experimental synchronization and amplification, worms were transferred to fresh plates and grown at 20 °C. All experiments were conducted on NGM plates containing 100 µg/mL carbenicillin and 1 mM Isopropyl β-D-1-thiogalactopyranoside (IPTG) and were seeded with *E. coli* HT115 (DE3), as described in the RNAi Administration section. Worms were synchronized by either a 6-h egg-lay or hypochlorite treatment. For hypochlorite treatment, gravid Day 1 adults from confluent high growth medium (HGM) or NGM plates were washed into a conical tube with M9 buffer. After pelleting at $1000 \times g$ for 30 s, the worms were incubated in a solution of 20% bleach (sodium hypochlorite), 5% NaOH, and 75% $H_2O$ with gentle shaking until the adults dissolved, leaving only eggs behind. The eggs were then pelleted, washed three times with M9 buffer, and counted to ensure consistency across experiments.

### C. elegans strains
The following strains were obtained from the Caenorhabditis Genetics Center (CGC): N2: *C. elegans* var. Bristol (wild-type), AGP33a: *nhr-49(nr2041);* glmEx8 [*nhr-49*p::NHR-49::GFP*; myo-2*p::mCherry], LIU1: ldrIs1 [*dhs-3*p::DHS-3::GFP; unc-76(+)], STE68: *nhr-49(nr2041)*, JJ2586: cox-4(zu476[COX-4::eGFP::3xFLAG]) I, WBM170: wbmEx57 [*acs-2*p::GFP + rol-6(su1006)], and SJ4103: zcIs14 [myo-3::GFP(mit)], and GA2001: wuIs305 [*myo-3*p::Queen-2m]. The following strains were previously generated in our laboratory: PMD166: utsIs4 [*nhr-49*p::NHR-49::GFP; myo-3p::mCherry]; *nhr-49(nr2041)*, PMD142: ldrIs1[*dhs-3*p::DHS-3::GFP; unc-76(+)]; *nhr-49(nr2041)*, PMD319: *nhr-49(syb10204*[nhr-49::3xHA::TurboID]), PMD320: *nhr-49c(syb10203*[3xHA::TurboID::nhr-49c]), and PMD124: utsIs3[*rab-11.2*p::YFP::unc-54 3'UTR]. The following strains were generated for this study: PMD192: utsEx24 [*pept-1*p::YFP::unc-54 3' UTR], PMD261: utsIs10 [*pept-1*p::NHR-49::YFP::unc-54 3'UTR], *nhr-49(nr2041)*, PMD262: utsIs11 [*pept-1*p::NHR-49(Δ295−422)::YFP::unc-54 3'UTR]; *nhr-49(nr2041)*, PMD263: utsIs15 [*pept-1*p::NHR-49(S114A)::YFP::unc-54 3'UTR]; *nhr-49(nr2041)*, PMD314: utIs13 [pept-1p::NHR-49(s114d)::YFP::unc54 3'UTR]; *nhr-49(nr2041)*, PD17: *kin-19(syb8063[kin-19::mNeonGreen2])*, and PMD311: wuIs305 [*myo-3p::Queen-2m*]; *nhr-49(nr2041)*.

### Molecular cloning
All plasmids were constructed using restriction enzyme digestion and ligation following standard protocols from New England Biolabs (NEB).

The parent vector, pNB13 (a generous gift from the laboratory of Andrew Dillin), which contains a Yellow Fluorescent Protein (YFP) tag and a 3′ untranslated region (UTR), served as the backbone for this study. The *pept-1* promoter (1407 bp upstream of the transcription start site) was amplified from isolated *C. elegans* genomic DNA and cloned upstream of the YFP tag to generate the pLMT1 vector. Subsequently, cDNA encoding the NHR-49 isoform C, with an N-terminal 6x-histidine (His) tag, was obtained from GENEWIZ and inserted downstream of the *pept-1* promoter and upstream of the YFP tag to create the pLMT2 expression vector. Polymerase Chain Reaction (PCR) was performed using the NEB Q5 High-Fidelity 2x Master Mix (Cat. #M0492S). PCR products were confirmed via agarose gel electrophoresis and purified using the NEB Monarch PCR & DNA Cleanup Kit (Cat. #T1130L). Inserts and vectors were ligated at a 7:1 molar ratio using NEB T4 DNA Ligase (Cat. #M0202S). Mutations and truncations were introduced into pLMT2 using the QuikChange Lightning Site-Directed Mutagenesis Kit (Cat. #NC9620881, Agilent). All final plasmid constructs were transformed into NEB 5-alpha competent *E. coli* (Cat. #C2987U) and verified by Sanger sequencing (Eurofins Genomics LLC, Louisville, KY). Oligonucleotides and primers used can be found in Supplementary Data 2.

### Transgenic strain generation

Transgenic strains were generated by microinjecting plasmids into the gonads of Day 1 adult N2 wild-type worms at a concentration of 100 ng/μL. For transgenic constructs that did not yield stable extrachromosomal arrays after three independent trials, the plasmid was linearized by restriction enzyme digestion and amplified from promoter to 3′UTR. The full-length transgene was then amplified by PCR, purified using the NEB Monarch PCR & DNA Cleanup Kit (Cat. #T1130L) and injected at a concentration of 100 ng/μL. Once a stable extrachromosomal array was established (indicated by consistent fluorescent expression across generations), strains were selected for integration into the genome. Integration was performed by exposing L4-stage transgenic larvae to UV irradiation at 0.4 J cm⁻² using a UV cross-linker. The worms were then allowed to recover at 20 °C on OP50-seeded NGM plates. Progeny exhibiting stable transgene expression in every animal were isolated and frozen down for long term preservation. Additionally, integrated transgenic strains were outcrossed to the N2 wild-type strain a minimum of three times to remove background mutations and ensure genetic consistency.

### Single worm genotyping

Individual worms were transferred into a PCR tube containing 10 μL of lysis buffer (10 mM Tris, pH 8.2; 50 mM $MgCl_2$; 0.45% Tween-20; 60 μg/mL Proteinase K). The samples were then subjected to a three-step thermal lysis protocol: frozen at −80 °C for 15 min, followed by incubation at 60 °C for 60 min, and finally, proteinase K inactivation at 95 °C for 15 min. A 5 μL aliquot of the resulting worm lysate was used as the DNA template for PCR to confirm genetic background. PCR products were subsequently analyzed via electrophoresis on a 1% agarose gel.

### RNAi administration

For all experiments, *C. elegans* strains were cultured on *E. coli* HT115 (DE3) expressing double-stranded RNA (dsRNA) from the Ahringer or Vidal RNAi libraries. The L4440 empty vector (EV) RNAi served as a control. To prepare the RNAi-expressing bacteria, single colonies were used to inoculate small cultures (1–5 mL) before being scaled up to larger cultures in terrific broth (TB) supplemented with 100 μg/mL carbenicillin. Cultures were shaken at 37 °C for 15 h. Following this, 1 mM IPTG was added to induce dsRNA expression, and the cultures were shaken for an additional 4 h at 37 °C. The bacterial cells were then harvested by centrifugation at 4000 × g for 12 min. The bacterial pellet

was resuspended to 1/5th of the original culture volume and seeded onto NGM plates containing 1 mM IPTG and 100 μg/mL carbenicillin.

### Fasting-refeeding assays

Age-synchronized worms were cultured on either experimental or control RNAi plates until Day 1 of adulthood. On Day 1 of adulthood, worms were harvested by washing plates with M9 buffer and subsequently washed twice with fresh M9 buffer via centrifugation (1000 × g, 30 s). Worms were next transferred to room temperature NGM RNAi plates (100 μg/mL carbenicillin, 1 mM IPTG) that were either unseeded (fasting conditions) or seeded with freshly generated RNAi bacteria (*ad libitum* control) for 24 h.

On Day 2, following the 24-h feeding or fasting period, worms were collected by washing with M9 buffer. To separate adults from larvae and eggs, the samples were allowed to settle by gravity for 5 min in a conical tube. The supernatant, containing the younger stages, was aspirated. This gravity-based separation was repeated once. For fasted worms, which often failed to settle effectively, an additional centrifugation step (1000 × g, 30 s) was performed during the washing step. All worms were then transferred to RNAi plates containing freshly generated food (control or experimental RNAi) for an additional 24 h before being collected for downstream assays on Day 3 of adulthood.

### Large particle flow cytometry

Large-particle flow cytometry was performed using a COPAS FP-250 flow cytometer (Union Biometrica, Holliston, MA) equipped with an automated sample introduction system, the LP Sampler, for 96-well plate analysis[106]. M9 buffer was used as the sample solution and COPAS GP sheath reagent (PN: 200-5070-100, Union Biometrica) as the sheath solution. Consistent PMT laser power was maintained for each fluorophore. Flow data was acquired using FlowPilot software (Union Biometrica), and subsequent data analysis and normalization are detailed in the Statistical Analysis methods section.

The experimental design for fasting and refeeding is illustrated in Supplementary Fig. 1a. Age-synchronized worms were distributed onto three replicate plates for each of three conditions: (1) continuously fed, (2) fasted for 24 h followed by refeeding, and (3) continuously fasted.

On Day 1, worms from one plate per condition were collected, washed, and analyzed via flow cytometry to establish baseline readings. The remaining worms were transferred to fresh plates according to their respective conditions: (1) fresh food plates, (2) empty plates for fasting, or (3) empty plates for continuous fasting. On Day 2, worms from a second plate per condition were collected and analyzed. The remaining worms were transferred to their next treatment: (1) fresh food plates, (2) food plates for refeeding, or (3) empty plates to continue fasting. On Day 3, the final set of worms was collected and analyzed. Flow cytometry gating strategy is detailed in Supplementary Fig. 1e. For comparative analyzes, we focused on three key time points: Day 1 (fed), Day 2 (fasted), and Day 3 (refed). Signal changes were evaluated relative to the Day 1 baseline, as it represents the highest initial signal prior to fasting.

For ATP analysis, two separate plates of age-synchronized worms per condition were prepared. One plate was analyzed on a large particle flow cytometer (LPFC) with a 405 nm excitation laser, while the other was excited at 488 nm. Green fluorescent protein (GFP) fluorescence was measured for each excitation and normalized to the respective time-of-flight (TOF) values. The average GFP signal from the 488 nm-excited worms within a given condition was compared to the corresponding 405 nm-excited worms to generate a 405/488 fluorescence ratio. This ratio was then normalized to the fed control condition. As a negative control, 5 mM sodium azide was added to a separate control well 5 min before LPFC analysis to irreversibly inhibit cytochrome C oxidase and establish a baseline for low ATP levels.

## Lifespan analysis

Age-synchronized worm populations were obtained by hypochlorite treatment to isolate eggs. The eggs were transferred to RNAi plates seeded with 150 μL of either experimental RNAi bacteria or empty vector (EV) control. On Day 1 of adulthood (with timing adjusted for developmental variations caused by RNAi), 5-fluoro-2'-deoxyuridine (FUdR) was added to the plates at a final concentration of 100 μg/mL to inhibit progeny development. Lifespan assays were performed with 10–20 worms per plate, using 10 plates per condition for a total of at least 100 worms per condition, unless noted elsewhere. Worms were inspected every other day, and mortality was scored based on a lack of response to a gentle touch with a platinum wire. Animals were censored from the analysis if they burrowed, ruptured, experienced internal hatching of progeny, desiccated, or disappeared. Plates with fungal or bacterial contamination were excluded entirely. For fasting lifespan analysis, Day 1 adult worms were collected, washed three times with fresh M9 buffer to remove residual *E. coli*, and transferred to unseeded NGM plates containing 100 μg/mL carbenicillin. These worms were cultured at 20 °C for a 24-h fasting period. Afterward, the fasted worms were washed and transferred to NGM plates seeded with their respective RNAi-expressing *E. coli*. FUdR was included on all plates, including the fasting plates and the recovery plates to ensure consistency across all conditions.

## RNA extraction

Worm pellets (~100 μL) were collected in TRIzol reagent (Cat. #15596026, Thermo Fisher Scientific), flash-frozen in liquid nitrogen, and stored at −80 °C. Total RNA was isolated using a freeze-thaw-vortex method, which involved three cycles of alternating liquid nitrogen freezing, room temperature thawing, and 30-s vortexing. RNA extraction was performed via a standard chloroform/phenol and isopropanol precipitation method. The resulting RNA pellets were washed twice with 75% ethanol, with gentle inversion and centrifugation to ensure complete removal of impurities. After aspirating the final ethanol wash, the pellets were air-dried for up to 2 h before being resuspended in 20 μL of molecular biology-grade water. RNA samples were stored at −80 °C until analysis. RNA concentration and purity were assessed using a DeNovix spectrophotometer (DS-11 FX+, DeNovix, Wilmington, DE), with quality control metrics recorded at the 260/280 nm and 260/230 nm absorbance ratios.

## Illumina RNA sequencing and data processing

RNA sequencing was performed to investigate gene expression changes under fasting-refeeding conditions and following *kin-19* RNAi treatment. Total RNA was extracted as described above and submitted to Novogene (Sacramento, CA) for quality control, mRNA enrichment, library preparation, and paired-end 150 bp Illumina sequencing. mRNA enrichment was performed using oligo(dT) beads, followed by fragmentation and first-strand cDNA synthesis using random hexamer primers. Second-strand cDNA was synthesized using a buffer containing dNTPs, RNase H, and *E. coli* DNA polymerase via nick-translation. Library preparation and PCR enrichment were carried out according to Novogene's standard protocols. The resulting cDNA libraries were quantified using a Qubit 2.0 fluorometer and fragment size distribution was assessed with an Agilent 2100 Bioanalyzer (Agilent Technologies, Santa Clara, CA). Post-sequencing data analysis was conducted using CLC Genomics Workbench (v. 9.0, CLC Bio, Aarhus, Denmark), with read counts expressed as counts per million (CPM) prior to normalization. For the fasting and refeeding experiment, we used a modified version of the flow cytometry protocol with quadruplicate samples for the following conditions: (A) Day 1 fed, (B) Day 2 fed, (C) Day 2 fasted, (D) Day 3 fed, and (E) Day 3 refed.

## Quantitative PCR (qPCR)

Worms were synchronized by hypochlorite treatment and cultured on either experimental RNAi or control plates at 20 °C until they reached the L4 larval stage or a specified age. Worms were harvested by washing plates with M9 buffer and pelleted via centrifugation at $1000 \times g$ for 30 s. Total RNA was then extracted as described in the RNA Extraction methods section. For cDNA synthesis, 1 μg of total RNA was reverse-transcribed using the QuantiTect Reverse Transcription Kit (Cat. #205311, Qiagen), following the manufacturer's instructions. qPCR reactions were prepared in 20 μL volumes using iTaq™ Universal SYBR® Green Supermix (Cat. #1725124, Bio-Rad) and 50 ng of cDNA per well. Amplifications were performed on a CFX384 Real-Time PCR System (Bio-Rad, Hercules, CA). Each sample was analyzed in technical triplicates with a minimum of three biological replicates per condition. PCR fidelity was confirmed by melting curve analysis, and any technical replicates that failed this quality control were excluded. Relative transcript levels for target genes were calculated using the ΔΔCt method and normalized to the geometric mean of two reference genes, *tba-1* and *Y45F10D.4*. Primers used can be found in Supplementary Data 2.

## Worm tracking

The movement and morphology of worms were analyzed using a worm tracking system (WormLab Imaging System, v2024.01.01 64-bit, MBF Bioscience, Williston, VT). Videos were recorded at 150 frames per segment under consistent illumination and magnification for each experimental condition. Individual worms were identified and tracked across video frames using thresholding algorithms. Descriptive statistics, including worm length and locomotion speed, were calculated for each animal. Worm length was normalized to the dimensions of the video frame to account for variations in size. Locomotion speed was calculated over a 10-s window by measuring the absolute distance traveled by the first five worms in each condition. Both worm length and speed measurements were normalized to their respective fed control groups to facilitate comparative analyzes. Normalized data was then statistically analyzed to evaluate differences across experimental groups.

## Brood count

Worm populations (F1) were age-synchronized by placing 10 adult, egg-laying worms (P0) on plates for a 4-h period. Once the synchronized progeny reached the L4 larval stage, single worms were transferred to individual 60 mm NGM plates seeded with *E. coli* OP50. Upon reaching adulthood, worms were subsequently moved to freshly-seeded 60 mm NGM plates every 24 h. The exhausted plates were incubated at 20 °C, and brood counts were performed 24 h after the adult was transferred. The number of viable progeny and unhatched eggs was recorded.

## Quantification of cytoplasmic localization

To assess the cytoplasmic localization of fluorescence, we adapted protocols[42] to focus exclusively on intestinal cells. In some cases, transgenic worms possessed extrachromosomal arrays. Thus, each intestinal cell displaying a fluorescent signal above age-matched, non-transgenic N2 worms, was scored based on the presence of fluorescence within the cytosol. A score of 1 was assigned if fluorescence was visible within the cytosol, while a score of 0 was assigned if fluorescence was present in the nucleus but the cytosolic signal failed to exceed that of the age-matched, non-transgenic N2 worms. The cumulative score was normalized to the number of cells analyzed and multiplied by 100 to yield a percentage of cytoplasm-localized cells per worm. The final data are presented as the average percentage of cytoplasmic localization for each experimental condition.

## Lipidomics

Samples from both control and *kin-19* RNAi-treated conditions were collected across three metabolic states: (A) Day 1 fed, (B) Day 2 fasted, and (C) Day 3 refed. Worm pellets were collected independently at different times, flash frozen in liquid nitrogen and stored at −80 °C. All frozen biological replicates (*n* = 4) were flash thawed and processed for lipidomic analysis at the same time. To ensure accurate normalization, 20% of each pellet was first processed using a bicinchoninic acid (BCA) assay to determine protein content. Briefly, neutral and polar fatty acids were analyzed following a three-phase liquid extraction[107]. The extracted lipids underwent base hydrolysis with potassium hydroxide, followed by derivatization with triethylamine and pentafluorobenzyl bromide. Samples were then analyzed by gas chromatography-mass spectrometry (GC-MS) with electron capture negative ionization. All data processing was performed using MassHunter software (Agilent Technologies, Santa Clara, CA).

## Thin layer chromatography

A protocol was adapted for *C. elegans*[108]. Briefly, ~3000 age-synchronized D1 worms per replicate were collected at the given conditions and washed three times with M9 buffer followed by centrifugation at 1000 × *g* for 30 s at room temperature. Worm pellet weight was recorded and then resuspended in 400 μL milliQ-H$_2$O. Samples were homogenized with zirconia and ceramic beads at 6500 RPM for 15 s followed by a 20 s pause, twelve times at 4 °C. All following steps were performed at room temperature. Lysate was diluted in 1 mL chloroform:methanol (2:1 v/v) and vortexed for 5 min at max speed. Contents were transferred to a new 5 mL glass vial. Original lysate tubes were washed with 1 mL methanol and subsequently combined into the same 5 mL vial. A total of 2 mL of chloroform was added to lysates, followed by vortexing at 1500 RPM for 1 min, and centrifugation at 1000 × *g* for 5 min. The addition of chloroform should generate two separate phases. The lower phase was aspirated with a glass pipette and transferred to a new 5 mL glass vial. Upper phases were then washed with 3 mL chloroform:methanol:milliQ-H$_2$O (2:1:0.1 v/v/v) in the same manner, and lower phases were combined with the previous fraction. A total of 1 mL 1 M KCl was added to the samples, followed by vortexing and centrifugation as described, and the upper phase was subsequently discarded. This was repeated twice followed by lower phase aspiration to a new 4 mL glass vial. Lipid extracts were dried under argon gas and resuspended in chloroform at a concentration of 0.5 mg/μL based on unprocessed worm pellet weight. Serial diluted standards resuspended in chloroform (2, 4, 8, 12, and 20 μL of neutral lipid cocktail [0.25 μg/μL cholesteryl oleate, 0.2 μg/μL glyceryl trioleate, 0.15 μg/μL oleic acid, 0.35 μg/μL cholesterol]) along with 10 μL per sample were loaded onto a TLC Silica Gel 60 plate (Cat. #1057210001, MilliporeSigma). Neutral lipids were separated on TLC plates using hexane:diethyl ether:acetic acid (80:20:1 v/v/v). Plates were air dried for 10 min and then stained with 3% copper (II) acetate in 8% phosphoric acid by aerosol spray, followed by incubation for 30 min at 145 °C. Lipid bands were captured via digital scanning and intensity was quantify using Fiji (ImageJ). Sample band intensity was measured per sample and calculated as percent fold change relative to fed controls.

## Sample preparation for microscopy

For live imaging, collected worms were washed with M9 and paralyzed by adding 50 μL of 200 mM levamisole. Worms were then mounted onto a microscope slide (Cat. #12-544-2, Fisher Scientific), covered with a glass coverslip (Cat. #48366-227, VWR) and sealed with clear nail polish. For fixed imaging, worms were first washed three times with 1 mL of 0.1% Tween in 1× PBS, resuspended in 1 mL of 100% methanol, and incubated at −30 °C for 5 min. Then, worms were washed twice with 1 mL of 0.01% Tween in 1× PBS. Following the final wash, the supernatant was aspirated to a final volume of 200 μL. A 1 μL aliquot of

DAPI stain (1 mg/mL stock) was added, and the samples were incubated in the dark for 5 min, followed by two washes with 0.01% Tween in 1× PBS. Finally, the supernatant was aspirated and worms were resuspended in 25 μL of 75% glycerol before being mounted onto a microscope slide.

## Microscopy

Confocal microscopy was performed using a Leica TCS SP8 confocal microscope (Leica, Wetzlar, Germany) equipped with a photomultiplier tube, two HyD hybrid detectors, and lasers at 405 nm, 488 nm, 552 nm, and 638 nm. Images were captured with Leica PL APO CS2 objectives (10×/0.40 NA air, 40×/1.30 NA oil, and 63×/1.40 NA oil). Image processing and analysis were conducted using Leica LAS X software (v. 3.5.5). For DHS-3::GFP quantification in Fig. 1c, Supplementary Fig. 1c, d, Z-stacks comprising 30 optical sections were acquired across a total depth of 8.7 μm along the proximal intestine to account for tissue thickness. Images were collected in LIGHTNING mode with the Leica LAS X software, enabling adaptive deconvolution for each voxel using Leica's Super-Resolution setting. Z-stacks were then analyzed in Fiji (ImageJ) with the 3D Objects Counter plugin. Thresholding was defined at the innermost Z-plane (i.e., 15th slice out of 30). Total volume represents the sum of the volume of all individual objects. Average lipid droplet volume was determined by dividing total volume by total object count. For dissecting scope microscopy, six worms per condition were immobilized on unseeded NGM plates containing 100 μg/mL carbenicillin plates using a 10 μL drop of 200 mM levamisole. Images were acquired using a ZEISS Axio Zoom.V16 (ZEISS, Oberkochen, Germany) stereo microscope equipped with a ZEISS PlanNeoFluar Z 1x/0.25 FWD 56 mm objective. A ZEISS AxioCam 305 mono camera was used to capture micrographs under transmitted light and standard fluorescence filter settings. Exposure settings and image processing parameters were held constant across all samples to ensure consistency. Image acquisition was performed by ZEISS ZEN software (v3.11.105.00000).

## Oil-Red-O staining

Neutral triglyceride fat stores were visualized using Oil-Red-O staining. The stain was prepared by dissolving 0.5 g of Oil-Red-O in 100 mL of isopropanol, rocking for 3 days, diluting to 60% with milliQ-H$_2$O, and filtering through a 0.45 μm filter. Approximately 500 worms per condition were collected using M9 buffer and pelleted by centrifugation (1000 × *g*, 30 s). The worms were washed three times with 1× PBS to remove residual bacteria. To permeabilize the cuticle, the worms were incubated for 1 h at room temperature with gentle rocking in 1× Modified Ruvkun's Witches Brew buffer (diluted from a 2× stock: 3.7% formaldehyde, 160 mM KCl, 40 mM NaCl, 14 mM EDTA, 0.2% βME). Following three washes with 1× PBS, worms were incubated in 60% isopropanol for 15 min with rocking before being stained for 14 h in 60% Oil-Red-O. After staining, the dye was removed and replaced with 1x PBS containing 0.01% Triton X-100, and the samples were gently rocked for 2 min. The worms were then washed twice with 1× PBS and mounted onto 2% agarose pads for visualization using a ZEISS Axio Observer.Z1 (ZEISS, Oberkochen, Germany) inverted microscope equipped with a ZEISS AxioCam MRc color camera and a ZEISS Plan-Apochromat 20×/0.8 WD 0.55 mm air immersion objective. Image acquisition was performed by ZEISS ZEN software (v3.11.105.00000).

## Preparation of worm extracts for proteomics

Age-synchronized worms were collected by washing plates with M9 buffer and pelleted via centrifugation at 1000 × *g* for 30 s. The pellets were washed once with M9 buffer, transferred to 1.5 mL microcentrifuge tubes, flash-frozen in liquid nitrogen, and stored at −80 °C. For protein extraction, worm pellets were homogenized via bead beating in non-denaturing lysis buffer (100 mM HEPES pH 7.4, 300 mM NaCl, 2 mM EDTA, 2% Triton X-100) supplemented with EDTA-free

protease inhibitor and PhosSTOP phosphatase inhibitor cocktails (Cat. # 4906845001, Roche). Lysates were clarified by centrifugation at $8000 \times g$ for 5 min at 4 °C, and the supernatants were collected. Protein concentrations were determined using the Pierce BCA Protein Assay Kit (Cat. # PI23225, Thermo Scientific) and all samples were adjusted to a uniform concentration.

## Immunoprecipitation

Immunoprecipitation (IP) of NHR-49::GFP was performed using either GFP-Trap Magnetic agarose beads (Cat. #gtma, ChromoTek) or SureBeads Protein A Magnetic Beads (Cat. #1614013, Bio-Rad) pre-incubated with a GFP Polyclonal Antibody (Rabbit; Cat. #A6455, Invitrogen). The type of beads used is detailed in the corresponding figure legends. For GFP-Trap beads, lysates were incubated with equilibrated beads for 1 h at 4 °C with end-over-end rotation. Beads were washed three times, and bound proteins were eluted by boiling in 80 µL of 2× Laemmli SDS sample buffer (Cat. #1610747, Bio-Rad) for 5 min. For SureBeads, the beads were pre-incubated with the GFP antibody, washed, and then incubated with lysates for 1 h at 4 °C with gentle agitation. Elution was performed by heating in 2× SDS sample buffer at 90 °C for 10 min. For co-immunoprecipitation (Co-IP) analysis, eluted proteins were resolved on a 10% SDS-PAGE gel, run ~1 cm, excised, and diced for downstream LC-MS analysis. For post-translational modification (PTM) analysis, the entire band corresponding to NHR-49::GFP was excised from the gel and processed as described in the Mass Spectrometry and Proteomic Analysis methods section.

## Mass spectrometry and proteomic analysis

For whole-worm lysate proteomics, lysates were mixed with equal parts 2× Laemmli SDS sample buffer (Cat. #1610747, Bio-Rad) and run 1 cm into an SDS-PAGE gel via electrophoresis prior to excision and dicing for analysis. For PTM analyzes, only the band corresponding to the protein of interest was excised and processed. Samples were digested overnight with mass spectrometry grade Trypsin protease (Cat. # 90057, Pierce) after reduction with dithiothreitol (DTT; Cat. #50-213-272, Fisher Scientific) and alkylation with iodoacetamide (Cat. #I1149, Sigma-Aldrich). Peptides were purified via solid-phase extraction on an Oasis HLB µElution plate (Waters Corporation, Milford, MA). A 2 µL aliquot of each sample was injected into a Orbitrap Fusion Lumos (Thermo Fisher Scientific, Waltham, MA) or Orbitrap Eclipse mass spectrometer (Thermo Fisher Scientific) coupled to an UltiMate 3000 RSLCnano UHPLC system (Thermo Fisher Scientific). Chromatographic separation was achieved on a 75 µm inner diameter, 75 cm long EasySpray column (Thermo Fisher Scientific) using a 90-min gradient elution.

Mass spectra were acquired in positive ion mode. Full MS scans were obtained in the Orbitrap mass spectrometer at a resolution of 120,000, and MS/MS spectra was recorded for up to 10 precursor ions per scan. MS data was processed with Proteome Discoverer (v3.0, Thermo Fisher Scientific). Peptides were identified by SEQUEST-HT searches against the *C. elegans* reviewed protein database from UniProt (UP000001940; downloaded May 5, 2022). Search parameters and filtering criteria included a precursor mass tolerance of 10 ppm, a fragment ion tolerance of 0.6 Da, and an allowance of up to three missed cleavages. Carbamidomethylation of cysteine was set as a fixed modification, and methionine oxidation was a variable modification. Peptide identifications were filtered to a 1% false discovery rate (FDR). Protein quantification was based on the summed peak intensities of all peptides assigned to a given protein.

## Western blot analysis

The primary protocol for Western blotting is outlined in the Preparation of Worm Extracts for Proteomics methods section. This method involves preparing worm extracts using a native lysis buffer, which is followed by BCA protein quantification to ensure equal protein loading. After normalization based on total protein concentration, equal volumes of lysate were mixed with 2× SDS sample buffer (Cat. #1610747, Bio-Rad). Alternatively, 30–50 age-synchronized worms were collected in microfuge tubes and rinsed once with M9 buffer. Excess M9 was aspirated and the remaining pellet was resuspended in 20 µL of molecular-grade water and 20 µL of 2× SDS sample buffer. The samples were immediately boiled at 90 °C for 15 min with intermittent vortexing to ensure complete degradation of the worms. In most cases, freshly boiled extracts were processed further as described below or, in some cases, lysates were stored frozen at −30 °C. Proteins in the worm extracts were resolved by SDS-PAGE using bis-acrylamide gels and transferred onto nitrocellulose membranes. Membranes were blocked for non-specific binding via incubation in 0.1% PBS-Triton X-100 containing 5% Albumin bovine fraction V [BSA] (Cat. #A30075, Research Products International) at 4 °C with overnight shaking. After blocking, membranes were incubated with primary and fluorescent-conjugated secondary antibodies for 1 h at room temperature with four 10-min washes with fresh 0.1% PBS-Triton X-100 after both the primary and secondary antibody incubations. Primary antibodies used were: polyclonal anti-GFP (1:7500, Rabbit; Cat. #A6455, Invitrogen), monoclonal anti-TUBA4A (1:5000, Mouse; Cat. #T6074, Sigma). Secondary antibodies used were: polyclonal anti-Mouse IgG IRDye 680RD (1:15,000, Goat, Cat. #926-68070, LICORbio), polyclonal anti-Rabbit IgG IRDye 800CW (1:15,000, Goat, Cat. #926-32211, LICORbio). Additional antibody details found in Supplementary Data 3. Membranes were visualized on a ChemiDoc MP Imaging System (Bio-Rad) and protein bands quantified using Image Lab software (v6.1.0, Bio-Rad).

## In vitro kinase assay

Two separate experiments were performed to evaluate kinase activity via an adapted assay[70]. Experiment 1: Kinase Activity on a Wild-Type NHR-49 Peptide to test linear range: The first experiment assessed the activity of three recombinant kinases, CSNK1A1 (Cat. #PV3850, Thermo Fisher Scientific), PKA (Cat. #14-440, Sigma-Aldrich), MAPK3/ERK1 (Cat. #PV3311, Thermo Fisher Scientific), on a wild-type peptide derived from NHR-49. The peptide sequence (residues 110–128) was D-P-V-G-S-P-T-S-G-A-S-L-N-G-T-P-F-K-K-K, with an additional lysine added for charge. Reactions were prepared in a buffer containing 50 mM Tris-HCl, 10 mM $MgCl_2$, 2 mM DTT, 1 mM ATP, 0.01% Triton X-100, 1 mM peptide, and varying concentrations of $[\gamma\text{-}^{32}P]$ ATP depending on radioactive decay. Reactions were initiated by adding the kinase (at varying concentrations to test for linear range), and the mixture was incubated at 30 °C for 10 min. The reaction was terminated by adding 200 mM phosphoric acid. Aliquots were spotted onto P81 phospho-cellulose filters, which were then washed three times for 10 min each in 75 mM phosphoric acid, followed by a 5-min wash in methanol. After air-drying, the incorporation of $[\gamma\text{-}^{32}P]$ was quantified using a LS6500 multi-purpose liquid scintillation counter (Beckman Coulter, Brea, CA) via Cerenkov counting. Experiment 2: Kinetics of $[\gamma\text{-}^{32}P]$ Incorporation: The second experiment evaluated the rate of $[\gamma\text{-}^{32}P]$ incorporation into two distinct peptides under the same buffer conditions as described above. A master mix was prepared and maintained at 30 °C. To create a time-course, 10 µL aliquots were removed at 30-s intervals over a 10-min period and added to separate reaction tubes pre-aliquoted with 200 mM phosphoric acid to terminate the reaction. The subsequent washing and quantification of $[\gamma\text{-}^{32}P]$ incorporation were performed as described in Experiment 1.

## Protein purification

*XPO1* (also known as CRM1) was purified as described[109] with a minor modification: 4 L of LB medium were used instead of 6 L. Briefly, CRM1 was purified from *E. coli* BL21 cells transformed with the CRM1 expression construct. Cells were grown in LB medium with ampicillin, induced with 0.5 mM IPTG at 25 °C for 10–12 h, and harvested after cooling to 4 °C. Pelleted cells were resuspended in Lysis A

buffer (40 mM HEPES pH 7.5, 2 mM magnesium acetate, 200 mM NaCl, 10 mM DTT, 1 mM benzamidine, 10 μg/mL leupeptin, 50 μg/mL AEBSF), and stored at −20 °C. After thawing, cells were lysed using a high-pressure homogenizer, and the lysate was clarified by centrifugation. The supernatant was applied to a glutathione-Sepharose column, which was sequentially washed with Buffers A, B, and C. CRM1 was released by on-column cleavage with TEV protease. The eluted protein was concentrated and further purified by size-exclusion chromatography. The final purity of CRM1 was confirmed by SDS-PAGE.

With respect to *NHR-49*, codon-optimized constructs for full-length NHR-49 and a fragment spanning residues 1–155 were cloned into the pET-28a(+) vector (GenScript, Piscataway, NJ). Recombinant proteins were expressed in *E. coli* Rosetta 2(DE3) cells (Cat. # 714003, MilliporeSigma) cultured in LB medium with 50 μg/mL kanamycin and 25 μg/mL chloramphenicol. Protein expression was induced with 1 mM IPTG at an $OD_{600}$ of 0.6–0.8, followed by incubation at 20 °C for 5 h. Cells were harvested at an $OD_{600}$ of 2.5–3.0 and stored at −80 °C. For lysis, pellets were resuspended in $His_6$ lysis buffer (50 mM Tris-HCl pH 7.5, 300 mM NaCl, 1% Triton X-100, 10 mM imidazole, 5% glycerol, 5 mM β-mercaptoethanol [βME], 1 mM PMSF). Cell lysis was performed by sonication, and lysates were clarified by centrifugation at $17,000 \times g$ for 30 min at 4 °C. The supernatant was incubated with 1 mL of TALON affinity resin (Cat. #635501, Takara Bio). The resin was washed once with $His_6$ lysis buffer and three times with $His_6$ wash buffer (50 mM Tris-HCl pH 7.5, 300 mM NaCl, 20 mM imidazole, 5% glycerol, 2.5 mM βME). Bound proteins were eluted with $His_6$ elution buffer (50 mM Tris-HCl pH 7.5, 50 mM NaCl, 250 mM imidazole, 2.5 mM βME). Eluted proteins were buffer-exchanged into storage buffer (50 mM Tris-HCl pH 7.5, 50 mM NaCl, 5% glycerol, 2.5 mM βME) using a NAP-10 desalting column (Cat. # 17085401, Cytiva), concentrated to ~1 mg/mL with Amicon Ultra centrifugal filters 10 kDa cutoff (Cat. #UFC8010, MilliporeSigma), and flash-frozen in liquid nitrogen for storage at −80 °C. Purity and concentration were assessed by Coomassie-stained SDS-PAGE, with concentration determined by comparison to BSA standards.

With respect to *WWP-1* and *UBC-18*, plasmids encoding GST-tagged WWP-1 or UBC-18 in the pGEX-KG backbone were expressed in *E. coli* Rosetta 2(DE3). Cells were grown in LB medium with 100 μg/mL ampicillin and 25 μg/mL chloramphenicol. GST-WWP-1 expression was induced with 1 mM IPTG at an $OD_{600}$ of 0.6–0.8 and continued for 6 h at 20 °C. GST-UBC-18 expression was induced with 1 mM IPTG at an $OD_{600}$ of ~0.4 and continued for 3 h at 37 °C. Cells were harvested at an $OD_{600}$ of 1.0–1.5 and stored at −80 °C. Thawed pellets were resuspended in PBS lysis buffer (1× PBS pH 7.4, 1% Triton X-100, 5% glycerol, 1 mM DTT) containing Bacterial ProteaseArrest inhibitor cocktail (Cat. # 786-432, G-Biosciences). Cells were lysed by sonication, and the lysates were clarified by centrifugation. For purification, lysates were incubated with 1 mL of glutathione affinity resin (Cat. # L00206, GenScript) for 1 h at 4 °C. The resin was washed four times with PBS wash buffer (1x PBS pH 7.4, 1% Triton X-100, 5% glycerol, 1 mM DTT). GST-WWP-1 was eluted with glutathione elution buffer (50 mM Tris pH 8.0, 10 mM reduced glutathione) and buffer-exchanged into storage buffer (50 mM Tris pH 7.5, 50 mM NaCl, 5% glycerol, 1 mM DTT) using a NAP-10 column. The GST tag was removed from UBC-18 using a thrombin cleavage capture kit (Cat. #69022, Millipore), and the thrombin was subsequently removed with streptavidin agarose. Both purified proteins were concentrated, flash-frozen, and analyzed as described for NHR-49.

### Peptide array
A PepStar peptide array (JPT) was designed to span all 501 amino acids of NHR-49 isoform C. The array consisted of 15-mer peptides with an 11-amino acid overlap, for a total of 123 spots on the cellulose membrane. In this adapted protocol[110,111], the array-containing membrane was pre-treated with methanol for 10 min, washed three times with 1× TBS for 20 min each, and blocked overnight with 5% Albumin bovine fraction V [BSA] (Cat. #A30075, Research Products International). The next day, the membrane was incubated overnight at 4 °C with 350 nM human recombinant XPO1 and 1050 nM yeast recombinant GSP1 (RAN1 ortholog) in 2.5% BSA and XPO1 purification buffer. After three 20-min washes in 1× TBS at 4 °C, the peptide-bound XPO1 was transferred onto a PVDF membrane using a semi-dry blotting system (1.0 mA/cm², 4 °C, 30 min). The PVDF membrane was blocked overnight with 5% BSA, washed, and probed with an monoclonal anti-XPO1 antibody (1:1000, Mouse, Santa Cruz, Cat. #sc-74454). Subsequent detection was performed via incubation with a fluorescent-conjugated secondary antibody polyclonal anti-Mouse IgG IRDye 680RD (1:15,000, Goat, Cat. #926-68070, LICORbio) for 1 h at room temperature followed by four separate 10-min rinses with 1× TBS. The array-containing membranes were visualized on a ChemiDoc MP Imaging System (Bio-Rad), and the resulting images were processed using Image Lab software (v6.1.0; Bio-Rad).

### Ubiquitination assay
Ubiquitination reactions were performed in a 20 μL volume containing the following components: 100 nM human E1 ligase UBE1 (R&D Systems), 500 nM E2 ligase UBC-18, 500 nM GST-tagged WWP-1 or HECD-1 (ProteoGenix), 500 nM His₆-T7-tagged NHR-49, and 50 μM ubiquitin (R&D Systems). Reactions were initiated by the addition of 10× reaction buffer (250 mM Tris-HCl pH 7.5, 500 mM NaCl, 50 mM $MgCl_2$, 50 mM ATP, 5 mM DTT) and incubated at 37 °C for the specified durations. The reactions were terminated by adding an equal volume of 2× Laemmli SDS sample buffer (Cat. # 1610737, Bio-Rad). Reaction products were resolved by SDS–PAGE and analyzed via western blotting using an monoclonal anti-T7 primary antibody (1:5000, Mouse, Cat. #69522 MilliporeSigma) and a polyclonal HRP-conjugated anti-mouse secondary antibody (1:7500; Goat, Cat. #A4416, MilliporeSigma).

### In silico modeling of NHR-49
Protein-protein interactions were predicted using AlphaFold 3 technology on the AlphaFold Server. Protein sequences were obtained from UniProt. The confidence of the predicted structures is reported using two metrics: pTM and ipTM. pTM (predicted template modeling score): A score above 0.5 suggests that the predicted overall complex fold is likely similar to the true structure. ipTM (interface predicted template modeling score): This score measures the accuracy of the predicted relative positions of subunits within the complex. An ipTM score above 0.8 indicates high confidence in the prediction, while scores between 0.6 and 0.8 are considered a grey area, and scores below 0.6 are likely failed predictions. For ligand docking into the NHR-49 binding pocket, AutoDock-GPU (2021–present) was used. Data processing and analysis for this method were supported by the UT Southwestern Bioinformatics Core. All other structural modifications, annotations, and visualizations were performed using PyMOL (v3.1). Structures, including protein-protein or protein-ligand models, were imported into PyMOL for further analysis and visualization.

### GO Term/KEGG analysis
Quantitative analysis of proteomics and transcriptomics data was performed using WormEnrichr (https://maayanlab.cloud/WormEnrichr/). Specific methods for data cut-offs and sorting are detailed in the figure legends or text corresponding to each analysis.

### Statistics and reproducibility
All statistical analyzes were performed using Prism (v10.1.0, GraphPad, San Diego, CA) and Bioinformatics CLC Workbench (v9.5, Qiagen, Hilden, Germany). For large data sets, the ROUT method ($Q = 1\%$) was used to identify and exclude outliers. Comprehensive details on

statistical procedures, including sample sizes ($n$), measures of precision (e.g., SEM or 95% CI), specific statistical tests, and significance thresholds, are provided in the figure legends. All experiments denote 3 independent biological repeats unless specified in the figure legends. All data collection and analysis was performed blinded. Statistical significance was defined as $p < 0.05$.

### Reporting summary

Further information on research design is available in the Nature Portfolio Reporting Summary linked to this article.

## Data availability

Lipidomics data generated in this study is found in Supplementary Data 4. Transcriptomic data files generated in this study are deposited in the NCBI Gene Expression Omnibus (GEO) database under accession codes GSE314570 and GSE314580 (https://www.ncbi.nlm.nih.gov/geo/query/acc.cgi?acc=GSE314570, https://www.ncbi.nlm.nih.gov/geo/query/acc.cgi?acc=GSE314580).

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

## Acknowledgements

P.M.D and the members of the Douglas lab were supported by the Clayton Foundation for Research, the Welch Foundation (I-2061-20210327), the American Federation of Aging Research (AFAR 2023), the National Institutes of Health (R01AG076529, R01GM15385). L.T. was supported by a predoctoral fellowship from the National Institute of Health (F31GM140620). K.R.Z was supported by the National Science Foundation Graduate Research Fellowship Program (NSF GRFP 1000305318). We thank the Caenorhabditis Genetics Center (CGC) and SUNY Biotech for providing *C. elegans* strains. We are grateful to Dr. Yuh-Min Chook for guidance, plasmids, and laboratory space for the protein purification of XPO1/CRM1. We acknowledge the UT Southwestern Proteomics Core Facility, the UT Southwestern Lipidomics Core Facility, and Novogene for their services. We also thank the UT Southwestern Bioinformatics Core for their assistance with ligand docking. We appreciate critical reading and editing of the manuscript by Dr. Michael G. Douglas and Dr. Priyankaa Bhatia.

## Author contributions

L.T. jointly conceived the study with P.M.D; L.T. and P.M.D. designed the experiments, implemented the methodology, interpreted the data, and wrote the manuscript. Along with L.T. and P.M.D.; J.K., R.S.F., K.F., J.M.W., G.O., A.C.J, K.R.Z., J.O., S.T.B., A.J.D., E.G.W., P.M., A.W., and M.E.F., all performed experiments directly used in this manuscript. K.R.Z. executed large set data analysis. V.A.L. and V.S.T. provided resources, guidance, and experimental support for all kinase-related assays while J.G.M provided resources, guidance and experimental support for lipidomics. P.M.D., J.K., R.S.F., and S.L.B.A. assisted in manuscript and figure editing. M.E.F. purified and performed all in vitro NHR-49 ubiquitination assays along with A.C.J, A.J.D., and E.G.W. Each of the following acquired funding: L.T., K.R.Z., and P.M.D.

## Competing interests

The authors declare no competing interests.
