## [Transparent Peer Review file · Nature Communications]

Silencing lipid catabolism determines longevity in response to fasting

Corresponding Author: Professor Peter Douglas

Version 0:

Reviewer comments:

Reviewer #1

(Remarks to the Author)

Intermittent fasting is known to enhance longevity and metabolic health, yet the relative contributions of lipid catabolism activation versus its attenuation during fasting remain unclear. Using *C. elegans*, Tatge and colleagues revealed that silencing lipid catabolism via ligand-independent regulation of the nuclear hormone receptor NHR-49 is critical for lifespan extension. The authors further identified casein kinase KIN-19 as a phosphorylational regulator of NHR-49 and its-mediated metabolic plasticity and fasting-induced longevity. This mechanism highlights ligand-independent NHR-49 silencing as a key adaptive strategy for longevity under intermittent fasting.

In general, the data are rigorous, the manuscript is well crafted, and the findings are largely novel. I have the following comments that would further strengthen the manuscript's merit:

1. MDT-15 regulates the expression of NHR-49-dependent and -independent fatty acid metabolism genes during fasting. What is the role of MDT-15 in KIN-19/NHR-49-modulated longevity in response to intermittent fasting?
2. Intermittent fasting modulates longevity by enhancing mitochondrial function and dynamics. In addition to the role of KIN-19/NHR-49 in lipid deposition, what is the role of KIN-19-regulated NHR-49-mediated mitochondrial function in intermittent fasting?
3. In line with comment #2, energy production in *nhr-49* mutants and other strains was solely assessed using the ATP availability assay. Multiple parameters need to be analyzed to evaluate cellular energy production more convincingly.

Reviewer #2

(Remarks to the Author)

In the manuscript by Tatge et al, the authors attempted to establish a link between nuclear hormone receptor NHR-49 and adult lifespan in *C. elegans* after fasting. A series of experiments were also performed to identify the potential regulators of NHR-49, which included KIN-19/casein kinase 1 alpha 1 and the nuclear export machinery component XPO-1. NHR-49 has been shown to control the expression of lipid metabolic genes by multiple research groups. This in part stemmed from gene expression profiling of *C. elegans* under different nutritional conditions. In some cases, additional dimerization partners, which are also nuclear hormone receptors, have been identified. The prevailing view is that distinct NHR-49 heterodimers control specific subsets of metabolic genes. In this study, the authors proposed another layer of regulation through post-translational phosphorylation of NHR-49 by KIN-19, which appeared to modify its nuclear/cytoplasmic distribution and subsequent stability. However, the authors did not elaborate on how phosphorylation of NHR-49 cross-talked with known mechanisms of regulation. More importantly, ligand-independent control of nuclear hormone receptors by phosphorylation is not a new concept, one that is well known to a general audience. Therefore, the current study is of limited interest to researchers who study NHR-49 biology in *C. elegans*.

Major concerns:

1. The title is misleading. The authors used *acs-2*, *fat-7* and *rab-11.2* as transcriptional readouts throughout the manuscript. They are not solely defined as lipid catabolic genes, based on annotation and as studied by other research groups. Furthermore, the use of "intermittent fasting" in the title and throughout the text did not match the experimental design. As shown in Extended Data Fig. 1, animals were allowed to develop in the presence of food, removed from food for one day,

and then re-fed again. This scheme deviated from established experimental protocols for intermittent fasting or dietary restriction.

2. The use of integrated transgenes that contained many copies of minigenes, over-expressing wild type and mutant versions of NHR-49c::GFP reduced the potential to translate the experimental results and concepts into other physiological settings, or in other organisms. The rationale behind the design of the delta-ligand binding domain construct of NHR-49, which was used to demonstrate ligand-independent regulation of NHR-49, was unclear. The gross removal of part of the ligand binding domain might grossly disrupt the entire structure that is C-terminal to the DNA binding domain. This could possibly expose part of the protein that is normally folded, causing neomorphic phenotypes. No attempts were made to address or rule out this possibility.

3. No data was provided to demonstrate the functionality of the wild type NHR-49c::GFP fusion protein. For example, could the transgene reverse the lifespan extension seen in *nhr-49* mutant animals after 1-day fasting? In the absence of rescue data, it is unclear if the mode of regulation of NHR-49c::GFP, as suggested by the authors, is applicable to endogenous NHR-49.

4. Based on publicly available RNAseq data at Wormbase, the expression levels of NHR-49 target genes *acs-2* (FPKM=1.3), *fat-7* (FPKM=28.4) and *rab-11.2* (FPKM=0), are very low in young adult animals. As a reference, the *fat-7* paralog *fat-6* is highly expressed (FPKM=623). This raised the question on the absolute contribution of ACS-2, FAT-7 and RAB-11.2 proteins to NHR-49 dependent, fasting-induced longevity. The expression levels of the three target genes were compared in relative terms (for example Fig. 2e, 2f), rather than absolute terms, thus further hindering the proper assessment of their functional impact.

5. The quantitation of DHS-3::GFP fluorescence as a measurement of “lipid droplet dynamics” (Line 90) and “lipid droplet levels” (Line 122) was confusing. The expression of the DHS-3::GFP fusion protein was under the control of the *dhs-3* promoter, from a multi-copy transgene that over-expressed DHS-3::GFP. Therefore, the reduction in DHS-3::GFP fluorescence could be attributed to the fasting-induced repression of the transgene, which might not be directly related to lipid droplet dynamics or levels at the organelle level.

6. The ATP reporter used (*wuls305[my-3p::Queen-2m]*), for example in Fig. 1d, was only expressed in the body wall muscle. It was unclear if the measurements could be used to reflect ATP levels in other tissues, such as the intestine, hypodermis, which are also metabolically active.

Reviewer #3

(Remarks to the Author)

In this manuscript entitled “Silencing lipid catabolism determines longevity in response to intermittent fasting,”, the authors investigated how lipid metabolic changes regulate fasting-induced longevity in *Caenorhabditis elegans*. In this paper, the authors found that silencing lipid catabolism upon refeeding is crucial for lifespan extension. They identified the nuclear hormone receptor NHR-49 as a key regulator that promotes lipid catabolism during fasting, and its inactivation after refeeding, mediated by phosphorylation through casein kinase KIN-19, was essential for longevity. The study suggests that KIN-19 phosphorylates NHR-49, promoting its nuclear export and degradation, thereby silencing lipid catabolic genes. These results highlight that the proper regulation of lipid catabolism is a key determinant of fasting-induced longevity. Following are my concerns that the authors need to address.

Major comments

1. The authors rely heavily on transgenic expression of NHR-49 variants in an *nhr-49(nr2041)* background. This introduces significant potential artifacts due to non-physiological overexpression, and copy number variation. Given the importance of precise regulation of NHR-49 activity in this study, and relative easiness of CRISPR genome editing, it is essential for the authors to generate single-copy CRISPR knock-in strains that express tagged and mutant forms of NHR-49 from the endogenous locus.
2. This study uses RNAi knockdown of *kin-19* as the only approach to study its function. There is no *kin-19* loss-of-function mutant to validate the RNAi results. In addition, they did not attempt to test potential gain of function effect of *kin-19* by overexpression. The authors fail to demonstrate that endogenous KIN-19 phosphorylates NHR-49 in vivo through genetic epistasis or rescue experiments.
3. While the use of AlphaFold3-predicted iPTM scores is nice for preliminary in silico analysis, these predictions are not substitutes for biochemical validation. For instance, the interactions between KIN-19 and NHR-49, as well as XPO-1 and NHR-49, are primarily inferred via AI modeling and synthetic peptides, not co-immunoprecipitation of endogenous proteins. In addition, in vitro kinase assays failed to show phosphorylation of unprimed NHR-49 by recombinant KIN-19, but this is central to the model proposed. Moreover, the AI-predicted interaction improvement upon S114 phosphorylation (iPTM increase to 0.69) is suggestive but does not establish a functional or physical interaction. Overall, the conclusions about phosphorylation-driven attenuation and degradation of NHR-49 remain highly speculative in the absence of stronger biochemical evidence and need to be experimentally validated with rigorous biochemical experiments.
4. The authors admit that the priming phosphorylation at S114, essential for the proposed model, remains unidentified. They attempted to test MPK-1 but showed that it did not phosphorylate the relevant site. They did not pursue alternative kinases (e.g., PKA, AMPK) experimentally, leaving a significant mechanistic gap in their model.

Minor comments

1. It would be valuable to test additional dietary restriction paradigms using the *kin-19* and *nhr-49* genetic models to assess

whether the observed effects are specific to intermittent fasting or more broadly applicable.

2. In Extended Data Figure 1.i, the authors only presented downregulated KEGG pathways under IF conditions. Please provide the list of upregulated KEGG pathways to allow a more comprehensive interpretation of the transcriptomic changes induced by intermittent fasting.
3. Please add a brief explanation for why the authors utilized sodium azide as a negative control in the ATP sensor assay. This will help improve clarity for readers.
4. Please consider adding a brief discussion about how intermittent fasting modulates KIN-19 expression or activity. It could provide insights into upstream regulatory mechanisms that remain unexplored in this study.
5. It will be better to enhance the resolution of Figures 2c.

Version 1:

Reviewer comments:

Reviewer #1

(Remarks to the Author)

The authors have satisfied the request, and I have no further comments.

Reviewer #2

(Remarks to the Author)

Response Line 249: One exception is a mutational screen that identified three gain-of-function mutations within or near the ligand-binding pocket, a study that we cite in both the original and revised manuscript, which was pivotal in guiding our engineering of the ligand-binding domain truncation strain (Lee, Goh et al. 2016).

The gain of function nhr-49 mutants were originally identified by the Pilon lab (PMID: 24068966). Please amend citation in the manuscript.

Response Line 495: Additionally, we performed thin layer chromatography (TLC), which revealed that 24 hours was sufficient to deplete triglyceride stores (TAG), while preserving minimal lipid reserves in the form of free fatty acids and cholesterol. Unlike the 48-hour fast, which not only exhausted TAGs, but also available free fatty acids and cholesterol.

No details were provided for the method used to perform the TLC analysis (Fig. 1a). Depending on the solvent system, additional lipid species should be detected by TLC (PMID: 9527856). Therefore, it is unclear if the entire TLC plate is displayed. For example, cholesterol esters are missing from the TLC analysis although they are well known to be stored in lipid droplets.

Response Line 573: We opted to retain helices 1, 3–5 as well as the critically helix 12, which undergoes conformational changes to mediate coactivator binding (Rastinejad, Huang et al. 2013). By retaining this region, we aimed to preserve co-factor interactions, while abolishing the receptor's ligand-binding function. The structural visualization of this truncation strategy is now included in Fig. 2a of the revised manuscript, and we have included additional text in the revised manuscript in lines 208-213 to further explain our rationalization.

The interpretation of the structural studies on nuclear receptors is over-simplistic. The ligand binding domain, as a well-folded structure, binds ligand, interacts with transcriptional co-factors, and interacts with each other (dimerization). (PMID: 37356665) The authors cannot rule out the possibility that the delta295-422 NHR-49 mutant has lost its dimerization function, or created new surface of the protein that permitted aberrant protein interactions. The presentation of Fig. 2a was misleading. It gave the impression that the remaining helices of the LBD adopted exactly the same conformation in the absence of the blackened helices. Caution should also be exercised when interpreting the alphafold predicted structure of NHR-49. The single conformation in the predicted structure has helix 12 in a position that does not entirely resemble its known position in other ligand-bound, activated LBD of nuclear receptors. Instead, the NHR-49 helix 12 appeared to block the well-studied co-activator binding surface. The structure of the nematode AceDAF-12 may serve as an additional reference (PMID: 22170062).

It is hard to interpret the results from the limited proteolysis assay (reviewer comment Fig. 6) because the same three fragments present in wildtype and the delta295-422 mutant were shown. Which region of NHR-49 did these three fragments represent? Does any of them correspond to part of the LBD? Should there be a fragment that corresponded to 295-422, which is only present in the wildtype sample? The incomplete results of this assay casted doubt on its utility as a readout of structural integrity, especially the LBD.

Response Line 789: On this foundation, we employed DHS-3::GFP to track lipid droplet dynamics within intestinal cells, the major site of lipid storage in *C. elegans*, and have clarified as such at lines 102-109 in the revised manuscript. We note that DHS-3::GFP has been the community standard for over a decade, used in at least 15 independent publications.

Manuscript Line 102-103: "Moving forward, we used a fluorescence-based system to monitor, in real time, the dynamics of TAG-enriched lipid droplets in living animals during fasting and refeeding."

The original query was not about the validity of using DHS-3::GFP as a lipid droplet marker. What was problematic was the use of GFP fluorescence intensity (y-axis label, Fig. 1d, 1h, 4c, 4h) as a measurement of lipid droplet dynamics, which should mean the size and number of lipid droplets. The change in GFP fluorescence could be due to transcriptional regulation of dhs-3 (as shown by reviewer comment Fig. 7), which may or may not be the result of a change in the size and number of lipid droplets. The authors should directly measure the size and number of lipid droplets in control and treatment groups.

Reviewer #3

(Remarks to the Author)

The authors responded to my comments in a highly constructive manner and have successfully addressed my concerns. The manuscript is now significantly improved.

Version 2:

Reviewer comments:

Reviewer #2

(Remarks to the Author)

In the revised manuscript, the authors have sufficiently addressed my previous comments.

Response to Reviewers for:
Silencing lipid catabolism determines longevity in response to fasting
Tatge, L., et al

Reviewer's comments are in black and/or **bolded** and *italicized*.

Author's responses to comments are in blue.

**Reviewer #1** (Remarks to the Author):

Intermittent fasting is known to enhance longevity and metabolic health, yet the relative
contributions of lipid catabolism activation versus its attenuation during fasting remain unclear.
Using *C. elegans*, Tatge and colleagues revealed that silencing lipid catabolism via ligand-
independent regulation of the nuclear hormone receptor NHR-49 is critical for lifespan extension.
The authors further identified casein kinase KIN-19 as a phosphorylation regulator of NHR-49 and
its-mediated metabolic plasticity and fasting-induced longevity. This mechanism highlights ligand-
independent NHR-49 silencing as a key adaptive strategy for longevity under intermittent fasting.

In general, the data are rigorous, the manuscript is well crafted, and the findings are largely
novel. I have the following comments that would further strengthen the manuscript's merit:

1. MDT-15 regulates the expression of NHR-49-dependent and -independent fatty acid
metabolism genes during fasting. **What is the role of MDT-15 in KIN-19/NHR-49-modulated**
**longevity in response to intermittent fasting?**

**Background:** MDT-15, first characterized in (Taubert, Van Gilst et al. 2006), is defined as a
subunit of the mediator complex, and a transcriptional co-activator interacting with NHR-49.
Initially, it was identified for its role in lipid homeostasis. Since then, MDT-15 has been implicated
in a wide array of regulatory networks, such as oxidative stress by interacting with SKN-1 (Goh,
Martelli et al. 2014); directly or indirectly through FOXO signaling (Zhang, Judy et al. 2013); innate
immune response and detoxification (Pukkila-Worley, Feinbaum et al. 2014); hypoxia and
extracellular matrix injury (Vozdek, Long et al. 2018); toxic metal stress (Shomer, Kadhim et al.
2019); mitochondrial surveillance pathways involved in immunity, detoxification, and antiviral
pathways (Mao, Breen et al. 2022). It also regulates amino acid catabolism (Frankino, Siddiqi et
al. 2022); propionic acid detoxification (Goh, Beigi et al. 2023); and has been most recently
implicated in transgenerational signaling associated with pathogen perception (Pender, Dishart et
al. 2025).

**Approach:** In response, we implemented an orthogonal strategy to evaluate the potential
relationship between MDT-15 and NHR-49 in the fasting response and longevity. While MDT-15
is known to participate in multiple longevity-associated pathways, our aim was to test whether it
interacts directly with NHR-49 and/or functions within the same regulatory axis in specific nutrient
states.

- (1) We re-analyzed all our available NHR-49 interactome datasets to evaluate physical
associations with MDT-15 via interaction profiling. These included: co-
immunoprecipitation/mass spectrometry (IP/MS) of NHR-49::GFP in fed and starved
conditions, IP/MS of NHR-49::YFP wild-type, S114A, and $\Delta 295$ –422 truncation
variants, IP/MS of NHR-49::GFP under *kin-19* RNAi, and TurboID proximity labeling
of NHR-49 tagged at both N- and C-termini.
(2) We next analyzed *mdt-15* transcriptional dynamics across multiple conditions to
explore regulatory links between MDT-15 and NHR-49.
(3) We assessed functional relevance by examining the genetic interaction between *mdt-*
*15* and *nhr-49* during fasting by performing lifespan assays in both wild-type and *nhr-*
*49(nr2041)* mutant worms, with or without *kin-19* and *mdt-15* RNAi treatment.

**Results:** First, we examined the potential for direct interaction between NHR-49 and MDT-15 in
the context of basal and fasting conditions. In our NHR-49::GFP IP/MS with starvation, we
identified one mediator subunit, MDT-4, interacting with NHR-49::GFP under starvation conditions
(Coverage – 4%, Peptides – 1, PSMs – 1, unique peptides – 1). No mediator subunits were found
in the NHR-49::YFP IP/MS with the full length, S114A mutation, or $\Delta 295$ –422 truncation strains.

In our NHR-49::GFP IP/MS with *kin-19* RNAi compared to empty vector control, MDT-15 was the
 only mediator protein detected. However, this was a low-confidence interaction due to its low
 coverage (2%), single peptide (1), PSM (1), and unique peptide (1). Lastly, in our endogenous
 TurboID proximity labeling subjected to LC-MS/MS, we detected five mediator subunits, MDT-
 6/18/4/30 and MDT-15. For MDT-6: Coverage – 11%, Peptides – 3, PSMs – 4, unique peptides –
 3. For MDT-18: Coverage – 3%, Peptides – 1, PSMs – 3, unique peptides – 1. For MDT-4:
 Coverage – 14%, Peptides – 5, PSMs – 23, unique peptides – 5. For MDT-30: Coverage – 3%,
 Peptides – 1, PSMs – 1, unique peptides – 1. For MDT-15: Coverage – 3%, Peptides – 3, PSMs
 57 – 5, unique peptides – 3. In summary, MDT-15 was only detected in two of our four datasets, with
 58 little coverage, PSMs, and abundances. It is worth noting that MDT-4 was identified with high
 confidence in two of the four NHR-49 interaction datasets. Based on our data and a lack of
 reporting from other groups, it remains unclear whether NHR-49 directly interacts with MDT-15.

 Next, we examined expression profiles to determine whether MDT-15 abundance changes during
 fasting and whether this was dependent on NHR-49. First, our fasting/refeeding transcriptomic
 analysis revealed that *mdt-15* is significantly upregulated during fasting and recovers to baseline
 levels upon refeeding. Given that MDT-15 regulates fatty acid metabolism genes that are both
 dependent and independent of NHR-49 during fasting, this observation is consistent with prior
 studies of MDT-15 function (Taubert, Van Gilst et al. 2006). Second, we analyzed our previously
 published datasets in which wild-type (N2) and *nhr-49(nr2041)* mutant worms were cultured on
 empty vector or *rab-11.1* RNAi. In these studies, (Watterson, Arneaud et al. 2022, Watterson,
 Tatge et al. 2022), *rab-11.1* RNAi induces a malabsorption state marked by a dramatic reduction
 in intracellular lipid levels. Importantly, *mdt-15* transcript levels remained unchanged in the *nhr-*
 *49* mutant on both empty

vector and *rab-11.1*
 RNAi, suggesting MDT-
 15 plays an independent
 role in nutrient uptake.
 Similarly, *mdt-15*
 transcript abundance
 was not affected by *kin-*
 *19* RNAi, which we
 propose mimics NHR-49
 hyperactivation akin to
 fasting (Please see
 **Reviewer Comment**
 **Figure 1**). Together,
 these data suggest that
 while *mdt-15* transcript
 levels are responsive to
 transient fasting, chronic
 lipid depletion via RNAi does not alter its expression.

Figure 1 – Transcriptomics of *mdt-15* from three different datasets. The panels, from left to right, show data from: Fed, fasted, and refeed conditions; *rab-11.1* RNAi in both N2 (wild-type) and *nhr-49(nr2041)* mutant strains; and *kin-19* RNAi.

 Lastly, we examined aging phenotypes to determine whether an epistatic interaction exists
 between MDT-15 and NHR-49 during transient fasting or chronic *kin-19* RNAi treatment. Using
 wild-type (N2) and *nhr-49(nr2041)* null mutants in combination with *mdt-15* RNAi, we observed
 intriguing phenotypes that could warrant an independent investigation. In wild-type animals, *mdt-*
 *15* RNAi abolishes fasting-induced lifespan extension. Surprisingly, however, a modest lifespan
 extension persisted in *nhr-49* mutants treated with *mdt-15* RNAi under fasting conditions (Please
 see **Reviewer Comment Figure 2**). These results indicate that NHR-49 and MDT-15 act within

the same genetic pathway in the basal fed
 state. During acute fasting, the absence of
 NHR-49 appears to override the effects of *mdt-15*
 RNAi knockdown, implying that NHR-49
 likely functions downstream of MDT-15. A
 similar trend was observed under chronic lipid
 depletion: fasting-induced lifespan extension
 occurred in *nhr-49* mutants but not in wild-type
 animals co-treated with *kin-19* and *mdt-15*
 RNAi. Together, these findings indicate that
 while MDT-15 is required for fasting-induced
 lifespan extension, its role may be indirect and
 upstream of NHR-49.

 While we agree that these data are compelling
 and merit further investigation, including them
 in the current manuscript would detract from its
 overall clarity and focus. For this reason, we
 have chosen not to incorporate these results
 into the revised version.

Figure 2 - Lifespan analysis of worms under fed and fasted conditions (24 hours of dietary deprivation at day 1 of adulthood): **(Top)** Lifespan analysis of N2 (wild-type) and *nhr-49(nr2041)* mutant worms treated with *mdt-15* RNAi. **(Bottom)**: Lifespan analysis of worms treated with a mixture of *mdt-15* and *kin-19* RNAi.

2. Intermittent fasting modulates longevity by enhancing mitochondrial function and dynamics. In
 addition to the role of KIN-19/NHR-49 in lipid deposition, **what is the role of KIN-19-regulated**
 **NHR-49-mediated mitochondrial function in intermittent fasting?**

In response to the reviewer's insightful question regarding the role of KIN-19 in NHR-49-mediated
 mitochondrial function during fasting, we expanded our analysis beyond lipid regulation to assess
 mitochondrial physiology and cellular energetics (also see response to Comment #3 with more
 insight on the function of ATP as a surrogate read out). Fasting is well known to extend lifespan
 in part by enhancing mitochondrial biogenesis, dynamics, and oxidative metabolism. Given NHR-
 49's established role in mitochondrial β -oxidation and our findings that KIN-19 modulates NHR-
 49 activity, we sought to determine, in addition to this comment, whether KIN-19 influences
 mitochondrial function through NHR-49 during fasting.

- (1) We reanalyzed our transcriptomic datasets from both fasting/refeeding and chronic *kin-19*
 RNAi treatments for mitochondrial associated genes, including those involved in the
 mitochondrial unfolded protein response in *hsp-6* and *hsp-60*. Both chaperones, along
 with numerous mitochondrial annotated genes, were differentially regulated in a similar
 fashion across fasting and *kin-19* RNAi, suggesting that KIN-19 and fasting converge on
 shared mitochondrial stress pathways (see revised manuscript, Lines 118-123, 413-
 416, and Extended Data Fig. 1e,f and Extended Data Fig. 6d,e).
- (2) To functionally assess cellular energetics, we attempted whole-worm ATP quantification
 using a luminescence-based assay (Invitrogen #A22066). Despite multiple trials with large
 worm pellets (>500 μ L), ATP concentrations consistently fell below the assay's dynamic
 detection range, when compared with picomole standards. Thus, we rely on the previously
 characterized ATP fluorescence reporter, which displayed minimal changes during fasting
 and refeeding in wild-type (N2) and *nhr-49(nr2041)* mutants when compared to irreversible
 inhibition of the mitochondrial cytochrome C oxidase (electron transport chain complex IV)
 via sodium azide (NaN_3); wild-type (N2) data moved to Extended Data Fig. 1g, *nhr-
 49(nr2041)* mutants moved to Extended Data Fig. 2f, and *kin-19* RNAi conditions move
 to Extended Data Fig. 6j,l.
- (3) Importantly, we quantified mitochondrial respiration by measuring oxygen consumption
 rates (OCR) in live animals under fasting/refeeding. Worms subjected to *kin-19* RNAi
 consumed 50.4% less oxygen than controls (average 61.3 pmol O_2 /min vs. 121.7 pmol
 O_2 /min, respectively), confirming a substantial impairment in bioenergetic capacity; see
 lines 434-439 and Fig. 4g.
- (4) We assessed mitochondrial morphology using an endogenously tagged COX-4::GFP
 strain as well as an ectopically-expressed, muscle-specific *myo-3p::GFP* reporter. In both
 the intestinal and muscle mitochondrial networks, *kin-19* RNAi phenocopied *atp-3* RNAi
 (mitochondrial electron transport chain complex V or ATP synthase subunit), producing
 fragmented mitochondria consistent with mitochondrial dysfunction (Extended Data Fig.
 6h,i). We have included additional text in the revised manuscript describing these
 mitochondrial morphology changes in lines 439-445.

In summary, this multi-tiered approach, which integrates transcriptomics, relative ATP levels,
 oxygen consumption rates, phenotypic readouts in animal velocity, and mitochondrial morphology,
 demonstrates that KIN-19 modulates mitochondrial physiology through NHR-49 during fasting at
 both transcriptional and functional levels. These data provide a substantial framework to address
 the reviewer's question and further support the central role of KIN-19 in metabolic regulation.

3. In line with comment #2, energy production in *nhr-49* mutants and other strains was solely
 assessed using the ATP availability assay. **Multiple parameters need to be analyzed to**
 **evaluate cellular energy production more convincingly.**

We appreciate the reviewer for raising this point, also mentioned by Reviewer #2. While ATP
 production is certainly important, it is not a central focus for the manuscript. Our rationale was to
 employ a validated transgenic tool in arguably the most metabolically demanding tissue during
 nutrient deprivation, where relative ATP fluctuations could be measured and linked to
 physiological function. Previous studies have already established the body wall muscle as a
 metabolically demanding tissue with high mitochondrial content, where age-dependent changes
 in mitochondrial function and morphology are readily observed as hallmarks of aging (Herndon,
 Schmeissner et al. 2002, Regmi, Rolland et al. 2014).

The body wall muscle was chosen because (i) bi-
 fluorescent ATP sensors are validated in this tissue, (ii)
 it exhibits clear age-dependent changes in
 mitochondrial function and output, and (iii) ATP
 fluctuations can be correlated with established
 behavioral readouts such as locomotor velocity. In
 contrast, although the pharyngeal muscle and
 hypodermis are also highly active, to our knowledge no
 ATP sensors have been reported or validated for these
 tissues. Nonetheless, we did find an intestinal ATP
 FRET (Förster Resonance Energy Transfer) sensor that
 has recently been characterized (Soto, Rivera et al.
 2020), but encountered substantial interference from the
 characteristic autofluorescence within the gut, leading to
 marginal and difficult-to-interpret results (Please see
 **Reviewer Comment Figure 3**).

Therefore, in response to the reviewer's concern that
 more parameters are needed to address cellular energy
 production, we sought to strengthen our analysis with
 complementary approaches outside of the available
 fluorescent reporters. As mentioned in the previous
 section (2), we performed whole-worm ATP
 quantification, oxygen consumption rate (OCR) analysis,
 and mitochondrial morphological imaging. In parallel, we
 mined our -omics datasets to provide a broader
 perspective on the mitochondrial transcriptome upon
 fasting/refeeding and *kin-19* RNAi.

To summarize our points from point (2), whole-worm ATP quantification using a luciferase-based
 commercial assay (Invitrogen #A22066) was unsuccessful. Despite multiple trials with increasing
 worm numbers, ATP concentrations consistently fell below the assay's dynamic detection range,
 even relative to standards in the picomole range. In contrast, oxygen consumption rate (OCR)
 analysis revealed robust differences in metabolic activity between control and *kin-19* RNAi worms
 after fasting and refeeding (day 3), confirming significant bioenergetic changes (see revised
 manuscript **Fig. 4g**). Complementing the OCR results, analysis of both endogenous (COX-
 4::GFP) and overexpressed (*myo-3p::GFP*) mitochondrial markers showed reduced
 mitochondrial content and increased fragmentation with *kin-19* RNAi, consistent with
 mitochondrial dysfunction (see revised manuscript **Extended Data Fig. 6h,i** as well as additional

Figure 3 - Intestinal FRET Sensor Analysis. Measures real-time ATP levels by ratioing the YFP to GFP fluorescence (both excited at 405 nm), which correlates with ATP concentration. Controls include an *atp-3* RNAi treatment to deplete ATP and a mutated, catalytically dead kinase to confirm signal specificity. These controls validate the sensor's ability to detect physiological changes in ATP levels.

text included in the revised manuscript in **lines 433-444**). Lastly, transcriptomics revealed
fluctuation in the transcriptional dynamics for several mitochondrial-related genes, many of which
overlap between fasting and *kin-19* RNAi (see revised manuscript **Fig. 6d,e** and additional text
included in **lines 118-123** and **413-416** of the revised manuscript).

Given the limited availability of validated ATP assessment tools for *C. elegans*, the technical
caveats of alternative approaches, and the fact that ATP quantification and/or cellular energy
production was not the primary focus of this study but rather a surrogate indicator of longevity, we
have opted to present all mitochondrial readout data in the Extended Data and to de-emphasize
its role in the main text.

**Reviewer #2** (Remarks to the Author):

In the manuscript by Tatge et al, the authors attempted to establish a link between nuclear
hormone receptor NHR-49 and adult lifespan in *C. elegans* after fasting. A series of experiments
were also performed to identify the potential regulators of NHR-49, which included KIN-19/casein
kinase 1 alpha 1 and the nuclear export machinery component XPO-1. NHR-49 has been shown
to control the expression of lipid metabolic genes by multiple research groups. This in part
stemmed from gene expression profiling of *C. elegans* under different nutritional conditions. In
some cases, additional dimerization partners, which are also nuclear hormone receptors, have
been identified. The prevailing view is that distinct NHR-49 heterodimers control specific subsets
of metabolic genes. In this study, the authors proposed another layer of regulation through post-
translational phosphorylation of NHR-49 by KIN-19, which appeared to modify its
nuclear/cytoplasmic distribution and subsequent stability. However, the authors **did not elaborate**
**on how phosphorylation of NHR-49 cross-talked with known “mechanisms” of regulation.**

Through extensive mutational analysis and RNAi treatment, multiple studies have shown that
NHR-49 contributes to diverse physiological roles in *C. elegans* (Doering, Ermakova et al. 2023).
However, most of these studies remain largely phenomenological and do not define the underlying
molecular mechanisms of NHR-49 regulation. Direct quote - “Although no kinase has yet been
shown to directly target NHR-49, evidence for such regulation has begun to emerge,” (Doering,
Ermakova et al. 2023). For this reason, we found it challenging to directly “elaborate on how
phosphorylation of NHR-49 cross-talks with known mechanisms of regulation,” outside of our
proposed model: KIN-19 phosphorylates NHR-49 to attenuate fasting-induced transcriptional
activation.

Only recently was ChIP-seq data reported for NHR-49 (Gopal, Chaturbedi et al. 2025), however,
this analysis was not performed under fasting conditions (the focus of our investigation) and the
study has not yet undergone peer-review. Nevertheless, several groups have identified enzymes,
including kinases and E3 ligases, whose knockdown or mutation can modulate NHR-49 activity.
However, based on our literature survey, no study has defined distinct residues or reported direct
or indirect interactions between these enzymes and NHR-49. One exception is a mutational
screen that identified three gain-of-function mutations within or near the ligand-binding pocket, a
study that we cite in both the original and revised manuscript, which was pivotal in guiding our
engineering of the ligand-binding domain truncation strain (Lee, Goh et al. 2016).

At the transcriptional level, NHR-49 activity is known to be influenced by genetic interactions with
the mediator subunit MDT-15 and by heterodimerization with other nuclear receptors (Taubert,
Van Gilst et al. 2006, Pathare, Lin et al. 2012, Zeng, Li et al. 2021). In response to this reviewer’s
concern, we have incorporated our response to reviewer #1 on the interaction between how KIN-
19–mediated modulation of NHR-49 impacts MDT-15 expression levels.

**Background:** MDT-15, first characterized in (Taubert, Van Gilst et al. 2006), is
defined as a subunit of the mediator complex, and a transcriptional co-activator interacting
with NHR-49. Initially, it was identified for its role in lipid homeostasis. Since then, MDT-
15 has been implicated in a wide array of regulatory networks, such as oxidative stress
through interacting with SKN-1 (Goh, Martelli et al. 2014), directly or indirectly through
FOXO signaling (Zhang, Judy et al. 2013), innate immune response and detoxification
(Pukkila-Worley, Feinbaum et al. 2014), hypoxia and extracellular matrix injury (Vozdek,
Long et al. 2018), toxic metal stress (Shomer, Kadhim et al. 2019), mitochondrial
surveillance pathways involved in immunity, detoxification, and antiviral pathways (Mao,
Breen et al. 2022). It also regulates amino acid catabolism (Frankino, Siddiqi et al. 2022),
propionic acid detoxification (Goh, Beigi et al. 2023), and was most recently implicated in

transgenerational signaling associated with pathogen perception (Pender, Dishart et al.
2025).

**Approach:** In response, we implemented an orthogonal strategy to evaluate the potential
relationship between MDT-15 and NHR-49 in the fasting response and longevity. While
MDT-15 is known to participate in multiple longevity-associated pathways, our aim was to
test whether it interacts directly with NHR-49 and/or functions within the same regulatory
axis in specific nutrient states.

- (1) We re-analyzed all our available NHR-49 interactome datasets to evaluate physical
associations with MDT-15 via interaction profiling. These included: co-
immunoprecipitation /mass spectrometry (IP/MS) of NHR-49::GFP in fed and starved
conditions, IP/MS of NHR-49::YFP wild-type, S114A, and Δ 295–422 truncation
variants, IP/MS of NHR-49::GFP under *kin-19* RNAi, and TurboID proximity labeling
of NHR-49 tagged at both N- and C-termini.
(2) We next analyzed *mdt-15* transcriptional dynamics across multiple conditions to
explore regulatory links between MDT-15 and NHR-49.
(3) We assessed functional relevance by examining genetic interaction between *mdt-15*
and *nhr-49* during fasting by performing lifespan assays in both wild-type and *nhr-*
*49(nr2041)* mutant worms, with and without *kin-19* and *mdt-15* RNAi treatment.

**Results:** First, we examined the potential for direct interaction between NHR-49 and MDT-
15 in the context of basal and fasting conditions. In our NHR-49::GFP IP/MS with
starvation, we identified one mediator subunit, MDT-4, interacting with NHR-49::GFP
under times of starvation (Coverage – 4%, Peptides – 1, PSMs – 1, unique peptides – 1).
No mediator subunits were found in the NHR-49::YFP IP/MS with the full length, S114A
mutation, or Δ 295–422 truncation strains. In our NHR-49::GFP IP/MS with *kin-19* RNAi
compared to empty vector control, MDT-15 was the only mediator protein detected;
however, this interaction was low confident due to its coverage of 2%, peptides = 1, PSMs
= 1, and unique peptides = 1. Lastly, in our endogenous TurboID proximity labeling subject
to LC-MS/MS, we detected five mediator subunits, MDT-6/18/4/30 and MDT-15. For MDT-
6: Coverage – 11%, Peptides – 3, PSMs – 4, unique peptides – 3. For MDT-18: Coverage
299 – 3%, Peptides – 1, PSMs – 3, unique peptides – 1. For MDT-4: Coverage – 14%, Peptides
300 – 5, PSMs – 23, unique peptides – 5. For MDT-30: Coverage – 3%, Peptides – 1, PSMs
301 – 1, unique peptides – 1. For MDT-15: Coverage – 3%, Peptides – 3, PSMs – 5, unique
peptides – 3. In summary, MDT-15 was only detected in two of our four datasets, with little
coverage, PSMs, and abundances. It is worth noting that MDT-4 was identified with high
confidence in two of the four NHR-49 interaction datasets. Based on our data and a lack
of reporting from other groups, it remains unclear whether NHR-49 directly interacts with
MDT-15.

Next, we examined expression profiles to determine whether MDT-15 abundance changes during fasting and whether this was dependent on NHR-49. First, our fasting/refeeding transcriptomic analysis revealed that *mdt-15* is significantly upregulated during fasting and recovers to baseline levels upon refeeding. Given that MDT-15 regulates fatty acid metabolism genes that are both dependent and independent of NHR-49 during fasting, this observation is consistent with prior studies of MDT-15 function (Taubert, Van Gilst et al. 2006). Second, we analyzed our previously published datasets in which wild-type (N2) and *nhr-49(nr2041)* mutant worms were cultured on empty vector or *rab-11.1* RNAi. In these studies, (Watterson, Arneaud et al. 2022, Watterson, Tatge et al. 2022), *rab-11.1* RNAi induces a malabsorption state marked by a dramatic reduction in intracellular lipid levels.

Importantly, *mdt-15* transcript levels remained unchanged in the *nhr-49* mutant on both empty vector and *rab-11.1* RNAi suggesting an MDT-15 plays an independent role in nutrient uptake. Similarly, *mdt-15* transcript abundance was not affected by *kin-19* RNAi, which we propose mimics NHR-49 hyperactivation akin to fasting (Please see **Reviewer Comment Figure 1**). Together, these data suggest that while *mdt-15* transcript levels are responsive to transient fasting, chronic lipid depletion via RNAi does not alter its expression.

Lastly, we examined aging phenotypes to determine whether an epistatic interaction exists between MDT-15 and NHR-49 during transient fasting or chronic *kin-19* RNAi treatment. Using wild-type (N2) and *nhr-49(nr2041)* null mutants in combination with *mdt-15* RNAi, we observed intriguing phenotypes that could warrant an independent investigation. In wild-type animals, *mdt-15* RNAi abolished fasting-induced lifespan extension. Surprisingly, however, a modest lifespan extension

Figure 4 – Transcriptomics of *mdt-15* from three different data sets, left to right: Fed/Fasted/Refed, *rab-11.1* RNAi in an N2 and *nhr-49(nr2041)* strains, and *kin-19* RNAi.

Figure 5 - top to bottom: lifespan analysis upon fed and fasted treatment with (top) *mdt-15* RNAi in a N2 and *nhr-49(nr2041)* mutant background; and (bottom) a mixture of *mdt-15* and *kin-19* RNAi.

persisted in *nhr-49* mutants treated with *mdt-15* RNAi under fasting conditions (Please
 see **Reviewer Comment Figure 2**). These results indicate that NHR-49 and MDT-15 act
 within the same genetic pathway in the basal fed state. During acute fasting, the absence
 of NHR-49 appears to override the effects of *mdt-15* RNAi knockdown, implying that NHR-
 49 likely functions downstream of MDT-15. A similar trend was observed under chronic
 lipid depletion: fasting-induced lifespan extension occurred in *nhr-49* mutants but not in
 wild-type animals co-treated with *kin-19* and *mdt-15* RNAi. Together, these findings
 indicate that while MDT-15 is required for fasting-induced lifespan extension, its role may
 be indirect and upstream of NHR-49.

 While we agree that these data are compelling and merit further investigation, including
 them in the current manuscript would detract from its overall clarity and focus. For this
 reason, we have chosen not to incorporate these results into the revised version.

Regarding heterodimerization, we have clarified how excision of the ligand-binding domain of
 NHR-49 provides insight into its potential to act as either a homodimer or a heterodimer with other
 receptors in response to Reviewer #2's comments on the ligand binding domain truncation strain
 (see **lines 208-213 and 240-246** as well as **Fig. 2a** in the revised manuscript).

In addition, our prior work (Watterson, Arneaud et al. 2022, Watterson, Tatge et al. 2022) identified
 a subcellular regulatory mechanism for NHR-49 involving vesicular association and
 nucleocytoplasmic trafficking under lipid-depleted conditions. While that study employed
 ectopically overexpressed transgenes, our more recent work (Tatge 2025) demonstrates that the
 endogenous long isoform C of NHR-49, unlike the more abundant shorter isoforms in Day 1
 adults, exhibits a similar nucleocytoplasmic regulatory behavior. Building on these observations,
 we now show in this manuscript that phosphorylation at a single residue, S114, is sufficient to
 alter NHR-49 subcellular localization, favoring its nuclear accumulation.

More importantly, ligand-independent control of nuclear hormone receptors by phosphorylation is
 not a new concept, one that is well known to a general audience. Therefore, the **current study**
 **is of limited interest to researchers who study NHR-49 biology in *C. elegans*.**

We appreciate the reviewer's perspective, but we believe the significance of this work extends
 well beyond *C. elegans* NHR-49 biology.

- (1) Our finding that impairing activation of metabolic stress responses and lipid catabolism is
 dispensable for fasting-induced lifespan extension is both unexpected and broadly
 relevant to multiple fields, including aging, lipid metabolism, and other cellular stress
 responses.
- (2) This study introduces a new paradigm of fasting-induced lifespan extension and defines
 how both overall lipid droplet abundance and select fatty acid species fluctuate during this
 metabolic challenge.
- (3) We identify a non-canonical mechanism of exportin (XPO-1) in the absence of a canonical
 nuclear export signal, an observation only reported for the first time earlier this year (Ge,
 Brickner et al. 2025), which broadens understanding of alternate transcription factor
 interactors.
- (4) We show that casein kinase 1A1 (KIN-19 in worms) acts as a central attenuator of β -
 oxidation and provides a molecular mechanism underlying this regulation.
- (5) Because both casein kinase and exportin are highly conserved, and the relevant NHR-49
 residues are conserved in mammalian Hepatic Nuclear Factor 4, these findings have
 strong potential for translation beyond worms and are active areas of ongoing research in
 our laboratory.

Overall, the central focus of our manuscript is to understand how metabolic stress (caused by
fasting and refeeding) impacts age determination. We believe that the mechanisms uncovered
here will be of interest to the broader research community focused on aging, metabolism, and
stress-response biology.

Major concerns:

**1. The title is misleading.... Furthermore, the use of “intermittent fasting” in the title and**
**throughout the text did not match the experimental design.**

We thank the reviewer for their insightful comment. The study examines the effects of a single
fasting event on Day 1 of adulthood, rather than repeated fasting cycles, and we agree that this
does not align with the conventional definition of “intermittent fasting.” Our original use of the term
was intended to emphasize the biological importance of attenuation and reactivation dynamics
observed in our data. However, we recognize that this terminology may be misleading.
Accordingly, we have removed the word “intermittent” throughout the manuscript where it refers
to our experimental paradigm to ensure clarity and accuracy. To portray this more accurately, we
utilize the terms: transient fast or fasting throughout the revised manuscript. Our new schematic
can be found in the revised manuscript at **Extended data Fig. 1a**.

**The authors used *acs-2*, *fat-7* and *rab-11.2* as transcriptional readouts throughout the**
**manuscript. They are not solely defined as lipid catabolic genes, based on annotation and**
**as studied by other research groups.**

We thank the reviewer for this important clarification. Indeed, *acs-2*, *fat-7*, and *rab-11.2* are not
exclusively lipid catabolic genes, just as NHR-49 itself is not limited to regulating fasting-specific
responses. However, a substantial portion of the NHR-49 and *C. elegans* lipid metabolism
literature highlights *acs-2* (a fatty acid CoA synthetase) and *fat-7* (a Δ -9 desaturase) as important
metabolic enzymes during fasting. We selected these genes from a broader set of anabolic and
catabolic transcripts identified in our fasting and refeeding RNA-sequencing dataset based on
their widespread use as functional metabolic readouts. Both have been extensively cited in the
literature, and transcriptional repression of either gene, via RNAi or mutation, has been shown to
alter neutral lipid deposition in a manner consistent with their established roles: *acs-2*
knockdown/mutation leads to increased fat accumulation (Ashrafi, Chang et al. 2003, Van Gilst,
Hadjivassiliou et al. 2005, Zhang, Bakheet et al. 2011), while *fat-7* knockdown reduces lipid
deposition (Van Gilst, Hadjivassiliou et al. 2005, Horikawa, Nomura et al. 2008, Lemieux, Liu et
al. 2011). As cited:

“Changes in dietary energy nutrients, fasting, and refeeding affect hepatic ACS, CPT-I, and ACC
mRNA expression, and these results will serve to enhance our understanding of the molecular
mechanisms underlying regulation of fatty acid metabolism.” (Ryu, Sohn et al. 2005)

“[These] Δ -9 desaturases regulate the ratio of unsaturated to saturated fats and are
transcriptionally sensitive to the metabolic state of the cell, as they are induced during periods of
anabolic activity and repressed during periods of fat use (that is, starvation)” (Lemieux, Liu et al.
2011)

To streamline the narrative, we focused on *acs-2* as a representative catabolic gene for several
reasons: (1) It is among the most established NHR-49 targets (Van Gilst, Hadjivassiliou et al.
2005); (2) its expression is robustly induced by fasting and repressed by refeeding; (3) transgenic
tools exist and have been validated in prior studies; (4) consistent with recently reported NHR-49
ChIP-seq datasets (Gopal, Chaturbedi et al. 2025), transcript abundance for this gene targets

was abolished in *nhr-49(nr2041)* mutant animals under basal feed conditions. In parallel, *fat-7*
served as a representative repressed transcript that is also dependent on NHR-49 and matches
the above criteria. While *rab-11.2* was not reported in these NHR-49 ChIP-seq datasets, we
selectively used it for our candidate-based RNAi screens (*kinase* and *nucleocytoplasmic factors*)
due to its dynamic, starvation-responsive expression profile, as first described in (Watterson,
Tatge et al. 2022) and again in (Watterson, Arneaud et al. 2022). Its negligible baseline expression
and robust activation after 24 hours of fasting made it a practical and sensitive readout for
identifying starvation-activated genes, and screening “hits” that were identified based on the *rab-*
*11.2* reporter were further validated with *acs-2* and *fat-7*. We have updated the manuscript at **lines**
**153-156** to clarify our rationale and have added in the appropriate citations.

As shown in Extended Data Fig. 1, animals were allowed to develop in the presence of food,
removed from food for one day, and then re-fed again. **This scheme deviated from established**
**experimental protocols for intermittent fasting or dietary restriction.**

As mentioned above, we have removed the term “intermittent” from all instances related to our
experimental design. Regarding our deviation from certain established protocols in the *C. elegans*
field, our primary motivation was to avoid potential developmental complications associated with
fasting during larval stages and to maximize the likelihood of observing lifespan extension. A
foundational study in *C. elegans* lipid biology by Van Gilst and colleagues (Van Gilst,
Hadjivassiliou et al. 2005) introduced a “short-term food deprivation” RT-qPCR strategy. They
employed two main approaches: a kinetic assay, in which early L4 larvae were washed and
transferred to empty plates, then harvested after 1, 2, 4, 8, or 12 hours of fasting; and a fasting
assay, in which animals at 24h (L2), 36h (L3), 48h (L4), or 96h (Day 2) were fasted for 12 hours
and then collected. While this study defined key metabolic genes and their fasting kinetics,
including *acs-2* and *fat-7*, it failed to report aging phenotypes.

Another study (Kaeberlein, Smith et al. 2006) demonstrated that complete dietary deprivation in
early adulthood can robustly extend *C. elegans* lifespan. However, we deliberately diverged from
this method to incorporate a refeeding paradigm because our primary motivation was to avoid
confounding effects of prolonged starvation on healthspan and stress resilience, and instead
model a more physiologically relevant, transient nutrient deprivation that better reflects fasting–
refeeding cycles as observed in higher organisms. Additionally, since Kaeberlein et al. reported
lifespan extension independent of canonical longevity pathways such as *daf-2* and *sir-2.1*, we
sought to explore whether refeeding after short-term fasting could reveal distinct regulatory inputs,
particularly through lipid metabolic and stress-responsive transcriptional networks like NHR-49
and β -oxidation.

Another notable paradigm (Uno, Honjoh et al. 2013) involved a 48-hour fasting period followed
by food reintroduction. This study reported a near doubling of lifespan. However, we observed
dramatically high worm censorship rates (due to *C. elegans* specific “bag-of-worm” phenotypes)
in our preliminary experiments, making it challenging to assess health span and post-fasting
effects. To circumvent these issues, we adopted a more abbreviated 24-hour fasting period
followed by refeeding. From our previous studies (Watterson, Tatge et al. 2022), we had already
reported that 24-hours of starvation was sufficient to dramatically deplete availability of the 20-
carbon prenol lipid, geranylgeranyl pyrophosphate, whose conjugation to RAB GTPases plays a
critical role in NHR-49 activity and activation of lipid catabolism. Additionally, we performed thin-
layer chromatography (TLC), which revealed that 24 hours was sufficient to deplete triglyceride
stores (TAG), while preserving minimal lipid reserves in the form of free fatty acids and cholesterol.
Unlike the 48-hour fast, which not only exhausted TAGs, but also available free fatty acids and
cholesterol. Notably, we sought to limit the availability of neutral storage lipids in TAGs rather than
inducing membrane stress resulting from cholesterol exhaustion. Overall, this 24-hour fasting
window offered a metabolic “sweet spot” that triggers nutrient stress while preserving the capacity

for meaningful refeeding responses. Additionally, we prioritized modeling transient fasting–
 refeeding, rather than chronic deprivation, to more accurately reflect physiological conditions in
 higher organisms and probe distinct regulatory inputs such as how NHR-49 is silenced. To this
 point, we now include additional text in body of the revised manuscript at **lines 84-91** as well as
 the new TLC experiment in **Fig. 1a**.

**2. The use of integrated transgenes that contained many copies of minigenes, over-**
 **expressing wild type and mutant versions of NHR-49c::GFP reduced the potential to**
 **translate the experimental results and concepts into other physiological settings, or in**
 **other organisms.**

We appreciate the reviewer’s concern, which was also shared with reviewer #3. This too was a
 prominent concern for our group and one that we have been working to address long before
 submitting this present manuscript. In brief, a number of endogenously tagged NHR-49 worm
 strains (both N- and C-terminal) were either generate or obtained from other groups. With these
 strains, we performed extensive microscopy and proteomics to investigate how different isoforms
 of NHR-49 (at endogenous levels) were differentially distributed throughout the cell. The results
 from this study support the observations and interpretations made from worm strains that
 ectopically overexpressed NHR-49. We believe that these functional differences between the
 endogenous NHR-49 isoforms are an important observation for the field; one that should not be
 buried in the supplement of this story. To this end, we rapidly disseminated these discoveries to
 the greater research community and made them the central focus of their own study. Thus, please
 reference the new results, which were recently peer-reviewed and published in a short format
 article (Tatge 2025).

Please find a list of all the endogenously tagged NHR-49 strains used in the (Tatge 2025)
 publication plus an additional NHR-49 mutant strains created for these revisions. Due to the gene
 structure and presence of alternative translational start sites, N-terminal tagging selectively labels
 the long c-isoform of NHR-49, while C-terminal tagging labels all isoforms A-E. Our panel of
 endogenous strains are as follows:

Table 1 - Endogenous NHR-49 strains

Strain Name	SunyBiotech Name	Isoform Coverage	Alias	Genotype	Publication Reference/ Original Owner
PMD300	PHX9651	iso C	SfGFP::NHR-49	nhr-49c(syb9651[sfGFP::nhr-49c])	Tatge 2025/ Douglas Lab
PMD342	PHX5674	all	NHR-49::mCherry	nhr-49(syb5674[nhr-49::mCherry])	Tatge 2025/ Douglas Lab
STA07	PHX2863	all	NHR-49::GFPNovo2	nhr-49(syb2863[nhr-49::GFPNovo2])	Tatge 2025/ Taubert Lab
STA08	PHX2927	all	NHR-49::HA	nhr-49(syb2927[nhr-49::HA])	Tatge 2025/ Taubert Lab
PMD320	PHX10203	iso C	TurboID::3xHA::NHR-49	nhr-49c(syb10203 [3xHA::TurboID::nhr-49c])	Tatge 2025/ Douglas Lab
PMD319	PHX10204	all	NHR-49::3xHA::NHR-49	nhr-49(syb10204 [nhr-49::3xHA::TurboID])	Tatge 2025/ Douglas Lab

Using these new transgenic tools, we observed a nucleocytoplasmic distribution in the intestine,
 consistent with the localization of ectopically overexpressed NHR-49c::YFP (introduced at **lines**

**213-215** of the revised manuscript) and with NHR-49::GFP (Watterson, Arneaud et al. 2022,
Watterson, Tatge et al. 2022). Specifically, we saw this pattern using confocal microscopy of
sfGFP::NHR-49 and through LC-MS/MS analysis of cytosolic protein enrichment in
TurboID::3xHA::NHR-49 proximity labeling experiments. In contrast, C-terminally tagged versions
(e.g., GFPNovo2, mCherry, and TurboID) strongly favored a nuclear localization, as shown by
both microscopy and LC-MS/MS (Tatge 2025). These isoform-specific differences in subcellular
distribution help explain the discrepancies observed with cDNA-derived overexpression
constructs and provide valuable insight into NHR-49 biology (Tatge 2025).

In the present manuscript under review, we leveraged these endogenously tagged TurboID strains
both at the N- and C-terminus of NHR-49 and conducted a post-translational modification (PTM)
analysis on the resulting biotinylated proteins. Out of over 11,000 peptide fragments, 53 PTMs
were detected (37 phosphorylation, 16 ubiquitination). Among the phosphorylated peptides, three
were mapped to NHR-49. Most notably, we again detected phosphorylation at S114, with 15
PSMs in both the N- and C-terminal TurboID strains, ranking it as the 5th most abundant PTM
overall. We also detected a phosphorylation site on NHR-49 at S131 with a PSM of 2, observed
only in the C-terminal tagged TurboID strain. Furthermore, an additional low-confidence
phosphorylation was detected between residues 110 to 128, though it could not be precisely
mapped on the respective peptide, potentially due to its transient nature. Besides the S114
phosphorylation that has been identified in three independent data sets (endogenous, NHR-
49::GFP overexpression with fasting, and NHR-49::GFP with *kin-19* RNAi), other common
proteins that we discuss in this manuscript from NHR-49 overexpression datasets, were also
identified in the TurboID proximity labeling datasets (Tatge 2025), such as XPO-1 and KIN-19. In
the revised manuscript, we now include the Tatge, 2025 citation, additional text described the
above findings with the TurboID tagged NHR-49 between **lines 254-263** and new results in
**Extended Data Figure 4i**.

**The rationale behind the design of the delta-ligand binding domain construct of NHR-49,**
**which was used to demonstrate ligand-independent regulation of NHR-49, was unclear.**
**The gross removal of part of the ligand binding domain might grossly disrupt the entire**
**structure that is C-terminal to the DNA binding domain. This could possibly expose part of**
**the protein that is normally folded, causing neomorphic phenotypes. No attempts were**
**made to address or rule out this possibility.**

We appreciate the reviewer's comment regarding the design of the ligand binding domain deletion
(Δ LBD). While engineering this truncation, our goal was to ensure the disruption of ligand-binding
while preserving other structural and functional domains of the receptor. Given the modular
organization of nuclear hormone receptors, including a DNA-binding domain (DBD), hinge region,
and multi-helical ligand-binding domain (LBD) (Mangelsdorf, Thummel et al. 1995, Rastinejad
1998), a complete deletion of the LBD risked destabilizing the protein or eliminating other
important regulatory elements. Therefore, we adopted a more conservative, structure-based
truncation strategy, which entailed removing residues 295 through 422. With this truncation, we
excised the predicted helices 6–10 of the LBD, which are highly conserved and known to
coordinate ligand binding in canonical nuclear receptors (Eeckhoutte, Oxombre et al. 2003,
Rastinejad 2023). We opted to retain helices 1, 3–5 as well as the critically helix 12, which
undergoes conformational changes to mediate coactivator binding (Rastinejad, Huang et al.
2013). By retaining this region, we aimed to preserve co-factor interactions, while abolishing the
receptor's ligand-binding function. The structural visualization of this truncation strategy is now
included in **Fig. 2a** of the revised manuscript, and we have included additional text in the revised
manuscript in **lines 208-213** to further explain our rationalization.

To evaluate the structural integrity of the $\Delta 295-422$ truncation, we performed limited proteolysis
 to assess general stability of the different NHR-49 proteins. By comparing the trypsin digestion
 profiles of four NHR-49 variants: wild-type, S114A, S114D, and $\Delta 295-422$, we observe that the
 wild-type transgenic protein exhibited the greatest resistance to proteolysis, indicating a more
 compact conformation. This is consistent with previous studies showing that ligand binding
 promotes proper folding and structural stability of nuclear hormone receptors (Darimont, Wagner
 et al. 1998). While all three mutant transgenic proteins (S114A, S114D, and $\Delta 295-422$) displayed
 increased sensitivity to proteolysis when compared to wild-type, digestions patterns of the various
 immuno-reactive proteolytic fragments were similar for all transgenic proteins, indicating similar
 folding despite the mutants adopting a more open, labile confirmation (please see **Reviewer**
 **Comments Figure 4**).

**Figure 6** – Trypic digestion pattern of different NHR-49::YFP fusion proteins as determined by immunoblot.

 In parallel, we also
 performed co-
 immunoprecipitation
 mass spectrometry on
 the wild-type, S114A,
 and $\Delta 295-422$.
 Importantly, these
 mutant strains (S114A
 and $\Delta 295-422$) retain
 the ability to interact
 with nuclear localized
 proteins over the
 wildtype strain. These
 proteins include but
 are not limited to the
 nuclear lamin (LMN-1), the spectrin-repeat containing nuclear envelope protein (ANC-1), and the nucleolin homolog (NUCL-1) (please see **Reviewer Comments Figure 5**).

**Figure 7** – Raw Co-IP Mass spectrometry values of nuclear-localized proteins
 interacting with NHR-49 (S114A) and the ligand binding truncation strains.

 Altogether, these data indicate the LBD truncation retains a similar yet more labile confirmation,
 which is still capable of retaining a large portion of its nuclear interaction network.

To test for potential neomorphic activity, we examined both transcriptional and physiological
 phenotypes of $\Delta 295\text{--}422$ in the *nhr-49(nr2041)* mutant background. Under fed conditions, the
 $\Delta 295\text{--}422$ failed to rescue canonical NHR-49 transcriptional targets or physiological traits such
 as cumulative progeny production and lipid storage (displayed in Fig. 2d-f and Extended Data
 Fig. 4e-g of the revised manuscript). However, under fasting conditions, the truncated receptor
 robustly induced canonical transcriptional targets, consistent with a ligand-independent activation
 mechanism, and retained its ability to attenuate upon refeeding (please see Fig. 2g,h in the
 revised manuscript). This indicates that the truncation retains regulated activity in
 response to nutrient stress, despite its impaired function under basal conditions.
 To further explore potential context-specific activation, we assessed the
 ability of $\Delta 295\text{--}422$ to rescue NHR-49-dependent heat shock phenotypes, which
 are unrelated to fasting. Under thermal stress, the truncation was able to rescue
 the *nhr-49(nr2041)* thermotolerance phenotype, performing comparably to its
 wild-type transgenic counterpart (please see Reviewer Comments Figure 6).

Figure 8 – Lifespan post heat shock for 4 hours at day 2 of adulthood at 34°C.

634 In summary, the $\Delta 295\text{--}422$ truncation
 does not exhibit signs of protein misfolding (based on limited proteolysis and interaction network
 dynamics via co-immunoprecipitation/LC-MS/MS) and retains ligand-independent transcriptional
 activity under stress conditions including fasting or heat. However, the NHR-49 LBD mutant did
 not rescue some phenotypes under permissive conditions (transcriptional activity in a fed state,
 lipid deposition assessed by Oil-Red-O staining, and brood counting). These results support the
 interpretation that the truncation does not produce neomorphic gain-of-function effects but rather
 reveals stress-dependent mechanisms of NHR-49 regulation that are independent of ligand
 binding. However, we opted to omit the heat stress rescue data from the revised manuscript to
 maintain focus on lipid metabolism.

 **3. No data was provided to demonstrate the functionality of the wild type NHR-49c::GFP fusion protein.** For example, could the transgene reverse the lifespan extension seen in *nhr-49* mutant animals after 1-day fasting? In the absence of rescue data, it is unclear if the mode of regulation of NHR-49c::GFP, as suggested by the authors, is applicable to endogenous NHR-49.

We believe that this reviewer refers to the intestinal-specific NHR-49c::YFP transgenic strain,
 which we generated for this study versus the all-tissue NHR-49c::GFP used in prior studies by
 several other groups. In the original submission, we included data on brood count, neutral lipid
 staining with Oil-Red-O, relative transcript quantification via RT-qPCR, and western blot of the
 fusion protein. For reference in the revised manuscript, all NHR-49c::YFP transgenic experiments
 were performed in the *nhr-49* mutant background and are now included in the respective figures
 Fig. 2b-h,k; Extended Data Fig. 4d-g,j,k; Fig. 5b,e; and Extended Data Fig. 7a,d,i,j.

The reviewer is correct in noting that we did not include lifespan data for this fusion protein. Having
 characterized that our NHR-49c::YFP overexpression constructs reflects similar
 nucleocytoplasmic dynamics and interactomes as NHR-49c at endogenous levels (Tatge 2025),
 we felt confident embarking upon this aging study as we had already confirmed that this transgene

rescued all of the phenotypes noted in the above paragraph. For the aging analysis, we performed
lifespan analysis on the transgenic animals ectopically expressing this fusion protein within the
animal's intestine. In brief, we observed that intestinal NHR-49c::YFP rescued lifespan deficits
typical of the *nhr-49(nr2041)* mutant and maintained plasticity when fasted and refed. Thus, we
feel confident that this transgene is functional and acting similar to the endogenous form despite
its ectopic overexpression. This new data has now been incorporated into the revised manuscript
in **Extended Data Fig. 4d** with additional text included in the results section at **lines 217-220**.

4. Based on publicly available RNAseq data at Wormbase, the expression levels of NHR-49 target
genes *acs-2* (FPKM=1.3), *fat-7* (FPKM=28.4) and *rab-11.2* (FPKM=0), are very low in young adult
animals. As a reference, the *fat-7* paralog *fat-6* is highly expressed (FPKM=623). **This raised the**
**question on the absolute contribution of ACS-2, FAT-7 and RAB-11.2 proteins to NHR-49**
**dependent, fasting-induced longevity. The expression levels of the three target genes were**
**compared in relative terms (for example Fig. 2e, 2f), rather than absolute terms, thus further**
**hindering the proper assessment of their functional impact.**

We appreciate the reviewer's thoughtful comment and welcome the opportunity to clarify both the
rationale for gene selection and the basis for reporting expression changes in relative terms. Our
study is centered on understanding how NHR-49 regulates transcriptional programs in response
to fasting and refeeding. As such, *acs-2* and *fat-7* were selected not because of their direct or
functional contributions to longevity *per se*, but rather as surrogate markers of NHR-49 activity.
Whereas a paralog such as *fat-6* has been previously published to not be a direct read out of
NHR-49 activity (Van Gilst, Hadjivassiliou et al. 2005). Investigating the functional roles of these
metabolic enzymes in the context of age regulation is outside the scope of this present study. For
this reason, we opted not to conduct knockdown or overexpression experiments for these
individual enzymes but rather focus on larger network-related changes mediated by NHR-49.

As mentioned in response to comment #1, *acs-2* and *fat-7* are well-characterized and seemingly
direct transcriptional targets of NHR-49 as their expression is abolished in *nhr-49(nr2041)* mutants
(Van Gilst, Hadjivassiliou et al. 2005, Taubert, Van Gilst et al. 2006) and recently deposited NHR-
49 ChIP-seq datasets reporter FAT-7 and ACS-2 as the 2nd (adjusted $p = 3.75 \times 10^{-12}$) and 39th
(adjusted $p = 0.00099$) most significant NHR-49 targets, respectively, compared to FAT-6, which
was not significant (adjusted $p = 0.711$) (Gopal, Chaturbedi et al. 2025). Overall, these published
results are consistent with our data that was presented in the original manuscript (now in **Fig. 1g**;
**Fig. 2d,f-h**; **Extended Data Fig. 2c,d** and **Extended Data Fig. 4e** of the revised manuscript).
Despite over 15,000 genes being reported in NHR-49 ChIPseq datasets (Gopal, Chaturbedi et al.
2025), RAB-11.2 was absent from this list. However, it is worth noting that *rab-11.2* does not
appear to be a direct target of NHR-49 and at no point in this manuscript, nor in our original
(Watterson, Tatge et al. 2022) publication, did we make this claim. Yet, *rab-11.2* is a highly
dynamic fasting-responsive gene that we utilized as an *in vivo* fluorescent reporter of transcription
(*rab-11.2p::YFP*) for screening purposes. Hits identified with this reporter were further validated
by testing their effects on the expression of *acs-2* and *fat-7*, confirming their relevance in NHR-49
signaling. We clarified our rationale for these three genes in **lines 317-321** of the revised
manuscript.

In response to the concern about comparing gene expression in relative rather than absolute
terms, RT-qPCR is inherently a comparative method, designed to assess fold-changes in
expression relative to stable internal controls. In our experiments, we used two validated
housekeeping genes (*Y45F10D.4* and *tba-1*) to ensure robust normalization across conditions.
This normalization accounts for technical variation and allows reliable comparison of
transcriptional changes, which are standard practice for RT-qPCR studies.

For clarity in these revisions, we
 have included RPKMs and total raw
 counts for each gene of interest (see
 **Reviewer Comment Figure 7**). In
 brief, in our poly(A)-enriched RNA-
 seq datasets, *acs-2* expression
 increases from 18.60 to 607.55 CPM
 upon fasting (32.6-fold), *fat-7*
 decreases from 18.72 to 1.95 CPM
 (9.6-fold), and *rab-11.2* increases
 from 0.09 to 1.38 CPM (15.3-fold).
 Under *kin-19* RNAi, *acs-2* rises
 from 27.94 to 226.42 CPM (8.1-fold),
 *fat-7* decreases from 18.21 to 1.47
 CPM (12.4-fold), and *rab-11.2*
 increases from 0.01 to 48.16 CPM (4,816-fold).

It is important to note that absolute
 transcript abundance correlates
 poorly with the corresponding
 protein abundance. For this reason,
 we also leverage whole-worm
 proteomics and observe consistent
 trends: ACS-2 protein abundance
 increases from 3.16×10^6 to
 4.63×10^6 (46% increase), while
 FAT-7 decreases from 3.14×10^5
 to 9.92×10^4 (68% decrease)
 under *kin-19* RNAi. Thus, whether
 transcript abundance is considered in
 absolute or relative terms, the
 regulatory patterns are consistent,
 and, more importantly, the
 corresponding changes in protein
 abundance follow the same trends.

We also noted differences in overall
 transcript abundance between our
 RNA-seq data (sequenced in
 2022-2025) and the Wormbase-
 738 referenced ModEncode datasets
 (published in 2010). In addition
 to advancements in sequencing
 technology and analytics over the
 last 15 years, we attribute
 difference in absolute transcript
 levels (between multiple datasets
 generated in our lab and the
 single ModEncode dataset
 reported in Wormbase) to
 differences in library preparation
 methods. Unfortunately, the
 reviewer neglected to provide
 citations on the specific values
 being reported. Thus, it is
 745 difficult to compare our datasets
 when experimental conditions,
 strains, and food sources for the
 reported values are absent.
 Nonetheless, we closely examined
 Wormbase to better understand
 the reviewer's concern. It
 appears as though the reviewer
 cited RiboZero-based datasets
 (e.g., FPKM = 1.3 for *acs-2*)
 with a $n = 1$, which are not
 directly comparable to our
 PolyA enriched analysis. When
 considering aggregated PolyA+
 datasets from WormBase, the
 mean and median FPKM for
acs-2 in adult worms are
 62.69 and 35.63, respectively,
 values more consistent with
 our findings. Similarly, *fat-7*
 shows a median FPKM of 28.4.
 For this reason, we have
 maintained transcript changes
 in relative terms through most
 of the revised manuscript for
 ease of interpretation.

Finally, we respectfully emphasize
 that the functional impact of a
 gene cannot be inferred from its
 absolute transcript expression
 alone. For instance, take into
 753 account Weber's Law: "...
 discrimination thresholds are
 754 proportional to the intensity of
 physical stimulus." (Weber 1996).
 In Weber's work, he investigated
 sensory modalities such as weight
 discrimination, hearing, and
 756 vision, concluding that for a
 perception to occur, the ratio of
 the new stimulus to the
 757 background stimulus, essentially
 a fold-change, was necessary.
 Researchers from Harvard
 Systems biology in 2011
 posited that this too shall be
 a consideration moving forward
 at the level of transcription
 (Goentoro and Kirschner 2009),
 especially in the case of their
 experimentation probing the
 WNT

Figure 9 – Raw RPKM and Total Counts of *acs-2*, *fat-7*, and *rab-11.2* in two poly-A enriched RNA sequencing data sets.

pathway. “In analogy to Weber’s law in sensory physiology, some gene transcription networks
may be tuned to respond to fold-changes, rather than absolute levels of signals, as a way to
reduce the consequences of stochastic, genetic and environmental variation.” (Goentoro and
Kirschner 2009). (Over 160 citations). Thus, under stress conditions such as fasting, fold-changes
in expression can be more biologically meaningful. Even genes with low basal expression can
have substantial cellular effects when significantly up- or down-regulated. Ultimately the functional
importance of transcript abundance and changes depends on overall protein stability and turnover
rates. In our data, all three genes showed large, reproducible fold-changes that reflect meaningful
transcriptional regulation.

While determining the absolute contribution of ACS-2, FAT-7, and RAB-11.2 to longevity is an
important objective for future research, it lies beyond the scope of this study. Our conclusions
instead center on transcriptional regulation, supported by consistent trends observed across
RNA-seq, qPCR, and proteomics datasets, which together underscore the utility of these genes
as reliable functional surrogates for NHR-49 activity.

**5. The quantitation of DHS-3::GFP fluorescence as a measurement of “lipid droplet**
**dynamics” (Line 90) and “lipid droplet levels” (Line 122) was confusing.** The expression of
the DHS-3::GFP fusion protein was under the control of the *dhs-3* promoter, from a multi-copy
transgene that over-expressed DHS-3::GFP. Therefore, the reduction in DHS-3::GFP
fluorescence could be attributed to the fasting-induced repression of the transgene, which might
not be directly related to lipid droplet dynamics or levels at the organelle level.

We thank the reviewer for this comment and agree that clarification is warranted regarding the
interpretation of DHS-3::GFP fluorescence. To first establish a biochemical baseline, we
quantified total lipid levels in whole worms by TLC, which confirmed a fasting-induced decrease
in TAGs and other lipid classes (now included in the revised manuscript in **Fig. 1a**). In fact, we
observe that 24 hours of fasting resulted in a 61% in TAG levels by TLC compared to the 71%
reduction in DHS-3::GFP fluorescence. To highlight this consistency between TAG levels and
DHS-3::GFP, we have included additional text in the revised manuscript at **lines 88-89 and 105-**
**109**. Moreover, lipidomics further revealed reductions in energy-dense species including select
free fatty acids, which recovered upon refeeding, again demonstrating that the worms engage in
metabolically regulated lipid mobilization (in **Fig. 1b** and **Extended Data Fig. 1b** of the revised
manuscript).

On this foundation, we employed DHS-3::GFP to track lipid droplet *dynamics* within intestinal
cells, the major site of lipid storage in *C. elegans*, and have clarified as such at **lines 102-109** in
the revised manuscript. We note that DHS-3::GFP has been the community standard for over a
decade, used in at least 15 independent publications (Zhang, Na et al. 2012, Na, Zhang et al.
2015, Liu, Xu et al. 2018, Zhu, Liu et al. 2018, Cao, Hao et al. 2019, Xie, Zhang et al. 2019, Chen,
Lemieux et al. 2020, Wang, Xia et al. 2021, Zeng, Li et al. 2021, Qin, Wang et al. 2022, Watterson,
Tatge et al. 2022, Xie, Liu et al. 2022, Fu, Zhang et al. 2024, Laranjeira, Berger et al. 2024, Taylor,
Hartman et al. 2024). Given its repeated validation across the field and our complementary TLC
data in **Fig. 1a** of the revised manuscript, we are confident that DHS-3::GFP fluorescence serves
as a robust surrogate for lipid droplet biology and overall TAG levels.

That said, we appreciate the reviewer’s point that DHS-3::GFP is expressed from a multi-copy
 transgene and could, in principle, be influenced by transcriptional regulation. This is entirely
 consistent with fasting biology: *dhs-3* transcripts rise during fasting while DHS-3 protein
 abundance falls as droplets are consumed. In
 our *kin-19* RNAi datasets, where both
 transcriptomics and proteomics were
 available, we indeed observed modest
 transcriptional activation of *dhs-3* coupled
 with mild increase in DHS-3 protein levels
 (Please see **Reviewer comment figure 7**).

Figure 7 – Transcript and proteomic changes upon *kin-19* RNAi of *dhs-3*.

To address this concern more directly, we
 expanded our validation with an alternate lipid
 droplet markers. Both an ectopically
 expressed PLIN-1::mCherry strain and our
 newly generated CRISPR-engineered
 endogenous PLIN-1::mKate2 strain
 recapitulated the fasting-induced fluorescence loss observed with the DHS-3::GFP transgene
 (please see **Reviewer Comment Figure 8**).

Figure 10 – Fed and fasted images of the PLIN-1::mCherry (left), and the endogenous PLIN-1::mKate2 (right).

This convergence across biochemical assays (TLC, lipidomics) and independent genetic reporters (endogenous PLIN-1, overexpressed PLIN-1, prior studies) demonstrates that the reduced DHS-3::GFP signal reflects bona fide lipid droplet depletion/remodeling, not merely transgene repression.

In short, while we agree clarity was needed, we would also note that DHS-3::GFP has satisfied peer review in over 15 previous studies precisely because it does what it claims to

833 do: report lipid droplet levels. With the addition of our biochemical (reported in the revised
 manuscript at **lines 88-92 and 107-108**) and PLIN-1 validation (only for these revisions), we
 believe the case is now stronger than ever. We thank the reviewer for prompting us to refine the
 text and believe that these clarifications resolve the concern.

6. The ATP reporter used (*wuls305[my-3p::Queen-2m]*), for example in Fig. 1d, was only
 expressed in the body wall muscle. **It was unclear if the measurements could be used to**
 **reflect ATP levels in other tissues, such as the intestine, hypodermis, which are also**
 **metabolically active.**

We thank the reviewer for raising this point, a point also raised by Reviewer #1. We agree that
 the *wuls305[myo-3p::Queen-2m]* fluorescence reporter measures relative ATP levels specifically
 in the body wall muscle (Galimov, Pryor et al. 2018). Tissue-specific differences in energy
 homeostasis and ATP production are certainly important, but a detailed comparison across
 tissues was outside the scope of this study. Our rationale was to employ a well-validated tool in
 arguably the most metabolically active tissue during fasting where relative ATP fluctuations can
 be measured and linked to physiological function. As such, previous studies have established the
 body wall muscle as a metabolically demanding tissue with high mitochondrial content, where
 age-dependent changes in mitochondrial function and morphology are readily observed as
 hallmarks of aging (Herndon, Schmeissner et al. 2002,
 Regmi, Rolland et al. 2014).

In brief, the body wall muscle was chosen because (i) bi-
 fluorescent ATP sensors are extensively validated in this
 tissue, (ii) it exhibits clear age-dependent changes in
 mitochondrial function and output, and (iii) ATP
 fluctuations can be correlated with established behavioral
 readouts such as locomotor velocity. In contrast, although
 the pharyngeal muscle and hypodermis are also highly
 active, to our knowledge no ATP sensors have yet been
 reported or validated for these tissues. Nonetheless, we
 did find an intestinal ATP FRET (Förster Resonance
 Energy Transfer) sensor that has recently been
 characterized (Soto, Rivera et al. 2020), but encountered
 substantial interference from the characteristic
 autofluorescence of the gut, leading to only marginal and
 difficult-to-interpret changes (Please see **Reviewer**
 **comment Figure 9**).

Figure 11 - Intestinal FRET sensor testing.

Therefore, in response to the reviewer's concern that our assessment of ATP fluctuations were
 limited to body wall muscle, we sought to strengthen our analysis with complementary approaches
 outside of the available fluorescent reporters. To this end, we went back to the bench and
 performed whole-worm ATP quantification, oxygen consumption rate (OCR) analysis, and
 mitochondrial morphological imaging. In parallel, we mined our -omics datasets to provide a
 broader perspective on the mitochondrial transcriptome upon fasting/refeeding and *kin-19* RNAi.

First, whole-worm ATP quantification using a luciferase-based commercial assay (Invitrogen
 #A22066) was unsuccessful. Despite multiple trials with increasing worm numbers, ATP
 concentrations consistently fell below the assay's dynamic detection range, even relative to
 standards in the picomole range. In contrast, oxygen consumption rate (OCR) analysis revealed
 robust differences in metabolic activity between control and *kin-19* RNAi worms after fasting and
 refeeding (Day 3), confirming significant bioenergetic changes (see Fig. 4g in the revised
 manuscript). Complementing the OCR results, analysis of both endogenous (*COX-4::GFP*) and
 ectopically expressed (*myo-3p::GFP*) mitochondrial markers showed reduced mitochondrial
 content and increased fragmentation with *kin-19* RNAi, consistent with mitochondrial dysfunction
 (see **Extended Data Fig. 6h,i** in the revised manuscript). Lastly, transcriptomics revealed vast

transcriptional dynamics for mitochondrial related genes, many of which overlap between fasting
and *kin-19* RNAi (see **Extended Data Fig. 6d,e** in the revised manuscript).

Given the limited availability of validated ATP measurement tools for *C. elegans*, the technical
caveats of alternative approaches, and the fact that ATP quantification was not the primary focus
of this study, we have opted to present all mitochondrial readout data in the Extended Data
Figures and to de-emphasize its role in the main text. Nonetheless, we have clarified this rationale
and included an additional citation which highlights the body-wall muscle as having higher
mitochondrial membrane potential than other non-muscle tissue as well as being poised to meet
the high energy demand associated with fasting induced foraging (now in the revised manuscript
in **lines 124-127**).

**Reviewer #3** (Remarks to the Author):

In this manuscript entitled “Silencing lipid catabolism determines longevity in response to
intermittent fasting,” the authors investigated how lipid metabolic changes regulate fasting-
induced longevity in *Caenorhabditis elegans*. In this paper, the authors found that silencing lipid
catabolism upon refeeding is crucial for lifespan extension. They identified the nuclear hormone
receptor NHR-49 as a key regulator that promotes lipid catabolism during fasting, and its
inactivation after refeeding, mediated by phosphorylation through casein kinase KIN-19, was
essential for longevity. The study suggests that KIN-19 phosphorylates NHR-49, promoting its
nuclear export and degradation, thereby silencing lipid catabolic genes. These results highlight
that the proper regulation of lipid catabolism is a key determinant of fasting-induced longevity.

Following are my concerns that the authors need to address.

Major comments

1. The authors rely heavily on transgenic expression of NHR-49 variants in an *nhr-49(nr2041)*
background. This introduces significant potential artifacts due to non-physiological
overexpression, and copy number variation. **Given the importance of precise regulation of
NHR-49 activity in this study, and relative easiness of CRISPR genome editing, it is
essential for the authors to generate single-copy CRISPR knock-in strains that express
tagged and mutant forms of NHR-49 from the endogenous locus.**

We appreciate the reviewer’s concern, which was shared with reviewer #2. This too was a
prominent concern for our group and one that we have been working to address long before
submitting this present manuscript. In brief, a number of endogenously tagged NHR-49 worm
strains (both N- and C-terminal) were either generate or obtained from other groups. With these
strains, we performed extensive microscopy and proteomics to investigate how different isoforms
of NHR-49 (at endogenous levels) were differentially distributed throughout the cell. The results
from this study support the observations and interpretations made from worm strains that
ectopically overexpressed NHR-49. We believe that these functional differences between different
endogenous NHR-49 isoforms is an important observation for the field that we did not want to
hide in the supplement of this present story under review. Rather, we wanted to rapidly
disseminate these discoveries to the greater research community and make them the central
focus of their own story. These results were recently peer-reviewed and published in a short
format article (Tatge 2025).

Please find a list of all the endogenously tagged NHR-49 strains used in the (Tatge 2025)
publication plus an additional NHR-49 mutant strains created for these revisions. Due to the gene
structure and presence of alternative translational start sites, N-terminal tagging selectively labels
the long c-isoform of NHR-49, while C-terminal tagging labels all isoforms A-E. Our panel of
endogenous strains are as follows:

Table 2 - Endogenous NHR-49 strains

Strain Name	SunyBiotech Name	Isoform Coverage	Alias	Genotype	Publication Reference/ Original Owner
PMD300	PHX9651	iso C	SfGFP::NHR-49	nhr-49c(syb9651[sfGFP::nhr-49c])	Tatge 2025/ Douglas Lab
PMD342	PHX5674	all	NHR-49::mCherry	nhr-49(syb5674[nhr-49::mCherry])	Tatge 2025/ Douglas Lab
STA07	PHX2863	all	NHR-49::GFPNovo2	nhr-49(syb2863[nhr-49::GFPnovo2])	Tatge 2025/ Taubert Lab
STA08	PHX2927	all	NHR-49::HA	nhr-49(syb2927[nhr-49::HA])	Tatge 2025/ Taubert Lab
PMD320	PHX10203	iso C	TurboID::3xHA::NHR-49	nhr-49c(syb10203 [3xHA::TurboID::nhr-49c])	Tatge 2025/ Douglas Lab
PMD319	PHX10204	all	NHR-49::3xHA::NHR-49	nhr-49(syb10204 [nhr-49::3xHA::TurboID])	Tatge 2025/ Douglas Lab

Using these new transgenic tools, we observed a nucleocytoplasmic distribution in the intestine,
 consistent with the localization of ectopically overexpressed NHR-49c::YFP (this study) and with
 NHR-49::GFP (Watterson, Arneaud et al. 2022, Watterson, Tatge et al. 2022). Specifically, we
 saw this pattern using confocal microscopy of sfGFP::NHR-49 and through LC-MS/MS analysis
 of cytosolic protein enrichment in TurboID::3xHA::NHR-49 proximity labeling experiments. In
 contrast, C-terminally tagged versions (e.g., GFPNovo2, HA, mCherry, and TurboID) strongly
 favored nuclear localization, as shown by both microscopy and LC-MS/MS (Tatge 2025). These
 isoform-specific differences in subcellular distribution help explain the discrepancies observed
 with cDNA-derived overexpression constructs and provide valuable insight into NHR-49 biology
 (Tatge 2025).

Importantly, with this data, we leveraged our endogenously tagged N- and C-terminal TurboID-
 NHR-49 strains and conducted a post-translational modification (PTM) analysis on the resulting
 biotinylated proteins. Out of over 11,000 peptide fragments, 53 PTMs were detected (37
 phosphorylation, 16 ubiquitination). Among the phosphorylated peptides, three were mapped to
 NHR-49. Most notably, we again detected phosphorylation at S114, with 15 PSMs in both the N-
 and C-terminal TurboID strains, ranking it as the 5th most abundant PTM overall. We also
 identified a novel phosphorylation site at S131 PSM of 2, observed only in the C-terminal strain,
 suggesting it is not present solely in the longest isoform C. An additional low-confidence
 phosphorylation was detected between residues 110 to 128, though it could not be precisely
 localized, potentially due to its transient nature (please reference **Extend Data Fig. 4i** in the
 revised manuscript). In addition to the S114 phosphorylation that has now been observed in three
 independent datasets (endogenous, NHR-49::GFP overexpression with fasting, and NHR-
 49::GFP with *kin-19* RNAi), other common interacting proteins (discussed in this manuscript from
 our overexpression datasets) were also identified in this proximity labeling data sets (Tatge 2025),
 such as XPO-1 and KIN-19. While we were able to generate and/or obtain these endogenously
 tagged NHR-49 strains, time and funding constraints during the revision process limited our ability
 to generate, validate and test individual points mutations but we hope to perform these
 experiments in future studies. We have added the new citation (Tatge, 2025) in revised manuscript
 (starting at **line 195**) and have included the relevant data regarding the further validation of XPO-
 1 and KIN-19 as NHR-49 interacting proteins in **Fig. 5c, Extended Data Fig. 5a and Extended**
 **Data Fig. 7b** via proximity labeling.

**2. This study uses RNAi knockdown of *kin-19* as the only approach to study its function.**
 **There is no *kin-19* loss-of-function mutant to validate the RNAi results. In addition, they**
 **did not attempt to test potential gain of function effect of *kin-19* by overexpression.** The
 authors fail to demonstrate that endogenous KIN-19 phosphorylates NHR-49 *in vivo* through
 genetic epistasis or rescue experiments.

We appreciate the reviewer's suggestion to further validate the role of KIN-19 through genetic
 loss- or gain-of-function models. With respect to a gain-of-function model, KIN-19 has previously
 been shown to aggregate when ectopically overexpressed in the worm (David, Ollikainen et al.
 2010, Lechler, Crawford et al. 2017, Huang, Wagner-Valladolid et al. 2019). The documented
 destabilization and assembly of overexpressed KIN-19 into seemingly amorphous aggregates
 throughout the worm was concerning, as we anticipated it could confound experimental results.
 Would phenotypes arising from KIN-19 overexpression reflect increased enzymatic activity or
 proteotoxicity due to aberrant protein aggregation, a process that has the strong potential to seed
 aggregation of other metastable proteins and induce global proteome destabilization? Importantly,
 we do not believe that KIN-19 normally functions, even at older ages, as an aggregate *in vivo*, as
 insertion of an mNeonGreen tag at the endogenous KIN-19 locus produces a reticular subcellular
 pattern that is more indicative of organellar association (please reference **lines 384-387** and
 **Extended Data Fig. 6a** of the revised manuscript).

It is worth noting that we attempted to engineer a loss-of-function mutation in the endogenous
 KIN-19 gene. Despite multiple trials using several different CRISPR guide strategies across
 hundreds of injected animals, we were unable to obtain homozygous worms carrying a
 catalytically inactive single-point mutant (D135A) in KIN-19 (see **lines 382-384** of the revised
 manuscript). Given these technical limitations, we opted to use RNAi, which reproducibly reduced
 *kin-19* transcript levels by 78.9% and protein levels by approximately 95%, as confirmed in our
 datasets (please reference **lines 383-386** and **Extended Data Fig. 5i,j** of the revised manuscript).
 While we acknowledge that complete null alleles provide advantages, we believe that this
 approach achieves a robust and selective reduction in KIN-19 activity sufficient to support our
 conclusions.

With this in mind, we have emphasized in the text why a loss-of-function strain was not available
 and we have added to our extended data the RNAi validation experiments demonstrating the
 reproducible reduction in *kin-19* transcript and protein levels. These data provide evidence that
 our approach effectively diminishes KIN-19 levels, allowing us to interpret phenotypic outcomes
 with confidence while avoiding confounding effects associated with overexpression or incomplete
 CRISPR-based mutagenesis.

**2. This study uses RNAi knockdown of *kin-19* as the only approach to study its function. There is**
 **no *kin-19* loss-of-function mutant to validate the RNAi results. In addition, they did not attempt to**
 **test potential gain of function effect of *kin-19* by overexpression. The authors fail to demonstrate**
 **that endogenous KIN-19 phosphorylates NHR-49 *in vivo* through genetic epistasis or**
 **rescue experiments.**

We appreciate the reviewer's suggestion and agree that demonstrating *in vivo* phosphorylation of
 NHR-49 by KIN-19 would further strengthen our model. We now provide multiple lines of evidence
 that endogenous NHR-49 is phosphorylated (as mentioned in response to comment 1), including
 at sites consistent with KIN-19 activity.

First, we initially identified S114 phosphorylation on NHR-49 in our fasting proteomics post-
 translation modification dataset using an overexpressed NHR-49::GFP strain (mentioned at **lines**
 **250-254** of the revised manuscript).
 After thoroughly screening a myriad
 of kinases that converge on three
 independent datasets, we went
 back to the NHR-49::GFP strain
 and wanted to observe if this
 phosphorylation would accumulate
 upon knock down of *kin-19*. Using a
 SuperSep Phos-Tag gel to validate
 a phosphorylation shift prior to
 submitting samples to proteomics,
 we observe a dramatic mobility shift
 by *kin-19* RNAi indicated that
 inhibition of this kinase results in
 increased NHR-49
 phosphorylation. Moreover, we
 observe this change in the NHR-
 49::GFP strain as well as the
 intestinal specific NHR-49::YFP strain
 generated for this study (please reference **Reviewer Comment Figure 10**).

Figure 10 – Immunodetection of NHR-49::GFP and NHR-49::YFP after resolution by phosTag SDS-PAGE.

Because we confirmed quantitatively through mass spectrometry that this specific S114
 phosphorylation reproducibly increased by 3-fold (please reference **Fig. 5a** of the revised
 manuscript), we did not include this western blot of these ectopically overexpressed NHR-49
 proteins. While our data strongly indicates that KIN-19 modulates NHR-49 phosphorylation *in*
 *vivo*, indirect or compensatory effects cannot be entirely excluded.

To directly probe endogenous NHR-49, we leveraged N- and C-terminal TurboID-tagged NHR-49
 strains. Streptavidin pull-downs followed by LC-MS/MS revealed 53 PTMs across >11,000 peptide
 fragments, including three phosphorylation sites on NHR-49. Notably, S114 was among the most
 abundant PTMs in both TurboID strains, while S131 was observed only in the C-terminal strain,
 suggesting isoform-specific phosphorylation (please reference **Extended Data Fig. 4i** of the
 revised manuscript). Additional residues within amino acids 110–128 (S117, S120), which are
 efficiently phosphorylated *in vitro*, were not highly detected in our endogenous or the
 overexpression data, potentially reflecting the transient nature of these modifications. This is
 consistent with previous reports (Inuzuka, Tseng et al. 2010, Francisco and Virshup 2022, Marzoll,
 Serrano et al. 2022) that casein kinase 1 targets substrates through rapidly propagated, short-
 lived phosphorylation motifs, “Repeats of this CK1 motif are found in many CK1 substrates in the
 pattern pSxx(S/T)xx(S/T)xx(S/T), and the phosphorylation of this upstream S/T allows
 propagation of the signal to multiple downstream serine/threonine residues. This phosphorylation
 of a primed substrate is very rapid.”(Francisco and Virshup 2022). Moreover, the nature of
 phosphatase(s) acting on these different phosphorylated residues is unknown and its feasible that
 high abundance or non-specific activity from other phosphatases mediate dephosphorylation
 within the worm extract despite the addition of phosphatase inhibitors. Overall, these data support
 a model in which primed phosphorylation at S114 facilitates a more extensive cascade at
 neighboring residues, potentially modulating NHR-49’s interaction with chromatin through
 electrostatic repulsion.

Taken together, while direct detection of all KIN-19-dependent phosphorylation events *in vivo*
 remains technically challenging due to their transient nature and low abundance, our integrated

proteomic, genetic, and biochemical approaches demonstrate that endogenous NHR-49 is
phosphorylated at physiologically relevant residues. These findings, in combination with our *in*
*vitro* reconstitution assays, strongly support a model in which KIN-19 phosphorylates NHR-49 to
regulate its activity *in vivo*.

3. While the use of AlphaFold3-predicted iPTM scores is nice for preliminary *in silico* analysis,
these predictions are not substitutes for biochemical validation. **For instance, the interactions**
**between KIN-19 and NHR-49, as well as XPO-1 and NHR-49, are primarily inferred via AI**
**modeling and synthetic peptides, not co-immunoprecipitation of endogenous proteins.**

We thank the reviewer for raising this important point. In our original manuscript, we initially
performed RNAi screens on kinases identified in complex with overexpressed NHR-49::GFP via
co-immunoprecipitation followed by LC-MS/MS analysis (please reference **line 289** and
**Extended Data Fig. 5a** in the revised manuscript). Utilizing a different set of transgenic NHR-
49::YFP overexpressing strains selective to the intestine, we again detect significant binding
between NHR-49 and KIN-19 (data not included in the revised manuscript). Moving forward, we
next sought to determine whether we could still observe an interaction between endogenous
NHR-49 and endogenous KIN-19, to ensure this was not an overexpression artifact. For this, we
utilized proximity labeling experiments with TurboID tags at both the N- and C-terminus of NHR-
49 and confirmed significant interactions between KIN-19 and NHR-49 (please reference **lines**
**291-299** and **Extended Data Fig. 5a** in the revised manuscript). In brief, we identified 41 kinases
in these endogenous labeling experiments, 22 of which were conserved between the 96 found in
the ectopically overexpressed NHR-49::GFP. Overall, this accounts for 3 different transgenic
NHR-49 strains (one ectopically overexpressed and two endogenous) in which we observe
complex formation via both co-immunoprecipitation and proximity labeling with KIN-19 via LC-
MS/MS analysis. It is worth noting that this trend also held true for XPO-1 (please see revised
**Fig. 5c** and **Extended Data Fig. 7b**).

To gain a mechanistic understanding of these interactions, we employed a four-pronged pipeline
to narrow down our candidate kinase. **First**, we cross-referenced multiple independent co-
immunoprecipitation/proximity labeling datasets to identify overlapping kinases. **Second**, we
screened these kinases using an *in vivo* fasting-responsive fluorescence reporter (*rab-*
*11.2p::YFP*), along with other well-known metabolic kinases to ensure broad coverage. **Third**, we
evaluated their predicted ability to bind NHR-49 *in silico*, both with and without the S114
phosphorylation site. **Finally**, we performed *in vitro* P-32 kinase assays to test for direct
phosphorylation, as kinase–substrate interactions are often transient and may escape detection
by co-immunoprecipitation or proximity labeling. To simplify the system, we used recombinant
proteins and NHR-49 peptides, enabling us to directly test whether the casein kinase could
transfer the γ -phosphate from ATP onto predicted residues in NHR-49, and to assess where XPO-
1 might bind NHR-49. While any single approach might be considered inconclusive, the
convergence of complementary genetic, proteomic, *in silico*, and *in vitro* strategies provides
compelling evidence that KIN-19 phosphorylates NHR-49.

3. ...**In addition, *in vitro* kinase assays failed to show phosphorylation of unprimed NHR-**
**49 by recombinant KIN-19, but this is central to the model proposed.** Moreover, the AI-
predicted interaction improvement upon S114 phosphorylation (iPTM increase to 0.69) is
suggestive but does not establish a functional or physical interaction. Overall, the conclusions
about phosphorylation-driven attenuation and degradation of NHR-49 remain highly speculative
in the absence of stronger biochemical evidence and need to be experimentally validated with
rigorous biochemical experiments.

We appreciate this important point and agree that clarifying the kinase activity of KIN-19 toward
unprimed NHR-49 is critical to the proposed model. In the revised manuscript, we now include
data from the preliminary *in vitro* kinase assays using recombinant CSNK1A1 on our unprimed
NHR-49 peptide (please reference **Fig. 3d** and **Extended Data Fig. 5d** in the revised manuscript).
While initial results did not include these controls because these were utilized to define the linear
range of detection for the kinase phosphorylation assay, our expanded data reveals that
CSNK1A1 exhibits basal phosphorylation activity on the unprimed NHR-49 peptide ($k_m = 106.6$;
$V_{max} = 10.98$). However, phosphorylation is markedly enhanced when S114 is pre-
phosphorylated ($k_m = 285.7$; $V_{max} = 68.63$) (please reference **lines 354-360** and **Fig. 3d** in the
revised manuscript). Overall, this is a 2.68-fold increase in K_m and a 6.25-fold increase in the
V_{max} of the of the pre-phosphorylated serine over the unprimed site.

This observation is consistent with previously reported mechanisms of CK1 family kinases, which
are capable of phosphorylating non-primed substrates but do so far more efficiently when an
upstream priming phosphorylation is present (Inuzuka, Tseng et al. 2010, Francisco and Virshup
2022, Marzoll, Serrano et al. 2022). Specifically, CK1s are known to act on substrates via
sequential phosphorylation of serine/threonine residues following a canonical pSxx(S/T) motif.
This priming-dependent enhancement aligns with our *in vivo* findings: S114 phosphorylation is
consistently detectable, whereas downstream residues S117 and S120 remain elusive, likely due
to the transient and rapid nature of their phosphorylation following S114 priming.

We have now incorporated these findings, along with appropriate controls and citations, into the
revised manuscript at **lines 354-368** and in **Fig. 3b,d** and **Extended Data Fig. 5d**, which further
supports our proposed model of a priming-dependent phosphorylation cascade mediated by KIN-
19.

4. The authors admit that the priming phosphorylation at S114, essential for the proposed model,
remains unidentified. They attempted to test MPK-1 but showed that it did not phosphorylate the
relevant site. **They did not pursue alternative kinases (e.g., PKA, AMPK) experimentally,**
**leaving a significant mechanistic gap in their model.**

We appreciate the reviewer's concern regarding the identity of the kinase responsible for priming
phosphorylation at S114. We agree that identifying the upstream kinase would further solidify our
proposed model. However, based on extensive discussions with kinase expert Dr. Tagliabracci, a
senior author on the paper, and insights from the kinase literature, it is important to note that
kinases are highly selective for well-characterized consensus motifs. The number of kinases
capable of phosphorylating a serine residue in the absence of a strong match to their consensus
is limited, and large-scale recombinant screening is impractical due to cost, availability, and
technical variability. Given that 41 kinases were identified in our endogenous NHR-49 TurboID
dataset, and 107 in our overexpression datasets, we developed a prioritized experimental pipeline
to focus our efforts (as mentioned in response to comment 3) and define a meaningful, biologically
relevant mechanism.

Before receiving these reviewer comments, we were already in the process of characterizing
endogenous NHR-49 strains for the (Tatge 2025) study. From this small study, we observed that
our endogenously tagged NHR-49 strains were indeed acting very similar to our overexpression
strains. We cross-referenced our endogenous NHR-49 proximity labeling datasets with co-
immunoprecipitations of NHR-49 overexpression datasets and found that 23 out of the 41 kinases
identified in our endogenous proximity labeling experiments overlapped – 56.1% – providing
confidence that the interactors observed in our overexpression system are biologically relevant
and consistent with endogenous NHR-49 behavior (Tatge 2025). Next, we screened these
candidates using the *rab-11.2* transcriptional reporter, which is sensitive to changes in fasting
activity. In parallel, we performed *in silico* AI-based protein::protein predictions. Finally, based on

these orthogonal filters, we tested the top candidates using *in vitro* kinase assays with
recombinant proteins.

Although MPK-1 was initially prioritized based on prior literature and consensus sequence, it failed
to activate our fasting-responsive fluorescence reporter upon RNAi knockdown. Moreover, there
was no significant *in silico* interaction, and ERK1 (MPK-1 homolog) did not phosphorylate S114
in our *in vitro* reconstitution system. Per the reviewer's suggestion, we subsequently purchased
and tested the ability of PKA to phosphorylate NHR-49. This kinase was included in our screen
because of its reported interactions with the human ortholog, HNF4 (Viollet, Kahn et al. 1997).
We observe that RNAi knockdown of KIN-2 (protein kinase cAMP-dependent type I regulatory
subunit beta homolog) modestly activated our fasting-responsive reporter in the initial screen, but
lacked *in silico* confirmation. However, the consensus motif for PKA is RRXS/TY or RRXpS/pT
(with X being any residue, and Y being a hydrophobic residue) and this does not align with the
local sequence surrounding S114. Importantly, our recombinant kinase assay showed no
significant phosphorylation at that site (please reference **lines 375-378** of the revised manuscript
and **Extended Data Fig. 5e-h**). Additionally, we also tested PKM2 (PYK-1), one of the top kinases
identified by PSM count in the endogenous data set, but it too failed to phosphorylate the relevant
region (data not shown). While we acknowledge that the priming kinase for S114 remains
unidentified, the combination of our proteomics, genetic, and *in vitro* data allowed us to construct
a rational, stepwise experimental framework for narrowing candidates in a highly selective kinase
landscape. We believe that this systematic approach reflects the biological and technical
constraints of identifying transient phosphorylation events and lays a strong foundation for future
discovery for the S114 kinase.

Minor comments

**1. It would be valuable to test additional dietary restriction paradigms** using the kin-19 and
nhr-49 genetic models to assess whether the observed effects are specific to intermittent fasting
or more broadly applicable.

We appreciate the reviewer's suggestion and agree that comparing different nutrient restriction
paradigms is important for understanding the specificity and breadth of the observed effects.
However, our study was designed to focus specifically on short-term, acute fasting rather than
chronic dietary restriction (DR), which is mechanistically distinct. Acute fasting has been shown
in recent studies, including work from our collaborators (Acosta-Rodríguez, Rijo-Ferreira et al.
2022) to elicit more pronounced effects on mouse lifespan and metabolic remodeling than
continuous lifelong DR.

To address the reviewer's point, we did explore an alternate fasting paradigm involving a 48-hour
fast, which has been previously reported to extend lifespan (Uno, Honjoh et al. 2013). However,
our results from this fasting paradigm raised concerns. In addition to high censorship rates
resulting from bag-of-worm phenotypes and vulval rupture, we severely impaired lipid availability.
In brief, we performed thin-layer chromatography (TLC), which revealed that 24 hours was
sufficient to deplete triglyceride stores (TAG) by 61%, while preserving minimal lipid reserves in
the form of free fatty acids and cholesterol. However, the 48-hour fast drastically depleted the
neutral lipid stores (TAGs reduced by 91%) and exhausted free fatty acids and cholesterol. For
this study, we sought to limit the availability of neutral storage lipids in TAGs rather than inducing
membrane stress resulting from cholesterol exhaustion. Overall, this 24-hour fasting window
offered a metabolic "sweet spot" that triggers nutrient stress while preserving the capacity for
meaningful refeeding responses. We agree this distinction is important and have clarified our
rationale at the beginning of the results section at **lines 85-92** and have included the TLC
experiment in **Fig. 1a** of the revised manuscript.

2. In Extended Data Figure 1.i, the authors only presented downregulated KEGG pathways
 under IF conditions. **Please provide the list of upregulated KEGG pathways to allow a more**
 **comprehensive interpretation of the transcriptomic changes induced by intermittent**
 **fasting.**

We appreciate the reviewer's suggestion and agree that including the upregulated KEGG
 pathways will provide a more complete view of the transcriptomic response to fasting. We have
 now added the list of significantly upregulated KEGG pathways to **Extended Data Fig. 2a,b** and
 updated the accompanying figure legend.

3. **Please add a brief explanation for why the authors utilized sodium azide as a negative**
 **control in the ATP sensor assay.** This will help improve clarity for readers.

We thank the reviewer for this request. Sodium azide was chosen as a negative control because
 it is a potent and irreversible inhibitor of mitochondrial complex IV. Unlike a number of other
 reversible mitochondrial inhibitors such as oligomycin or antimycin A, this compound is readily
 absorbed by worms, which rapidly abolishes animal motility and overall mitochondrial respiration.
 We have now clarified this point in the methods section of the revised manuscript (**Line 815**).

4. **Please consider adding a brief discussion about how intermittent fasting modulates**
 **KIN-19 expression or activity.** It could provide insights into upstream regulatory mechanisms
 that remain unexplored in this study.

We appreciate this insightful suggestion. We previously investigated this question by generating
 an endogenously tagged KIN-19::mNeonGreen strain to monitor dynamics of endogenous KIN-
 19 during transient fasting. Preliminary data suggest that KIN-19 protein levels decrease upon
 fasting, consistent with potential transcriptional and post-translational regulation. We confirmed

that *kin-19* transcript
 abundance remains
 unchanged in *nhr-*
 *49(nr2041)* mutants,
 ruling out the
 potential for
 autoregulatory
 feedback
 mechanisms (please
 see **Reviewer**
 **Comment Figure**
 **11**). However, due to
 the absence of
 definitive
 mechanistic insight,
 we chose not to
 include these data in
 the main manuscript.
 That said, this

remains an important direction for future work. We agree that leveraging our newly generated,
 endogenously tagged KIN-19::TurboID strain in combination with fasting paradigms would be a
 powerful approach to identify upstream regulators, including kinases or E3 ligases that may target
 KIN-19 degradation. In response to the reviewer's suggestion, we have added a brief discussion
 of these considerations to the revised manuscript (please reference **lines 609-612**).

Figure 12 – from left to right: (left) Flow cytometry of KIN-19::mNeon endogenously-tagged strain, *kin-19* transcriptomics for (middle) fed/fasted/refed and (right) *rab-11.1* RNAi versus Empty Vector (EV) in N2 or *nhr-49(nr2041)* strain.

**5. It will be better to enhance the resolution of Figures 2c.**

We thank the reviewer for this suggestion. We have replaced the original images with higher
resolution micrographs in **Fig. 2b** and **Extended Data Fig. 4k** of the revised manuscript.

Citations:

- Acosta-Rodríguez, V., F. Rijo-Ferreira, M. Izumo, P. Xu, M. Wight-Carter, C. B. Green and J. S. Takahashi
 (2022). "Circadian alignment of early onset caloric restriction promotes longevity in male C57BL/6J
 mice." Science **376**(6598): 1192-1202.
- Ashrafi, K., F. Y. Chang, J. L. Watts, A. G. Fraser, R. S. Kamath, J. Ahringer and G. Ruvkun (2003). "Genome-
 wide RNAi analysis of *Caenorhabditis elegans* fat regulatory genes." Nature **421**(6920): 268-272.
- Cao, Z., Y. Hao, C. W. Fung, Y. Y. Lee, P. Wang, X. Li, K. Xie, W. J. Lam, Y. Qiu, B. Z. Tang, G. Shui, P. Liu, J. Qu,
 B.-H. Kang and H. Y. Mak (2019). "Dietary fatty acids promote lipid droplet diversity through seipin
 enrichment in an ER subdomain." Nature Communications **10**(1): 2902.
- Chen, W.-W., G. A. Lemieux, C. H. Camp, T.-C. Chang, K. Ashrafi and M. T. Cicerone (2020). "Spectroscopic
 coherent Raman imaging of *Caenorhabditis elegans* reveals lipid particle diversity." Nature Chemical
 Biology **16**(10): 1087-1095.
- Darimont, B. D., R. L. Wagner, J. W. Apriletti, M. R. Stallcup, P. J. Kushner, J. D. Baxter, R. J. Fletterick and K.
 R. Yamamoto (1998). "Structure and specificity of nuclear receptor-coactivator interactions." Genes Dev
 **12**(21): 3343-3356.
- David, D. C., N. Ollikainen, J. C. Trinidad, M. P. Cary, A. L. Burlingame and C. Kenyon (2010). "Widespread
 Protein Aggregation as an Inherent Part of Aging in *C. elegans*." PLoS Biology **8**(8): e1000450.
- Doering, K. R. S., G. Ermakova and S. Taubert (2023). "Nuclear hormone receptor NHR-49 is an essential
 regulator of stress resilience and healthy aging in *Caenorhabditis elegans*." Front Physiol **14**: 1241591.
- Eeckhoutte, J., B. Oxombre, P. Formstecher, P. Lefebvre and B. Laine (2003). "Critical role of charged
 residues in helix 7 of the ligand binding domain in Hepatocyte Nuclear Factor 4alpha dimerisation and
 transcriptional activity." Nucleic Acids Res **31**(22): 6640-6650.
- Francisco, J. C. and D. M. Virshup (2022). "Casein Kinase 1 and Human Disease: Insights From the
 Circadian Phosphoswitch." Frontiers in Molecular Biosciences **Volume 9 - 2022**.
- Frankino, P. A., T. F. Siddiqi, T. Bolas, R. Bar-Ziv, H. K. Gildea, H. Zhang, R. Higuchi-Sanabria and A. Dillin
 (2022). "SKN-1 regulates stress resistance downstream of amino catabolism pathways." iScience **25**(7):
 104571.
- Fu, L., J. Zhang, Y. Wang, H. Wu, X. Xu, C. Li, J. Li, J. Liu, H. Wang, X. Jiang, Z. Li, Y. He, P. Liu, Y. Wu, X. Zou
 and B. Liang (2024). "LET-767 determines lipid droplet protein targeting and lipid homeostasis." Journal
 of Cell Biology **223**(6): e202311024.
- Galimov, E. R., R. E. Pryor, S. E. Poole, A. Benedetto, Z. Pincus and D. Gems (2018). "Coupling of Rigor
 Mortis and Intestinal Necrosis during *C. elegans* Organismal Death." Cell Rep **22**(10): 2730-2741.
- Ge, T., D. G. Brickner, K. Zehr, D. J. VanBelzen, W. Zhang, C. Caffalette, G. C. Moeller, S. Ungerleider, N.
 Marcou, A. Jacob, V. Q. Nguyen, B. Chait, M. P. Rout and J. H. Brickner (2025). "Exportin-1 functions as an
 adaptor for transcription factor-mediated docking of chromatin at the nuclear pore complex." Molecular
 Cell **85**(6): 1101-1116.e1108.
- Goentoro, L. and M. W. Kirschner (2009). "Evidence that fold-change, and not absolute level, of beta-
 catenin dictates Wnt signaling." Mol Cell **36**(5): 872-884.
- Goh, G. Y., K. L. Martelli, K. S. Parhar, A. W. Kwong, M. A. Wong, A. Mah, N. S. Hou and S. Taubert (2014).
 "The conserved Mediator subunit MDT-15 is required for oxidative stress responses in *Caenorhabditis
 elegans*." Aging Cell **13**(1): 70-79.
- Goh, G. Y. S., A. Beigi, J. Yan, K. R. S. Doering and S. Taubert (2023). "Mediator subunit MDT-15 promotes
 expression of propionic acid breakdown genes to prevent embryonic lethality in *Caenorhabditis
 elegans*." G3 (Bethesda) **13**(6).
- Gopal, S., A. Chaturbedi, T. Ramachandrupa, J. OuYang, R. Rodell and S. S. Lee (2025). "Nuclear Hormone
 Receptor NHR-49/HNF4alpha Couples Fertility Regulation to Resource Allocation and Longevity in *C.
 elegans*." bioRxiv.
- Herndon, L. A., P. J. Schmeissner, J. M. Dudaronek, P. A. Brown, K. M. Listner, Y. Sakano, M. C. Paupard, D.
 H. Hall and M. Driscoll (2002). "Stochastic and genetic factors influence tissue-specific decline in ageing
 *C. elegans*." Nature **419**(6909): 808-814.
- Horikawa, M., T. Nomura, T. Hashimoto and K. Sakamoto (2008). "Elongation and Desaturation of Fatty
 Acids are Critical in Growth, Lipid Metabolism and Ontogeny of *Caenorhabditis elegans*." The Journal of
 Biochemistry **144**(2): 149-158.

Huang, C., S. Wagner-Valladolid, A. D. Stephens, R. Jung, C. Poudel, T. Sinnige, M. C. Lechler, N. Schlörit,
 1309 M. Lu, R. F. Laine, C. H. Michel, M. Vendruscolo, C. F. Kaminski, G. S. Kaminski Schierle and D. C. David
 (2019). "Intrinsically aggregation-prone proteins form amyloid-like aggregates and contribute to tissue
 aging in *Caenorhabditis elegans*." *eLife* **8**: e43059.

Inuzuka, H., A. Tseng, D. Gao, B. Zhai, Q. Zhang, S. Shaik, L. Wan, X. L. Ang, C. Mock, H. Yin, J. M.
 Stommel, S. Gygi, G. Lahav, J. Asara, Z. X. Xiao, W. G. Kaelin, Jr., J. W. Harper and W. Wei (2010).
 "Phosphorylation by casein kinase I promotes the turnover of the Mdm2 oncoprotein via the SCF(beta-
 TRCP) ubiquitin ligase." *Cancer Cell* **18**(2): 147-159.

Kaeberlein, T. L., E. D. Smith, M. Tsuchiya, K. L. Welton, J. H. Thomas, S. Fields, B. K. Kennedy and M.
 Kaeberlein (2006). "Lifespan extension in *Caenorhabditis elegans* by complete removal of food." *Aging*
 *Cell* **5**(6): 487-494.

Laranjeira, A. C., S. Berger, T. Kohlbrenner, N. R. Greter and A. Hajnal (2024). "Nutritional vitamin B12
 regulates RAS/MAPK-mediated cell fate decisions through one-carbon metabolism." *Nature*
 *Communications* **15**(1): 8178.

Lechler, M. C., E. D. Crawford, N. Groh, K. Widmaier, R. Jung, J. Kirstein, J. C. Trinidad, A. L. Burlingame
 and D. C. David (2017). "Reduced Insulin/IGF-1 Signaling Restores the Dynamic Properties of Key Stress
 Granule Proteins during Aging." *Cell Reports* **18**(2): 454-467.

Lee, K., G. Y. Goh, M. A. Wong, T. L. Klassen and S. Taubert (2016). "Gain-of-Function Alleles in
 *Caenorhabditis elegans* Nuclear Hormone Receptor nhr-49 Are Functionally Distinct." *PLoS One* **11**(9):
 e0162708.

Lemieux, G. A., J. Liu, N. Mayer, R. J. Bainton, K. Ashrafi and Z. Werb (2011). "A whole-organism screen
 identifies new regulators of fat storage." *Nature Chemical Biology* **7**(4): 206-213.

Liu, Y., S. Xu, C. Zhang, X. Zhu, M. A. Hammad, X. Zhang, M. Christian, H. Zhang and P. Liu (2018).
 "Hydroxysteroid dehydrogenase family proteins on lipid droplets through bacteria, *C. elegans*, and
 mammals." *Biochimica et Biophysica Acta (BBA) - Molecular and Cell Biology of Lipids* **1863**(8): 881-894.

Mangelsdorf, D. J., C. Thummel, M. Beato, P. Herrlich, G. Schütz, K. Umesono, B. Blumberg, P. Kastner, M.
 Mark, P. Chambon and R. M. Evans (1995). "The nuclear receptor superfamily: the second decade." *Cell*
 **83**(6): 835-839.

Mao, K., P. Breen and G. Ruvkun (2022). "The *Caenorhabditis elegans* ARIP-4 DNA helicase couples
 mitochondrial surveillance to immune, detoxification, and antiviral pathways." *Proc Natl Acad Sci U S A*
 **119**(49): e2215966119.

Marzoll, D., F. E. Serrano, A. Shostak, C. Schunke, A. C. R. Diernfellner and M. Brunner (2022). "Casein
 kinase 1 and disordered clock proteins form functionally equivalent, phospho-based circadian modules
 in fungi and mammals." *Proc Natl Acad Sci U S A* **119**(9).

Na, H., P. Zhang, Y. Chen, X. Zhu, Y. Liu, Y. Liu, K. Xie, N. Xu, F. Yang, Y. Yu, S. Cichello, H. Y. Mak, M. C.
 Wang, H. Zhang and P. Liu (2015). "Identification of lipid droplet structure-like/resident proteins in
 *Caenorhabditis elegans*." *Biochimica et Biophysica Acta (BBA) - Molecular Cell Research* **1853**(10, Part A):
 2481-2491.

Pathare, P. P., A. Lin, K. E. Bornfeldt, S. Taubert and M. R. Van Gilst (2012). "Coordinate regulation of lipid
 metabolism by novel nuclear receptor partnerships." *PLoS Genet* **8**(4): e1002645.

Pender, C. L., J. G. Dishart, H. K. Gildea, K. M. Nauta, E. M. Page, T. F. Siddiqi, S. S. Cheung, L. Joe, N. O.
 Burton and A. Dillin (2025). "Perception of a pathogenic signature initiates intergenerational protection."
 *Cell* **188**(3): 594-605.e510.

Pukkila-Worley, R., R. L. Feinbaum, D. L. McEwan, A. L. Conery and F. M. Ausubel (2014). "The
 evolutionarily conserved mediator subunit MDT-15/MED15 links protective innate immune responses
 and xenobiotic detoxification." *PLoS Pathog* **10**(5): e1004143.

Qin, S., Y. Wang, L. Li, J. Liu, C. Xiao, D. Duan, W. Hao, C. Qin, J. Chen, L. Yao, R. Zhang, J. You, J.-S. Zheng,
 E. Shen and L. Wu (2022). "Early-life vitamin B12 orchestrates lipid peroxidation to ensure reproductive
 success via SBP-1/SREBP1 in *Caenorhabditis elegans*." *Cell Reports* **40**(12): 111381.

Rastinejad, F. (1998). Structure and Function of the Steroid and Nuclear Receptor DNA Binding Domain.
 *Molecular Biology of Steroid and Nuclear Hormone Receptors*. L. P. Freedman. Boston, MA, Birkhäuser
 Boston: 105-131.

Rastinejad, F. (2023). "The protein architecture and allosteric landscape of HNF4α." *Front Endocrinol*
 *(Lausanne)* **14**: 1219092.

- Rastinejad, F., P. Huang, V. Chandra and S. Khorasanizadeh (2013). "Understanding nuclear receptor form
and function using structural biology." *J Mol Endocrinol* **51**(3): T1-t21.
- Regmi, S. G., S. G. Rolland and B. Conradt (2014). "Age-dependent changes in mitochondrial morphology
and volume are not predictors of lifespan." *Aging (Albany NY)* **6**(2): 118-130.
- Ryu, M.-H., H. S. Sohn, Y. R. Heo, N. Moustaid-Moussa and Y.-S. Cha (2005). "Differential regulation of
hepatic gene expression by starvation versus refeeding following a high-sucrose or high-fat diet."
*Nutrition* **21**(4): 543-552.
- Shomer, N., A. Z. Kadhim, J. M. Grants, X. Cheng, D. Alhusari, F. Bhanshali, A. F. Poon, M. Y. Y. Lee, A.
Muhuri, J. I. Park, J. Shih, D. Lee, S. V. Lee, F. C. Lynn and S. Taubert (2019). "Mediator subunit MDT-
15/MED15 and Nuclear Receptor HIZR-1/HNF4 cooperate to regulate toxic metal stress responses in
*Caenorhabditis elegans*." *PLoS Genet* **15**(12): e1008508.
- Soto, J., M. Rivera, G. Broitman-Maduro and M. F. Maduro (2020). "Expression of a FRET-based ATP
Biosensor in the *C. elegans* Intestine." *MicroPubl Biol* **2020**.
- Tatge, L., Douglas PM (2025). "Isoform differences drive functional diversity of NHR-49." *MicroPubl Biol*.
- Taubert, S., M. R. Van Gilst, M. Hansen and K. R. Yamamoto (2006). "A Mediator subunit, MDT-15,
integrates regulation of fatty acid metabolism by NHR-49-dependent and -independent pathways in *C.*
*elegans*." *Genes Dev* **20**(9): 1137-1149.
- Taylor, S. K. B., J. H. Hartman and B. P. Gupta (2024). "The neurotrophic factor MANF regulates autophagy
and lysosome function to promote proteostasis in *Caenorhabditis elegans*." *Proceedings of the*
*National Academy of Sciences* **121**(43): e2403906121.
- Uno, M., S. Honjoh, M. Matsuda, H. Hoshikawa, S. Kishimoto, T. Yamamoto, M. Ebisuya, T. Yamamoto, K.
Matsumoto and E. Nishida (2013). "A Fasting-Responsive Signaling Pathway that Extends Life Span in
*C. elegans*." *Cell Reports* **3**(1): 79-91.
- Van Gilst, M. R., H. Hadjivassiliou, A. Jolly and K. R. Yamamoto (2005). "Nuclear hormone receptor NHR-
49 controls fat consumption and fatty acid composition in *C. elegans*." *PLoS Biol* **3**(2): e53.
- Van Gilst, M. R., H. Hadjivassiliou and K. R. Yamamoto (2005). "A *Caenorhabditis elegans* nutrient
response system partially dependent on nuclear receptor NHR-49." *Proceedings of the National*
*Academy of Sciences* **102**(38): 13496-13501.
- Viollet, B., A. Kahn and M. Raymondjean (1997). "Protein Kinase A-Dependent Phosphorylation
Modulates DNA-Binding Activity of Hepatocyte Nuclear Factor 4." *Molecular and Cellular Biology* **17**(8):
4208-4219.
- Vozdek, R., Y. Long and D. K. Ma (2018). "The receptor tyrosine kinase HIR-1 coordinates HIF-
independent responses to hypoxia and extracellular matrix injury." *Sci Signal* **11**(550).
- Wang, C., C. Xia, Y. Zhu and H. Zhang (2021). "Innovative fluorescent probes for in vivo visualization of
biomolecules in living *Caenorhabditis elegans*." *Cytometry Part A* **99**(6): 560-574.
- Watterson, A., S. L. B. Arneaud, N. Wajahat, J. M. Wall, L. Tatge, S. T. Beheshti, M. Mihelakis, N. Y.
Cheatwood, J. McClendon, A. Ghorashi, I. Dehghan, C. D. Corley, J. G. McDonald and P. M. Douglas
(2022). "Loss of heat shock factor initiates intracellular lipid surveillance by actin destabilization." *Cell*
*Reports* **41**(3): 111493.
- Watterson, A., L. Tatge, N. Wajahat, S. L. B. Arneaud, R. Solano Fonseca, S. T. Beheshti, P. Metang, M.
Mihelakis, K. R. Zuurbier, C. D. Corley, I. Dehghan, J. G. McDonald and P. M. Douglas (2022). "Intracellular
lipid surveillance by small G protein geranylgeranylation." *Nature* **605**(7911): 736-740.
- Weber, E. H. (1996). *EH Weber on the tactile senses*, Psychology Press.
- Xie, K., Y. Liu, X. Li, H. Zhang, S. Zhang, H. Y. Mak and P. Liu (2022). "Dietary *S. maltophilia* induces
supersized lipid droplets by enhancing lipogenesis and ER-LD contacts in *C. elegans*." *Gut Microbes* **14**(1):
2013762.
- Xie, K., P. Zhang, H. Na, Y. Liu, H. Zhang and P. Liu (2019). "MDT-28/PLIN-1 mediates lipid droplet-
microtubule interaction via DLC-1 in *Caenorhabditis elegans*." *Scientific Reports* **9**(1): 14902.
- Zeng, L., X. Li, C. B. Preusch, G. J. He, N. Xu, T. H. Cheung, J. Qu and H. Y. Mak (2021). "Nuclear receptors
NHR-49 and NHR-79 promote peroxisome proliferation to compensate for aldehyde dehydrogenase
deficiency in *C. elegans*." *PLoS Genet* **17**(7): e1009635.
- Zhang, J., R. Bakheet, R. S. Parhar, C.-H. Huang, M. M. Hussain, X. Pan, S. S. Siddiqui and S. Hashmi
(2011). "Regulation of Fat Storage and Reproduction by Krüppel-Like Transcription Factor KLF3 and Fat-
Associated Genes in *Caenorhabditis elegans*." *Journal of Molecular Biology* **411**(3): 537-553.

Zhang, P., M. Judy, S. J. Lee and C. Kenyon (2013). "Direct and indirect gene regulation by a life-extending
FOXO protein in *C. elegans*: roles for GATA factors and lipid gene regulators." Cell Metab **17**(1): 85-100.
Zhang, P., H. Na, Z. Liu, S. Zhang, P. Xue, Y. Chen, J. Pu, G. Peng, X. Huang, F. Yang, Z. Xie, T. Xu, P. Xu, G.
Ou, S. O. Zhang and P. Liu (2012). "Proteomic study and marker protein identification of *Caenorhabditis*
*elegans* lipid droplets." Mol Cell Proteomics **11**(8): 317-328.
Zhu, X., Y. Liu, H. Zhang and P. Liu (2018). "Whole-genome RNAi screen identifies methylation-related
genes influencing lipid metabolism in *Caenorhabditis elegans*." Journal of Genetics and Genomics **45**(5):
259-272.

Response Line 249: One exception is a mutational screen that identified three gain-of-function mutations within or near the ligand-binding pocket, a study that we cite in both the original and revised manuscript, which was pivotal in guiding our engineering of the ligand-binding domain truncation strain (Lee, Goh et al. 2016).

The gain of function nhr-49 mutants were originally identified by the Pilon lab (PMID: 24068966). Please amend citation in the manuscript.

We appreciate the reviewer's valuable guidance regarding the citation of the *nhr-49* gain-of-function alleles in the manuscript. We have amended the manuscript to include citations for both the original discovery (PMID:24068966) and the **Lee et al. (2016)** publication, which examined the downstream expression of direct NHR-49 targets associated with these specific mutants.

—

Response Line 495: Additionally, we performed thin layer chromatography (TLC), which revealed that 24 hours was sufficient to deplete triglyceride stores (TAG), while preserving minimal lipid reserves in the form of free fatty acids and cholesterol. Unlike the 48-hour fast, which not only exhausted TAGs, but also available free fatty acids and cholesterol.

No details were provided for the method used to perform the TLC analysis (Fig. 1a). Depending on the solvent system, additional lipid species should be detected by TLC (PMID: 9527856). Therefore, it is unclear if the entire TLC plate is displayed. For example, cholesterol esters are missing from the TLC analysis although they are well known to be stored in lipid droplets.

As we were consolidating the new data and text for the initial revision process, we neglected to include a methods section for the TLC and we greatly appreciate the reviewer catching this oversight. We have now included a detailed description of the lipid extraction and thin layer chromatography (TLC) procedure within the methods section of the revised manuscript at lines 923-950. We have also included a citation for the TLC method that we adapt for *C. elegans* (PMID:14861228).

Regarding the second part of this comment, the entire TLC plate from Fig. 1A can be found in the raw data section excel file associated with this manuscript and we have provided the same image below for ease of reference. In the top left of the image showing dilutions of the lipid standard, we do not detect a signal out of whole worm extracts that correspond to the sterol ester standard. This same solvent system typically detects a lipid signal migrating similar to the same sterol ester standards when we process human cells. Our inability to observe sterol esters by TLC in whole worm extracts is consistent with another worm report from the Watts lab (PMID: 26121959) in

which they conclude that “unlike other organisms, *C. elegans* lipid droplets contain very little cholesterol or sterol esters”. It is also worth noting that the Watt’s lab detected trace amounts of sterol esters from isolates worm lipid droplets, while our TLC analysis was performed on whole worm extracts. It remains unclear why *C. elegans* possess such low levels of sterol lipids but may be attributed to their sterol lipid auxotrophic nature. For this reason, worms require cholesterol in their media to ensure worm reproduction.

Response Line 573: We opted to retain helices 1, 3–5 as well as the critically helix 12, which undergoes conformational changes to mediate coactivator binding (Rastinejad, Huang et al. 2013). By retaining this region, we aimed to preserve co-factor interactions, while abolishing the receptor’s ligand-binding function. The structural visualization of this

truncation strategy is now included in Fig. 2a of the revised manuscript, and we have included additional text in the revised manuscript in lines 208-213 to further explain our rationalization.

The interpretation of the structural studies on nuclear receptors is over-simplistic. The ligand binding domain, as a well-folded structure, binds ligand, interacts with transcriptional co-factors, and interacts with each other (dimerization). (PMID: 37356665) The authors cannot rule out the possibility that the delta295-422 NHR-49 mutant has lost its dimerization function, or created new surface of the protein that permitted aberrant protein interactions. The presentation of Fig. 2a was misleading. It gave the impression that the remaining helices of the LBD adopted exactly the same conformation in the absence of the blackened helices. Caution should also be exercised when interpreting the alphafold predicted structure of NHR-49. The single conformation in the predicted structure has helix 12 in a position that does not entirely resemble its known position in other ligand-bound, activated LBD of nuclear receptors. Instead, the NHR-49 helix 12 appeared to block the well-studied co-activator binding surface. The structure of the nematode AceDAF-12 may serve as an additional reference (PMID: 22170062).

We appreciate the structural insight and agree that our interpretation of the ligand binding domain structure was over-simplified. To this end, we have de-emphasized this aspect in the re-revised manuscript. The underlying goal of Fig. 2 was to rule out ligand binding as a means of silencing the transcriptional activity of NHR-49, rather than focus on structural features of the nuclear receptor. Several groups have already reported structures for its close sequence homolog in HNF4A (PMID:12193589, 12220494, 23485969, 27496803) while complementary *in silico* modeling have been reported for NHR-49 (PMID:27618178).

We fully agree with the reviewer that loss of dimerization or the introduction of aberrant protein interactions cannot definitely be rule out with respect to the NHR-49 Δ 295-422 mutant. Yet, we are confident that removing a majority of the ligand binding domain impairs its ability to bind its dedicated ligand (although we can't rule out that we have fully abolished ligand binding). As a little background, we were initially concerned that this truncation had abolished all endogenous receptor function due to loss in the basal transcription of direct NHR-49 targets when transiently expressed into the *nhr-49* mutant background. However, this truncated form of NHR-49 both rescued lifespan defects typical of the *nhr-49* mutant and could both activate and repress transcriptional targets of NHR-49. While we debated whether to remove this strain from the manuscript, we ultimately believe it remains valuable to the community and may serve as a useful reference for others investigating mutations within the ligand-binding domain of NHR-49.

As mentioned above, we have de-emphasized the structural predictions for this ligand binding domain mutant and focus on what we believe is the most compelling data for the mutant (its ability to retain transcriptional activation and repression of a direct NHR-49 targets during fasting and refeeding). To this end, we replaced Fig. 2a which displayed the predicted NHR-49 structure with “blacked-out” helices corresponding to the truncated region. In its place, we provide a more simplistic schematic of both the full length and ligand binding domain truncation. Additionally, we replace the former text describing structural predictions based on the removal and retention of particular ligand binding domain helices with a more simplified description between lines 217-220. “Based on domain architecture comparison with HNF4 α ^{58,59}, we engineered an NHR-49 truncation (Δ 295-422) to disrupt ligand binding by removing a majority of the ligand binding domain, while retaining its analogous N-terminal, DNA-binding, hinge, and C-terminal domains (Fig. 2a).”

It is hard to interpret the results from the limited proteolysis assay (reviewer comment Fig. 6) because the same three fragments present in wildtype and the delta295-422 mutant were shown. Which region of NHR-49 did these three fragments represent? Does any of them correspond to part of the LBD? Should there be a fragment that corresponded to 295-422, which is only present in the wildtype sample? The incomplete results of this assay casted doubt on its utility as a readout of structural integrity, especially the LBD.

We included the limited proteolysis assay in the reviewer response for the sake of transparency and to ensure that all relevant data was available for evaluation. As mentioned in our initial comments, this assay represents a broad interpretation of trypsin digestion patterns observed in the various transgenic NHR-49 mutant strains generated in-house.

These transgenic strains possess YFP tags at their C-terminus and are immunoblotted with a polyclonal anti-GFP rabbit antibody. Thus, immuno-positive bands above 27 kDa can be interpreted as C-terminal fragments of NHR-49, the region where the ligand-binding domain is located. The detection of similar proteolytic digestion patterns across all NHR-49 mutant strains suggests that the C-terminal regions of the different NHR-49 mutants retain some degree of structural similarity based on the accessibility of protease digestion sites. However, we agree that this is not definite evidence and could be confusing to the reader. Thus, we have not included these blots in the manuscript but wanted to provide as much information to the reviewers and editor as possible. If either the editor and/or the reviewer feel that the manuscript could benefit from this limited proteolysis, we are happy to include it in the revised figures.

Response Line 789: On this foundation, we employed DHS-3::GFP to track lipid droplet dynamics within intestinal cells, the major site of lipid storage in *C. elegans*, and have clarified as such at lines 102-109 in the revised manuscript. We note that DHS-3::GFP has been the community standard for over a decade, used in at least 15 independent publications.

Manuscript Line 102-103: "Moving forward, we used a fluorescence-based system to monitor, in real time, the dynamics of TAG-enriched lipid droplets in living animals during fasting and refeeding."

The original query was not about the validity of using DHS-3::GFP as a lipid droplet marker. What was problematic was the use of GFP fluorescence intensity (y-axis label, Fig. 1d, 1h, 4c, 4h) as a measurement of lipid droplet dynamics, which should mean the size and number of lipid droplets. The change in GFP fluorescence could be due to transcriptional regulation of dhs-3 (as shown by reviewer comment Fig. 7), which may or may not be the result of a change in the size and number of lipid droplets. The authors should directly measure the size and number of lipid droplets in control and treatment groups.

We thank the reviewer for this clarification and perhaps used the wrong terminology. Our goal was to understand how exhaustion and replenishment of lipid reserves during fasting and refeeding impacted age determination. Thus, throughout the re-revised text, we have removed “lipid droplet dynamics” and more accurately stated that we aimed to monitor the “temporal dynamics of bulk neutral lipid availability”. Yet, we agree with the reviewer that further characterization of lipid

droplet size and number during this fasting and refeeding paradigm would strengthen the manuscript and further validate the flow cytometry and TLC analysis with respect to this new transient fasting paradigm.

To this end, we went back to the bench and now include new data within the re-revised manuscript, which measures average volume and number of DHS-3-positive droplets during fasting and refeeding. We include this new data in the re-revised manuscript with new fluorescent micrographs in Fig. 1c as well as quantifications of lipid droplet number and average volume in the Extended Data Fig. 1c,d. All associated data and measurements can be found in the raw data excel file associated with the re-revised manuscript and a detailed description of the imaging, processing and quantifications can be found in the methods section at lines 968-975. We have also included additional text in the re-revised manuscript to describe this new data in relation to the flow cytometry and TLC at lines 108-116.